# Expert Merging in Sparse Mixture of Experts with Nash Bargaining

**Dung V. Nguyen**[1]*   **Anh T. Nguyen**[2]*   **Minh H. Nguyen**[3]   **Luc Q. Nguyen**[2]   **Shiqi Jiang**[1]
**Ethan Fetaya**[4]   **Linh Duy Tran**[5]†   **Gal Chechik**[4]†   **Tan M. Nguyen**[1]†

[1]Department of Mathematics, National University of Singapore
[2]Viettel AI, Viettel Group
[3]Faculty of Mathematics and Informatics, Hanoi University of Science and Technology
[4]Bar Ilan University, Israel
[5]AI Imaging Team, Data Solution Department, FPT Software Japan
{dungnv, shiqijiang}@u.nus.edu,tanmn@nus.edu.sg
{ethan.fetaya, gal.chechik}@biu.ac.il
 minh.nh232331M@sis.hust.edu.vn
{anhnt21, lucnq1}@viettel.com.vn
linhtd32@fpt.com

## Abstract

Existing expert merging strategies for Sparse Mixture of Experts (SMoE) typically rely on input-dependent or input-independent averaging of expert parameters, but often lack a principled weighting mechanism. In this work, we reinterpret expert merging through the lens of game theory, revealing cooperative and competitive dynamics among experts. Based on this perspective, we introduce Nash Merging of Experts (NAMEx), a novel framework that incorporates Nash Bargaining into the merging process, enabling more balanced and efficient collaboration among experts. Additionally, we incorporate complex momentum into NAMEx to accelerate expert propagation with theoretical guarantees for convergence. Extensive experiments across language modeling, text classification, image classification, and zero-shot robustness under data corruption show that NAMEx consistently outperforms competing methods while integrating seamlessly with popular MoE architectures. Finally, we demonstrate NAMEx's scalability by applying it to large-scale systems, including Qwen1.5-MoE (14B) and DeepSeek-MoE (16B), where it proves effective in both zero-shot and fine-tuning settings. The code is publicly available at: https://github.com/anh147/NAMEx.

## 1 Introduction

Scaling up neural networks without proportional increases in computational cost is a key goal in modern deep learning. Sparse Mixture of Experts (SMoE) architectures offer a powerful solution: they selectively activate only a subset of expert modules for each input, thereby maintaining high capacity while preserving computational efficiency. Building on the classical Mixture of Experts (MoE) framework (Jacobs et al., 1991), SMoE leverages a dynamic gating mechanism to determine which experts participate in processing a given input. This sparsity allows extremely large models to be trained efficiently and has shown promise across natural language processing (Shazeer et al., 2017; Liu et al., 2024a; Yang et al., 2024), and computer vision (Ruiz et al., 2021; Puigcerver et al., 2024a; Nielsen et al., 2025) applications.

A core component of SMoE is the routing mechanism, which dynamically determines expert assignments. Significant efforts have focused on improving routing stability, load balancing, and expressiveness. For example, StableMoE (Dai et al., 2022) introduces a two-stage strategy to reduce routing variance; SMEAR (Muqeeth et al., 2024) proposes soft parameter merging via weighted averaging to bypass discrete selection; and HyperRouter (Do et al., 2023) uses hypernetworks to generate router parameters. Meanwhile, SoftMoE (Puigcerver et al., 2024a) blends sparse and dense routing, and patch-level routing (Chowdhury et al., 2023) improves sample efficiency in visual tasks. Beyond these, Switch Transformer (Fedus et al., 2022) simplifies routing to top-1 expert

---

*Co-first authors. †Co-last authors. Correspondence to: dungnv@u.nus.edu & tanmn@nus.edu.sg

selection with auxiliary load-balancing losses for stable large-scale training, GShard (Lepikhin et al., 2021a) introduces expert capacity constraints and scalable sharding strategies, and BASE Layers (Lewis et al., 2021) formulates routing as a balanced assignment problem to improve expert utilization. Hash-based routing (Roller et al., 2021) replaces learned routers with deterministic hashing to reduce overhead, while DSelect-k (Hazimeh et al., 2021) provides a differentiable relaxation for sparse expert selection to improve gradient flow. Collectively, these approaches span hard, sparse routing with explicit balancing objectives to soft and differentiable formulations, reflecting a broad design space aimed at enhancing stability, scalability, and specialization in SMoE systems.

Beyond routing, a complementary yet underexplored direction is *expert merging*. Instead of selecting a subset of experts per input, merging aims to combine all expert parameters into a unified model, either during training or at inference. This approach is especially appealing when deployment or memory constraints demand a single-expert representation. Merging is particularly valuable in autoregressive models (Zhong et al., 2024) and cross-domain transfer settings (Chen et al., 2022). However, most current merging techniques, such as soft-merging (Muqeeth et al., 2024; Zhong et al., 2024) and top-$k$ aggregation (He et al., 2023; Li et al., 2024), rely on heuristic weighting schemes that ignore the intricate dynamics between experts.

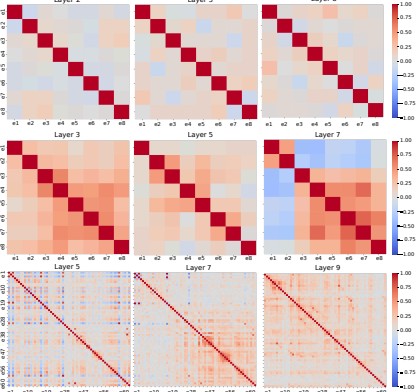

Figure 1: Cosine similarity of expert outputs in Swin-MoE (Liu et al., 2021) (top), Switch-Transformer (Fedus et al., 2022) (middle), and Qwen-MoE (Yang et al., 2024) (bottom). Swin-MoE shows stable mid-layer features, Switch-Transformer exhibits dynamic routing at Layer 8, and Qwen-MoE yields robust final representations at Layer 9–highlighting diverse expert interaction patterns.

Recent work has begun to address this limitation. (Nguyen et al., 2025) introduces a curvature-aware merging scheme, namely Curvature-aware merging of experts (CAMEx), that uses natural gradients to account for non-Euclidean geometry in parameter space. A variant, corresponding to the dynamic merging (Dynamic-Merg) mechanism in the CAMEx paper, which we refer to as Expert-Propagation CAMEx (EP-CAMEx), propagates a base expert across layers to promote inter-layer communication. However, despite its elegance, EP-CAMEx underperforms its static variant, likely due to insufficient coordination among expert contributions. This motivates a deeper question: *Can we interpret expert merging as a structured interaction among experts, rather than just a linear average?*

**Contribution.** In this paper, we frame expert merging as a cooperative-competitive game among experts. Drawing inspiration from multi-task learning, we adopt the *Nash Bargaining Solution* (NBS) (Nash, 1950) to derive merging coefficients from first principles based on each expert's contribution. Our method, named *Nash Merging of Experts* (NAMEx), treats expert domain vectors as utility functions in a bargaining game. By solving for the optimal agreement point, NAMEx ensures a fair and efficient merging process that reflects expert alignment and divergence.

To address the slow convergence of EP-CAMEx, we further integrate *complex momentum* (Lorraine et al., 2022) into the propagation process. This enhancement accelerates convergence while preserving stability, especially when expert interactions include adversarial or conflicting dynamics. We theoretically prove the convergence of NAMEx under mild conditions and provide a spectral radius-based bound for the convergence rate of NAMEx-Momentum. Our contribution is three-fold:

1. We develop NAMEx, a new expert merging method that integrates the Nash Bargaining optimization framework of (Navon et al., 2022) into EP-CAMEx (Nguyen et al., 2025), improving expert propagation at each SMoE layer.

2. We incorporate complex momentum into our NAMEx to enhance the stability and convergence speed of expert propagation across layers and provide theoretical guarantees.

3. We demonstrate that quaternion momentum presents a promising future direction for further improving expert merging.

Comprehensive experiments across diverse tasks–including WikiText-103 language modeling (Merity et al., 2017), GLUE text classification finetuning (Wang et al., 2018), and ImageNet-1k image classifica-

Figure 2: Architecture overview of (a) CAMEx (Nguyen et al., 2025), (b) Expert-Propagation CAMEx (Nguyen et al., 2025), and (c) our proposed merging method, NAMEx.

tion and zero-shot robustness under data corruption (Deng et al., 2009)–demonstrate the effectiveness of our approach, achieving superior accuracy compared to baseline methods while preserving advantages in computational efficiency. Moreover, we establish NAMEx's scalability by deploying it on large systems such as Qwen1.5-MoE (14B) and DeepSeek-MoE (16B), where it delivers strong performance in both zero-shot and fine-tuning scenarios.

**Organization.** Section 2 reviews SMoE, CAMEx, and Nash Bargaining; Section 3 introduces NAMEx and its momentum extension; Section 4 presents experiments with ablations; Section B discusses related work; and Section 6 concludes with limitations.

## 2 BACKGROUND

### 2.1 SPARSE MIXTURE OF EXPERTS

The Mixture of Experts (MoE) framework enables modular neural computation by combining multiple specialized sub-networks (*experts*) through a gating function (Jacobs et al., 1991). The Sparse Mixture of Experts (SMoE) variant enhances scalability by activating only a small subset of experts per input, significantly reducing computation during training and inference (Shazeer et al., 2017; Lepikhin et al., 2021b; Fedus et al., 2022).

Let $\mathbf{x} \in \mathbb{R}^d$ be an input and $f_i(\mathbf{x})_{i=1}^N$ denote expert outputs. A gating network computes weights $s_i(\mathbf{x})$ such that:

$$s_i(\mathbf{x};\theta_g) \geq 0, \quad \sum_{i=1}^N s_i(\mathbf{x};\theta_g) = 1, \quad F(\mathbf{x}) = \sum_{i=1}^N s_i(\mathbf{x};\theta_g) f_i(\mathbf{x}). \tag{1}$$

SMoE improves model capacity without linearly scaling compute, making it a central design in recent large-scale architectures.

### 2.2 CURVATURE-AWARE MERGING OF EXPERTS

CAMEx uses natural gradients to align merged experts more closely with the geometry of the parameter space, enhancing both pre-training and fine-tuning processes (Nguyen et al., 2025). Hence, CAMEx generalizes popular expert merging methods such as SMEAR (Muqeeth et al., 2024) and Lory (Zhong et al., 2024) and can be formulated as the following natural gradient-like merging scheme:

$$\hat{\mathbf{E}}_m^{(l)} = \mathbf{E}_m^{(l)} + \eta \sum_{i=1}^N \mathbf{M}_i^{(l)} \cdot (s_i^{(l)} * \tau_i^{(l)}), \tag{2}$$

where $\tau_i^{(l)} = \mathbf{E}_i^{(l)} - \mathbf{E}_m^{(l)}$ is the domain-vector of the $i$-th expert, representing its deviation from the base expert. $\mathbf{E}_i^{(l)}$, $\mathbf{E}_m^{(l)}$, and $\hat{\mathbf{E}}_m^{(l)}$ denote the weights of the $i$-th expert, the base expert, and the resulting merged expert that processes the input, respectively. Here, the base expert $\mathbf{E}_m^{(l)}$ is shared between tokens in layer $l$, just like in DeepSeek-V2 (Liu et al., 2024a) and V3 (Liu et al., 2024b). $\eta > 0$ denotes the stepsize for updating the base expert, and $\mathbf{M}_i^{(l)}$ is the curvature matrix for the $i$-th expert. EP-CAMEx is an extension of CAMEx, in which the base expert $\mathbf{E}_m^{(0)}$ is initialized at the first layer, and $\mathbf{E}_m^{(l)}$ and $\hat{\mathbf{E}}_m^{(l)}$ are updated at subsequent layers as follows:

$$\begin{cases} \mathbf{E}_m^{(l+1)} &= \mathbf{E}_m^{(l)} + \dfrac{\gamma}{N} \sum_{i=1}^N \mathbf{M}_i^{(l)} \cdot \tau_i^{(l)}, \\ \hat{\mathbf{E}}_m^{(l+1)} &= \mathbf{E}_m^{(l+1)} + \eta \sum_{i=1}^N \mathbf{M}_i^{(l+1)} \cdot (s_i^{(l+1)} * \tau_i^{(l+1)}). \end{cases} \tag{3}$$

Here, $\gamma > 0$ denotes the step size for the propagation of the base expert $\mathbf{E}_m^{(l)}$. In the first equation of system (3) above, if we view each domain-vector $\tau_i^{(l)}$ as a "gradient direction" attempting to pull the base expert toward the corresponding expert's domain, then the formulation can be interpreted as a dynamical system that updates $\mathbf{E}_m^{(l)}$ using Multiple-Gradient Descent Algorithm (MGDA) (Désidéri, 2012) to minimize the distance between $\mathbf{E}_m$ and $i$-th domain. Consequently, this can be framed as a multi-objective optimization or multi-task learning problem. We illustrate both CAMEx and EP-CAMEx in Figure 2(a) and (b).

### 2.3 NASH BARGAINING IN MULTI-TASK LEARNING

The Nash Bargaining Solution (NBS) (Nash, 1950) is a foundational concept in cooperative game theory, describing how multiple agents can reach a fair and Pareto-optimal agreement. A bargaining problem is typically defined by a agreement set of outcomes $\mathcal{S} \subseteq \mathbb{R}^N$ and a disagreement point $\mathbf{d} \in \mathbb{R}^N$, which specifies the utility each player receives if no agreement is reached. The NBS selects an outcome $\mathbf{u}^* \in \mathcal{S}$ that maximizes the product of individual gains over the disagreement point:

$$\mathbf{u}^* = \operatorname*{argmax}_{\mathbf{u} \in \mathcal{S}} \prod_{i=1}^{N} (u_i - d_i). \tag{4}$$

The disagreement point in the Nash Bargaining Problem is the fallback outcome each player receives if no agreement is reached. It serves as a baseline against which any cooperative agreement is measured, shaping the set of feasible solutions. Players often consider their disagreement point strategically, as improvements to it can strengthen their bargaining position and influence the final outcome.

Recent work by (Navon et al., 2022) demonstrates that multi-task learning (MTL) can be naturally framed as a bargaining game. In this setting, each task corresponds to a player, and the goal is to determine a shared parameter update direction $\Delta\boldsymbol{\theta}$ that benefits all tasks. The agreement set is typically constrained to a unit ball $B_\epsilon = \{\Delta\boldsymbol{\theta} \mid \|\Delta\boldsymbol{\theta}\| \leq \epsilon\}$, while the disagreement point is set to zero, indicating no parameter update. Each task $i$ provides a utility function

$$u_i(\Delta\boldsymbol{\theta}) = \boldsymbol{\tau}_i^\top \Delta\boldsymbol{\theta}, \tag{5}$$

where $\boldsymbol{\tau}_i$ is the gradient of the task-specific loss with respect to the model parameters. Under the assumption that these gradients are linearly independent, the NBS yields the optimal update direction:

$$\Delta\boldsymbol{\theta} = \sum_{i=1}^{N} \alpha_i \boldsymbol{\tau}_i, \quad \text{where } \mathbf{G}^\top \mathbf{G}\boldsymbol{\alpha} = 1/\boldsymbol{\alpha}, \tag{6}$$

with $\mathbf{G} = [\boldsymbol{\tau}_1, ..., \boldsymbol{\tau}_N]$ and $1/\boldsymbol{\alpha}$ denoting element-wise reciprocals.

This formulation provides a principled way to resolve conflicting gradients, balancing cooperative and adversarial dynamics among tasks. In this paper, we leverage this framework to reinterpret expert merging in SMoE and particularly, CAMEx, as a bargaining game among experts, where each domain vector $\boldsymbol{\tau}_i^{(l)}$, $i = 1, ..., N$, plays the role of a task gradient.

## 3 NASH MERGING OF EXPERTS

Building on the foundations of CAMEx and Nash Bargaining, we now introduce NAMEx–a novel method for merging experts in SMoE via Nash Bargaining. Rather than treating expert merging as a simple averaging task, NAMEx models it as a multi-agent bargaining game, where each expert proposes a directional update, i.e., its domain vector, and the merged expert is obtained through a principled aggregation reflecting both cooperation and competition. To address slow convergence in existing propagation methods like EP-CAMEx, we further introduce *complex momentum* into NAMEx, enabling faster and more stable propagation through SMoE layers. An overview of our approach is shown in Figure 2(c).

### 3.1 MERGING EXPERTS AS A BARGAINING GAME

Setting $\Delta\boldsymbol{\mathcal{E}}^{(l)}$ as an update direction for $\mathbf{E}_m^{(l)}$ of the $l$-th layer in the first equation of system (3), we adjust the expert-propagating updating step in EP-CAMEx as follows:

$$\begin{cases} \mathbf{E}_m^{(l+1)} &= \mathbf{E}_m^{(l)} + \gamma\Delta\boldsymbol{\mathcal{E}}^{(l)}, \\ \hat{\mathbf{E}}_m^{(l+1)} &= \mathbf{E}_m^{(l+1)} + \eta\sum_{i=1}^{N} \mathbf{M}_i \cdot (s_i^{(l+1)} * \boldsymbol{\tau}_i^{(l+1)}). \end{cases} \tag{7}$$

Like CAMEx, we view the domain-vectors $\boldsymbol{\tau}_i^{(l)} = \mathbf{E}_i^{(l)} - \mathbf{E}_m^{(l)}$ as analogous to a gradient step that pulls $\mathbf{E}_m$ toward $\mathbf{E}_i$'s domain. However, different from the formulation of EP-CAMEx in system (3), we remove the curvature matrix in the first equation to align with the Bargaining Game given by Algorithm 1 in (Navon et al., 2022). Our goal now is to find an optimal update vector $\Delta\boldsymbol{\mathcal{E}}^{(l)}$ which benefits all experts, i.e., finding $\boldsymbol{\alpha}^{(l)} = [\alpha_1^{(l)}, \alpha_2^{(l)}, ..., \alpha_N^{(l)}]$ to aggregate the domain-vectors $\boldsymbol{\tau}_i^{(l)}$ into $\Delta\boldsymbol{\mathcal{E}}^{(l)}$ as in Eqn. 7. We hypothesize that experts in SMoE engage in *mixed games* comprising both cooperative and competitive dynamics.

**Layer-Wise Expert Interaction Dynamics.** Following the analysis protocol described in (Lo et al., 2025), we observe that expert behavior varies by layer and architecture (see Figure 1), revealing both cooperative and adversarial patterns. For instance, in Swin-MoE (Liu et al., 2021), middle layers show high inter-expert similarity, while Qwen-MoE (Yang et al., 2024) concentrates alignment in deeper layers. This motivates a dynamic, layer-wise approach to merging–exactly what NAMEx provides. Please refer to Figure 8 and Figure 6 in Appendix G.4 for more analysis on the dynamic of expert interaction. For comparison regarding expert interaction patterns under the impact of Load Balancing loss, please refer to Figure 9 in Appendix G.4.

Adapting the bargaining game's formulation in (Navon et al., 2022), NAMEx solves the following problem:

> [**Bargaining of Expert Merging (BEM) Problem**] *Given an experts-merging problem with the set of expert parameters $\{\mathbf{E}_1, \mathbf{E}_2, ..., \mathbf{E}_N\}$ and the base expert's parameter $\mathbf{E}_m$, find an update vector $\Delta\boldsymbol{\mathcal{E}}$ within a ball $B_\epsilon$ of radius $\epsilon$ centered at zero, i.e., $B_\epsilon = \{\Delta\boldsymbol{\mathcal{E}} \mid \|\Delta\boldsymbol{\mathcal{E}}\| \leq \epsilon\}$.*

Inspired by (Navon et al., 2022), in this bargaining problem, we set the disagreement point to 0, corresponding to not updating $\mathbf{E}_m$. Similar to Eqn. 5, the utility function for each expert is defined as $u_i(\Delta\boldsymbol{\mathcal{E}}) = \boldsymbol{\tau}_i^\top \Delta\boldsymbol{\mathcal{E}}$, where $\boldsymbol{\tau}_i$ is the domain-vector for expert $i$, representing its deviation from the base expert and capturing its unique contribution to the merging process. Here, $\Delta\boldsymbol{\mathcal{E}}$ is equivalent to $\Delta\boldsymbol{\theta}$ in Eqn. 5. We have the following mild axiom on the Nash bargaining solution.

**Axiom 3.1** (Pareto optimality of Nash bargaining solution (Nash, 1950)). *The selected agreement must be Pareto efficient, i.e. no other feasible outcome should exist that improves one player's utility without reducing the utility of at least one other player.*

Under Axiom 3.1, the solution to the BEM Problem above is given by the following lemma.

**Lemma 3.2** (Nash Solution of Expert Merging). *Let $\mathbf{G}$ denote the $d \times N$ matrix whose columns are the domain-vectors $\tau_i$. The solution to*

$$\arg \max_{\Delta\boldsymbol{\mathcal{E}} \in B_\epsilon} \sum_{i=1}^{N} \log(\Delta\boldsymbol{\mathcal{E}}^\top \tau_i) \tag{8}$$

*is (up to scaling) $\Delta\boldsymbol{\mathcal{E}}^* = \sum_{i=1}^{N} \alpha_i \tau_i$, where $\alpha \in \mathbb{R}_+^N$ satisfies $\mathbf{G}^\top \mathbf{G}\alpha = 1/\alpha$, with $1/\alpha$ being the element-wise reciprocal.*

Note that, under Axiom 3.1, it can be proven that the Nash solution to the bargaining problem is not dominated by other solutions. A proof sketch for Lemma 3.2 is provided in Appendix C.1.

## 3.2 NAMEx as the Nash Solution of Expert Merging

We now formally define NAMEx as the Nash Bargaining Solution to the BEM problem.

**Definition 3.3** (NAMEx: Nash Merging of Experts). *Let $\{\mathbf{E}_1^{(l)}, ..., \mathbf{E}_N^{(l)}\}$ be the expert parameters and let $\mathbf{E}_m^{(l)}$ denote the base expert at layer $l$. Define the domain-vectors as $\boldsymbol{\tau}_i^{(l)} = \mathbf{E}_i^{(l)} - \mathbf{E}_m^{(l)}$, and let $\mathbf{G}^{(l)} = [\boldsymbol{\tau}_1^{(l)}, ..., \boldsymbol{\tau}_N^{(l)}]$ be the matrix formed by stacking these vectors. The NAMEx update direction $\Delta\boldsymbol{\mathcal{E}}^{(l)}$ is defined as:*

$$\Delta\boldsymbol{\mathcal{E}}^{(l)} = \sum_{i=1}^{N} \alpha_i^{(l)} \boldsymbol{\tau}_i^{(l)},$$

*where $\boldsymbol{\alpha}^{(l)} \in \mathbb{R}_+^N$ satisfies the Nash Bargaining equation: $\mathbf{G}^{(l)\top} \mathbf{G}^{(l)} \boldsymbol{\alpha}^{(l)} = 1/\boldsymbol{\alpha}^{(l)}$, with $1/\boldsymbol{\alpha}^{(l)}$ denoting the element-wise reciprocal. The NAMEx update then proceeds by plugging NAMEx update*

*direction into Eqn. 7:*

$$\begin{cases} \mathbf{E}_m^{(l+1)} & = \mathbf{E}_m^{(l)} + \gamma \sum_{i=1}^{N} \alpha_i^{(l)} \boldsymbol{\tau}_i^{(l)}, \\ \hat{\mathbf{E}}_m^{(l+1)} & = \mathbf{E}_m^{(l+1)} + \eta \sum_{i=1}^{N} \mathbf{M}_i \cdot (s_i^{(l+1)} * \boldsymbol{\tau}_i^{(l+1)}), \end{cases} \tag{9}$$

*where $\gamma, \eta \in \mathbb{R}_+$ are step-size coefficients, $\mathbf{M}_i$ is the curvature matrix for expert $i$, and $s_i^{(l+1)}$ are the routing weights at layer $l+1$.*

We summarize the implementation of NAMEx in Algorithm 1.

**Dissecting NAMEx.** We now discuss the behavior of NAMEx by studying the Nash Solution of the BEM Problem. First, if all $\tau_j$ are orthogonal, we obtain $\alpha_j = \dfrac{1}{\|\tau_j\|}$ and $\Delta\mathcal{E} = \sum_{j=1}^{N} \alpha_j \tau_j$, which is a scale-invariant solution. When $\tau_j$ are not orthogonal, we obtain

$$\alpha_j \|\tau_j\|^2 + \sum_{i \neq j} \alpha_i \tau_i^\top \tau_j = \frac{1}{\alpha_j}. \tag{10}$$

Lemma 3.2 allows us to calculate the optimal update direction $\Delta\mathcal{E}$ for an expert-propagation step at $l$-th layer as $\Delta\mathcal{E}^{(l)} = \sum_{i=1}^{N} \alpha_i \tau_i^{(l)}$.

Furthermore, assuming that EP-CAMEx obeys the update law in Eqn. 7 in (Désidéri, 2012), the norm $\|\tau_j\|$ is (nearly) identical between domain vectors, we can view the expert update step in (Nguyen et al., 2025) as a trivial solution (with a scaling factor) of Lemma 3.2, ignoring the interaction between experts. While they also apply curvature matrices to the expert propagating step, the learned curvature matrices provide no additional information about other experts. Thus, the conclusion still holds.

---

**Algorithm 1** Expert Merging via Nash Bargaining

1: **Initialize:** Model $M$ with $L$ SMoE layers, number of experts $N$, $\gamma, \eta \in \mathbb{R}^+$
2: $\quad H^{(t)} \in \mathbb{R}^{B \times S \times N}$: router logits at layer $t$
3: $\quad T^{(t)} \in \mathbb{R}^{B \times S \times D}$: token sequence at layer $t$
4: **for** $t = 1$ to $L$ **do**
5: $\quad$ **for** $i = 1$ to $N$ **do**
6: $\quad\quad \tau_i^{(t)} \leftarrow \mathbf{E}_i^{(t)} - \mathbf{E}_m^{(t)}$
7: $\quad$ **end for**
8: $\quad \mathbf{G}^{(t)} \leftarrow [\tau_1^{(t)}, \tau_2^{(t)}, ..., \tau_N^{(t)}]$
9: $\quad$ Solve for $\alpha$: $\left(\mathbf{G}^{(t)}\right)^\top \mathbf{G}^{(t)} \alpha = 1/\alpha$
10: $\quad \mathbf{E}_m^{(t+1)} \leftarrow \mathbf{E}_m^{(t)} + \gamma \sum_i \tau_i^{(t)} \alpha_i$
11: $\quad \mathbf{E}_m^{(t+1)} \leftarrow \mathbf{E}_m^{(t+1)} + \eta \sum_i H_i^{(t)} \cdot \tau_i^{(t)}$
12: **end for**

**Algorithm 2** NAMEx-Momentum

1: **Input:** $\gamma \in \mathbb{R}^+$, $\beta \in \mathbb{C}$, $\mu^{(0)} \in \mathbb{C}^d$, $\mathbf{E}^{(0)} \in \mathbb{R}^d$
2: **for** $j = 1$ to $L-1$ **do**
3: $\quad$ **for** $i = 1$ to $N$ **do**
4: $\quad\quad \boldsymbol{\tau}_i^{(j)} \leftarrow \mathbf{E}_i^{(j)} - \mathbf{E}_m^{(j)}$
5: $\quad$ **end for**
6: $\quad \mathbf{G}^{(j)} \leftarrow [\boldsymbol{\tau}_1^{(j)}, ..., \boldsymbol{\tau}_N^{(j)}]$
7: $\quad$ Solve $\boldsymbol{\alpha}$ from: $(\mathbf{G}^{(j)})^\top \mathbf{G}^{(j)} \boldsymbol{\alpha} = 1/\boldsymbol{\alpha}$
8: $\quad \Delta\mathcal{E}^{(j)} \leftarrow \sum_i \boldsymbol{\tau}_i^{(j)} \alpha_i$ {*Same update as NAMEx*}
9: $\quad \mu^{(j+1)} \leftarrow \beta \mu^{(j)} + \Delta\mathcal{E}^{(j)}$
10: $\quad \mathbf{E}^{(j+1)} \leftarrow \mathbf{E}^{(j)} + \Re(\gamma \mu^{(j+1)})$
11: $\quad$ *// Optional: Add residual alignment or router term if needed*
12: **end for**

---

In Eqn. 10, we can consider $\sum_{i \neq j} \alpha_i \tau_i^\top \tau_j = (\sum_{i \neq j} \alpha_i \tau_i^\top) \tau_j$ as the interaction between the $j$-th expert and the other experts. If the sum is positive, the experts cooperate, and the other domain-vectors aid the $j$-th expert. $\alpha_j$ decreases in this case. If the sum is negative, the other experts hamper the $j$-th expert, i.e., an adversarial behavior between experts, and therefore, $\alpha_j$ increases to ensure that Eqn. 10 holds.

## 3.3 INTEGRATING MOMENTUM INTO EXPERT MERGING

We hypothesize that one reason for EP-CAMEx's inferior performance compared to CAMEx is its reliance on a fixed number of update steps, constrained by the model's layer count. This limitation hinders the convergence of the base expert in later stages, leading to suboptimal performance. To mitigate this, we introduce momentum to accelerate convergence during optimization. In particular,

we adopt complex momentum (Lorraine et al., 2022), which has been shown to be more robust and effective than standard first-order methods across a wide range of cooperative and adversarial games. By integrating complex momentum into expert merging, we enhance the propagation of expert updates across layers and provide theoretical support for its improved convergence rate.

We present a formal definition NAMEx-Momentum below and summarize an algorithm to implement it in Algorithm 2.

**Definition 3.4** (NAMEx-Momentum: Nash Merging with Complex Momentum). *Let $\{\mathbf{E}_1,...,\mathbf{E}_N\}$ be the expert parameters and $\mathbf{E}_m$ the base expert at a given layer. Define the domain-vectors $\boldsymbol{\tau}_i = \mathbf{E}_i - \mathbf{E}_m$ and the matrix $\mathbf{G} = [\boldsymbol{\tau}_1,...,\boldsymbol{\tau}_N]$. At each iteration $j$, the update direction $\Delta\boldsymbol{\mathcal{E}}^{(j)} = \sum_{i=1}^{N} \alpha_i \boldsymbol{\tau}_i$ is computed where $\boldsymbol{\alpha}$ solves the Nash system:*

$$\mathbf{G}^\top \mathbf{G}\boldsymbol{\alpha} = 1/\boldsymbol{\alpha}.$$

*NAMEx-Momentum uses a complex momentum buffer $\mu^{(j)} \in \mathbb{C}^d$ to accumulate directional updates:*

$$\begin{cases} \mu^{(j+1)} &= \beta\mu^{(j)} + \Delta\boldsymbol{\mathcal{E}}^{(j)}, \\ \mathbf{E}_m^{(j+1)} &= \mathbf{E}_m^{(j)} + \Re(\gamma\mu^{(j+1)}) \\ \hat{\mathbf{E}}_m^{(l+1)} &= \mathbf{E}_m^{(l+1)} + \eta\sum_{i=1}^{N}\mathbf{M}_i \cdot (s_i^{(l+1)} * \boldsymbol{\tau}_i^{(l+1)}), \end{cases} \quad (11)$$

*where $\beta \in \mathbb{C}$ is the momentum coefficient, $\gamma \in \mathbb{R}^+$ is the step size, and $\Re(\cdot)$ denotes the real part.*

We provide a convergence guarantee for NAMEx-Momentum update in Proposition 3.5 below and Theorem C.3 in Appendix C. Their proofs are in Appendix C.3

**Proposition 3.5** (Convergence rate of NAMEx-Momentum). *There exist $\gamma \in \mathbb{R}^+, \beta \in \mathbb{C}$ so Algorithm 2 converges for NAMEx-Momentum.*

## 4 EXPERIMENTAL RESULTS

We evaluate NAMEx and its variants against baseline methods (SMoE, CAMEx, and EP-CAMEx) across diverse tasks: language modeling (WikiText-103 (Merity et al., 2017)), text classification (GLUE (Wang et al., 2018)), and image classification (ImageNet-1K (Deng et al., 2009)). To assess robustness, we include evaluations on corrupted datasets: ImageNet-A, ImageNet-O, and ImageNet-R (Hendrycks et al., 2021c;a). Results, averaged over five random seeds, show that: (1) NAMEx, leveraging Nash bargaining, consistently improves performance on both vision and language benchmarks; and (2) complex momentum provides additional gains. For MLP-based experts, we follow standard practice and merge parameters layerwise (Yadav et al., 2023; Yu et al., 2024; Matena & Raffel, 2022). Experiments are run on a 8×A100 server. Additional details are available in Appendix D.

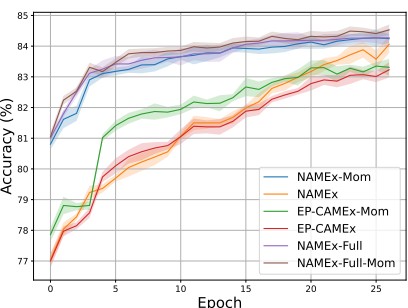

Figure 3: Top-1 Accuracy Evaluation of Swin Transformer Variants. Complex momentum enhances convergence speed and performance of both NAMEx and EP-CAMEx.

To match EP-CAMEx's training time, we fix the bargaining budget to 20 iterations per batch and evaluate two NAMEx variants: (1) NAMEx, which computes $\alpha$ once at the first layer and reuses it, showing strong performance over naive averaging; and (2) NAMEx-Full, which distributes the budget evenly across layers, inspired by (Navon et al., 2022). Update strategies are further discussed in Section 5.

In the tables that follow, NAMEx-Full results are highlighted in *grey*, with the best and second-best scores shown in **bold** and underlined, respectively.

### 4.1 LANGUAGE MODELING

We adopt the experimental setup of (Pham et al., 2024) and (Teo & Nguyen, 2024) for pre-training and evaluating on the WikiText-103 dataset. Table 1 presents the results of our methods on small-scale and medium-scale pre-training tasks using WikiText-103. For both small- and medium-scale pre-training, NAMEx-Full-Mom achieves the lowest validation/test perplexities, outperforming SMoE and CAMEx-based methods. NAMEx variants as well as momentum-equipped variants consistently surpass their counterparts across scales, proving the efficacy of Nash bargaining and momentum integration.

Table 1: Validation and test perplexity on WikiText-103 for small- and medium-scale pretraining.

| Model | Params | Small | | Medium | |
|---|---|---|---|---|---|
| | | Val PPL | Test PPL | Val PPL | Test PPL |
| SMoE (Top-1) | 70M / 216M | $86.64_{\pm.22}$ | $87.79_{\pm.31}$ | $38.60_{\pm.18}$ | $40.51_{\pm.25}$ |
| SMoE (Top-2) | 70M / 216M | $84.26_{\pm.12}$ | $84.81_{\pm.29}$ | $33.76_{\pm.19}$ | $35.55_{\pm.22}$ |
| SMEAR | 70M / 216M | $85.56_{\pm.20}$ | $87.24_{\pm.28}$ | $36.15_{\pm.17}$ | $37.42_{\pm.23}$ |
| CAMEx | 70M / 216M | $83.53_{\pm.19}$ | $84.48_{\pm.26}$ | $35.69_{\pm.15}$ | $36.53_{\pm.21}$ |
| *w/o momentum* | | | | | |
| EP-CAMEx | 70M / 216M | $83.89_{\pm.18}$ | $85.03_{\pm.24}$ | $35.78_{\pm.16}$ | $36.55_{\pm.22}$ |
| NAMEx | 70M / 216M | $83.30_{\pm.21}$ | $84.12_{\pm.29}$ | $35.14_{\pm.19}$ | $36.40_{\pm.27}$ |
| NAMEx-Full | 70M / 216M | $82.85_{\pm.17}$ | $83.16_{\pm.23}$ | $34.92_{\pm.14}$ | $36.21_{\pm.20}$ |
| *w/ momentum* | | | | | |
| EP-CAMEx-Mom | 70M / 216M | $82.90_{\pm.16}$ | $84.05_{\pm.22}$ | $35.09_{\pm.13}$ | $36.16_{\pm.19}$ |
| NAMEx-Mom | 70M / 216M | $82.63_{\pm.15}$ | $83.59_{\pm.21}$ | $34.89_{\pm.12}$ | $35.86_{\pm.18}$ |
| NAMEx-Full-Mom | 70M / 216M | $\mathbf{82.44_{\pm.14}}$ | $\mathbf{82.94_{\pm.20}}$ | $\mathbf{34.25_{\pm.11}}$ | $\mathbf{35.37_{\pm.17}}$ |

Table 2: Performance of T5-base variants on fine-tuning tasks for GLUE. All SMoE variants have 8 experts per layer. Following (Devlin et al., 2019), we conduct experiments on the GLUE benchmark.

| Model | Params | SST-2 | MRPC | CoLA | STSB | RTE | QNLI | MNLI |
|---|---|---|---|---|---|---|---|---|
| Dense | 220M | $93.34_{\pm.15}$ | $89.70_{\pm.11}$ | $58.06_{\pm.15}$ | $89.06_{\pm.22}$ | $74.36_{\pm.27}$ | $92.34_{\pm.14}$ | $86.36_{\pm.15}$ |
| SMoE (Top-1) | 1.0B | $94.26_{\pm.13}$ | $90.87_{\pm.12}$ | $56.78_{\pm.24}$ | $89.44_{\pm.29}$ | $70.75_{\pm.32}$ | $92.07_{\pm.13}$ | $86.38_{\pm.17}$ |
| SMoE (Top-2) | 1.0B | $94.35_{\pm.14}$ | $91.04_{\pm.12}$ | $58.43_{\pm.26}$ | $89.73_{\pm.28}$ | $74.98_{\pm.29}$ | $92.48_{\pm.16}$ | $86.72_{\pm.15}$ |
| CAMEx | 1.0B | $93.80_{\pm.14}$ | $91.16_{\pm.13}$ | $58.57_{\pm.24}$ | $89.47_{\pm.23}$ | $74.72_{\pm.35}$ | $92.60_{\pm.19}$ | $86.44_{\pm.12}$ |
| *w/o momentum* | | | | | | | | |
| EP-CAMEx | 1.0B | $93.69_{\pm.11}$ | $91.01_{\pm.14}$ | $58.29_{\pm.24}$ | $89.92_{\pm.31}$ | $75.81_{\pm.33}$ | $92.17_{\pm.15}$ | $86.94_{\pm.14}$ |
| NAMEx | 1.0B | $94.46_{\pm.12}$ | $92.01_{\pm.14}$ | $58.81_{\pm.36}$ | $90.12_{\pm.33}$ | $75.09_{\pm.22}$ | $92.86_{\pm.17}$ | $86.96_{\pm.12}$ |
| NAMEx-Full | 1.0B | $94.82_{\pm.15}$ | $92.80_{\pm.13}$ | $59.63_{\pm.22}$ | $90.27_{\pm.24}$ | $77.83_{\pm.31}$ | $93.23_{\pm.18}$ | $87.23_{\pm.14}$ |
| *w/ momentum* | | | | | | | | |
| EP-CAMEx-Mom | 1.0B | $94.61_{\pm.17}$ | $92.47_{\pm.13}$ | $59.31_{\pm.25}$ | $90.07_{\pm.23}$ | $76.17_{\pm.36}$ | $92.99_{\pm.13}$ | $86.80_{\pm.15}$ |
| NAMEx-Mom | 1.0B | $94.61_{\pm.14}$ | $93.02_{\pm.16}$ | $58.90_{\pm.41}$ | $90.06_{\pm.36}$ | $77.62_{\pm.37}$ | $93.11_{\pm.10}$ | $87.02_{\pm.14}$ |
| NAMEx-Full-Mom | 1.0B | $\mathbf{95.06_{\pm.12}}$ | $\mathbf{93.27_{\pm.14}}$ | $\mathbf{60.13_{\pm.32}}$ | $\mathbf{90.63_{\pm.27}}$ | $\mathbf{78.15_{\pm.30}}$ | $\mathbf{93.31_{\pm.14}}$ | $\mathbf{87.45_{\pm.11}}$ |

Table 3: Finetuning and zero-shot results on ImageNet-1k and corrupted variants.

| Model | Params | Acc@1 | Acc@5 | INet-O | INet-A | INet-R |
|---|---|---|---|---|---|---|
| SMoE | 50M | $83.15_{\pm.17}$ | $96.71_{\pm.12}$ | $43.34_{\pm.21}$ | $23.72_{\pm.18}$ | $38.02_{\pm.20}$ |
| SMEAR | 50M | $83.15_{\pm.14}$ | $96.91_{\pm.09}$ | $43.35_{\pm.19}$ | $24.14_{\pm.16}$ | $38.16_{\pm.22}$ |
| CAMEx | 50M | $83.29_{\pm.24}$ | $96.95_{\pm.13}$ | $50.69_{\pm.25}$ | $25.45_{\pm.21}$ | $38.37_{\pm.20}$ |
| *w/o momentum* | | | | | | |
| EP-CAMEx | 50M | $83.23_{\pm.25}$ | $96.93_{\pm.16}$ | $50.27_{\pm.28}$ | $24.22_{\pm.17}$ | $37.88_{\pm.23}$ |
| NAMEx | 50M | $84.06_{\pm.28}$ | $97.19_{\pm.18}$ | $50.30_{\pm.27}$ | $25.32_{\pm.15}$ | $38.56_{\pm.19}$ |
| NAMEx-Full | 50M | $84.27_{\pm.24}$ | $97.94_{\pm.14}$ | $50.66_{\pm.22}$ | $25.74_{\pm.16}$ | $38.70_{\pm.18}$ |
| *w/ momentum* | | | | | | |
| EP-CAMEx-Mom | 50M | $83.56_{\pm.12}$ | $97.03_{\pm.11}$ | $50.37_{\pm.20}$ | $33.22_{\pm.24}$ | $38.22_{\pm.19}$ |
| NAMEx-Mom | 50M | $84.28_{\pm.26}$ | $97.94_{\pm.12}$ | $51.22_{\pm.18}$ | $35.05_{\pm.19}$ | $38.82_{\pm.14}$ |
| NAMEx-Full-Mom | 50M | $\mathbf{84.52_{\pm.18}}$ | $\mathbf{98.11_{\pm.15}}$ | $\mathbf{51.34_{\pm.17}}$ | $\mathbf{35.27_{\pm.20}}$ | $\mathbf{38.96_{\pm.13}}$ |

## 4.2 TEXT CLASSIFICATION

We evaluate our method on downstream text classification tasks using the GLUE dataset (Wang et al., 2018), with all models built on the T5-Base backbone. As shown in Table 2, NAMEx-Full-Mom achieves the best results on all tasks. NAMEx consistently outperforms SMoE (Top-1 and Top-2 routing), CAMEx, and EP-CAMEx, highlighting the effectiveness of the Nash bargaining solution. The momentum-based extensions, NAMEx-Mom and EP-CAMEx-Mom, further enhance performance, demonstrating improved robustness and generalization.

## 4.3 IMAGE CLASSIFICATION

In this section, we evaluate our method on image classification tasks using the Swin-Transformer (Liu et al., 2021) and its MoE variant (Hwang et al., 2023). Specifically, we fine-tune Swin-MoE Small on ImageNet-1k, training all models for 30 epochs with a batch size of 96. For each MoE layer, we perform Algorithm 2, where apart from the first MoE layer that an $\mathbf{E}_m$ expert is initialized, all experts

Table 4: Performance comparison across routing strategies and models on MMLU, GSM8K, and ARC benchmarks. Left: original results. Right: fine-tuned DeepSeek - MoE variants on SmolTalk.

| Routing Strategy | Model | Zero-Shot | | | Fine-tuned (SmolTalk) | | |
|---|---|---|---|---|---|---|---|
| | | MMLU | GSM8K | ARC | MMLU | GSM8K | ARC |
| Linear | Deepseek-MoE | 44.77 | 16.53 | 49.15 | 45.21 | 17.10 | 49.50 |
| | EP-CAMEx | 44.85 | 16.63 | 49.26 | 45.33 | 17.24 | 49.62 |
| | **NAMEx-Full** (0 disagreement point) | **44.92** | **16.77** | **49.51** | **45.47** | **17.36** | **49.85** |
| | **NAMEx-Full** (mean disagreement point) | **44.93** | **16.75** | **49.52** | **45.47** | **17.39** | **49.84** |
| Cosine | Deepseek-MoE | 44.95 | 16.70 | 49.30 | 45.34 | 17.25 | 49.60 |
| | EP-CAMEx | 45.05 | 16.81 | 49.40 | 45.45 | 17.32 | 49.73 |
| | **NAMEx-Full** (0 disagreement point) | **45.10** | **16.88** | **49.60** | **45.66** | **17.53** | **49.92** |
| | **NAMEx-Full** (mean disagreement point) | **45.09** | **16.89** | **49.58** | **45.67** | **17.52** | **49.92** |
| Stable-MoE | Deepseek-MoE | 45.80 | 17.50 | 49.90 | 46.17 | 17.63 | 50.28 |
| | EP-CAMEx | 45.88 | 17.62 | 50.00 | 46.25 | 18.10 | 50.45 |
| | **NAMEx-Full** (0 disagreement point) | **45.95** | **17.70** | **50.15** | **46.42** | **18.23** | **50.64** |
| | **NAMEx-Full** (mean disagreement point) | **45.92** | **17.68** | **50.19** | **46.40** | **18.23** | **50.63** |

are merged into $\mathbf{E}_m$. We further evaluate NAMEx on another SMoE architecture ACMoE (Nielsen et al., 2025), we follow ACMoE training configurations, i.e., we train the NAMEx variants on top of the ACMoE backbone for 100 epochs with batchsize 512.

Table 3 shows NAMEx-Mom outperforming all baselines, with NAMEx close behind; even without momentum, NAMEx-Full matches NAMEx-Mom on clean benchmarks, confirming the value of layer-wise Nash solutions. Across distribution shifts (ImageNet-A/O/R (Hendrycks et al., 2021a;c)), NAMEx-Mom achieves the best zero-shot accuracy, with momentum variants showing the strongest gains, especially on ImageNet-A.

In Table 15 of Appendix Appendix G.4, across all ImageNet variants, the NAMEx-based models consistently outperforms the ACMoE Top-1 and Top-2 baselines. In particular, NAMEx-Full and NAMEx-Full-Mom set new best accuracies on both in-distribution metrics (Acc@1 and Acc@5) and out-of-distribution benchmarks (INet-O, INet-A, INet-R). This underlines the strong generalization ability of NAMEx. Even with the same parameter budget, NAMEx variants deliver better robustness to corruptions and distribution shifts.

### 4.4 Zero-shot and Finetuning on DeepSeek-MoE (16B) and Qwen1.5-MoE (14B)

We test NAMEx-Full at scale by integrating it into DeepSeek-MoE (Liu et al., 2024a) (16B parameters, 1 shared expert, and 63 routed experts) and Qwen1.5-MoE (14B parameters), evaluating in both zero-shot and SmolTalk fine-tuned settings. As shown in Table 4 below (for DeepSeek-MoE) and Table 10, 11 in Appendix F.1 (for Qwen1.5-MoE), NAMEx-Full consistently outperforms the baselines and EP-CAMEx across routing strategies and benchmarks (MMLU (Hendrycks et al., 2021b), GSM8K (Cobbe et al., 2021) and ARC (Clark et al., 2018)), demonstrating robust and generalizable gains in expert collaboration.

## 5 Empirical Analysis

**Synthetic Example.** To illustrate how NAMEx encourages balanced expert cooperation, we construct a toy SMoE model with three experts per layer. Utility trade-offs are visualized in a 3D space, where each axis represents the utility of one expert. Figure 11, Appendix G.4, shows that average-based expert merging may fail to reach the Pareto set, whereas NAMEx tends to produce more Pareto-efficient outcomes. This illustrates that NAMEx is not dominated by EP-CAMEx or linear average merging.

Table 5: Impact of varying the argument of $\beta$ on the performance of EP-CAMEx and NAMEx.

| $\phi$ | Model | SST-2 | MRPC | STS-B | RTE |
|---|---|---|---|---|---|
| $\pi/6$ | EP-CAMEx | 94.27 | 92.50 | 89.77 | 76.17 |
| | NAMEx | 94.83 | 92.82 | 89.68 | 76.42 |
| $\pi/12$ | EP-CAMEx | 94.61 | 92.42 | 89.53 | 76.23 |
| | NAMEx | 93.92 | 92.69 | 89.55 | 76.53 |
| $0$ | EP-CAMEx | 93.45 | 92.24 | 89.51 | 72.20 |
| | NAMEx | 93.56 | 91.66 | 89.51 | 75.09 |
| $-\pi/12$ | EP-CAMEx | 93.56 | 91.91 | 89.53 | 76.03 |
| | NAMEx | 93.92 | 92.60 | 90.15 | 75.57 |
| $-\pi/6$ | EP-CAMEx | 94.72 | 91.93 | 89.38 | 72.56 |
| | NAMEx | 94.50 | 92.75 | 89.45 | 76.64 |

**Number of Optimization Steps.** Table 6 shows that smaller step frequencies ($\Delta l = 1, 2, 5$) often improve or match baseline performance ($\Delta l = L$) but at the cost of higher runtime ($0.69s \rightarrow 4.70s$), underscoring a trade-off between accuracy and efficiency.

**Impact of Momentum $\mu$ and Step size $\gamma$.** The results in Table 5 demonstrate the impact of varying the argument $\phi$ of $\beta$ (fixed modulus 0.9). All tasks show declines at $\phi = 0$ (real momentum), suggesting non-zero arguments are critical. NAMEx consistently outperforms EP-CAMEx, reaffirming its robustness. Optimal results require task-specific $\phi$ tuning.

Table 6: Impact of varying the frequency of merging weight update steps on the performance of NAMEx.

| $\Delta l$ | SST-2 | MRPC | STS-B | RTE | Runtime (sec) |
|---|---|---|---|---|---|
| 1 | 94.88 | 92.85 | 90.32 | 77.26 | 4.70 |
| 2 | 94.95 | 92.38 | 90.37 | 76.89 | 2.29 |
| 5 | 95.18 | 92.09 | 90.13 | 77.98 | 1.14 |
| $L$ | 94.46 | 92.01 | 90.12 | 75.09 | 0.69 |

Table 7: Performance comparison between complex momentum and quaternion momentum.

| Model | MRPC | STS-B | RTE |
|---|---|---|---|
| EP-CAMEx-Mom | 92.47 | 90.07 | 76.17 |
| NAMEx-Mom | 93.02 | 90.06 | 77.62 |
| EP-CAMEx-Q | 92.52 | 90.35 | 77.12 |
| NAMEx-Q | **93.24** | **90.72** | **77.86** |

Finally, RTE exhibits higher sensitivity, peaking at $\phi = \pi/12$ and dropping sharply at $\phi = 0$ and $\phi = -\pi/6$, highlighting task-specific $\phi$ dependencies. For analysis on step size $\gamma$, please refer to Figure 10 in Appendix G.4.

**Beyond Complex Momentum.** Quaternions generalize complex numbers with richer 4D dynamics, enabling a quaternion momentum update $z_{t+1} = \beta z_t + \nabla f(x_t)$. While more complex and harder to tune, quaternion momentum can better stabilize high-dimensional optimization and handle rotations. As shown in Table 7, choosing $\beta = 0.8 + 0.3\mathbf{i} + 0.3\mathbf{j} + 0.3\mathbf{k}$ outperforms complex momentum, suggesting multi-buffer momentum is a promising direction with careful hyperparameter tuning.

Table 8: Performance comparison across number of NBS solving iterations for the Linear router in NAMEx-Full config (Qwen1.5-MoE and Deepseek-MoE). All results are slightly improved while maintaining marginal gaps. Throughout experiments, we use 2 CCP iterations per layer for NAMEx-Full (chosen config below).

| Model | No. Iterations | Zero-Shot | | | Fine-tuned (SmolTalk) | | |
|---|---|---|---|---|---|---|---|
| | | MMLU | GSM8K | ARC | MMLU | GSM8K | ARC |
| Qwen1.5-MoE | 2 (Chosen Config) | 61.87 | 60.55 | 50.95 | 62.10 | 61.00 | 51.35 |
| | 5 | 61.70 | 60.55 | 50.94 | 62.20 | 61.05 | 51.31 |
| | 20 | 61.94 | 60.57 | 50.96 | 62.14 | 61.03 | 51.34 |
| | 40 | 61.92 | 60.62 | 51.01 | 62.22 | 61.05 | 51.46 |
| | 60 | 61.81 | 60.48 | 51.08 | 62.15 | 60.98 | 51.39 |
| Deepseek-MoE | 2 (Chosen Config) | 45.05 | 16.86 | 49.58 | 45.63 | 17.47 | 49.92 |
| | 5 | 44.84 | 16.86 | 49.57 | 45.73 | 17.51 | 49.88 |
| | 20 | 45.15 | 16.87 | 49.60 | 45.63 | 17.47 | 49.92 |
| | 40 | 45.16 | 16.93 | 49.66 | 45.73 | 17.55 | 50.03 |
| | 60 | 44.93 | 16.77 | 49.72 | 45.68 | 17.46 | 49.96 |

**Number of CCP iterations.** Tab. 8 reports the bargaining budget ablation. In the zero-shot setting, performance remains within a narrow range: MMLU fluctuates between 44.8–45.2 (peaking at 40 iterations), GSM8K shows a slight increase up to 40 before dropping at 60, and ARC improves marginally with a best at 60. After SmolTalk fine-tuning, the curves flatten further–MMLU is tied at 5 and 40 iterations, while GSM8K and ARC peak at 40 with minimal variation overall. Overall, 20–40 iterations slightly outperform 2–5 in some cases, whereas 60 provides no consistent benefit and can degrade performance. Given the small margins and likely variance, we recommend 2 iterations for efficiency and 20 when marginal gains are desired.

**Roburtness to choices of disagreement point.** In Tab. 4 and Tab. 17 in Appendix G.4, we tried "mean" (standard average merging) as the disagreement point and compared it to 0. Across Linear, Cosine, and Stable-MoE, the deltas are tiny (about 0.04 on any metric), with no consistent winner. This shows the gains come from the bargaining weights, not the fallback choice. We keep 0 as the default because it is conservative, stable, and easy to interpret, and it leaves compute unchanged.

## 6   LIMITATION AND CONCLUSION

In this work, we address expert merging through game theory by proposing NAMEx, a method that integrates Nash Bargaining for equitable collaboration and complex momentum to accelerate convergence with theoretical stability guarantees. Experiments across diverse tasks demonstrate NAMEx's consistent superiority over existing methods, highlighting its adaptability to diverse tasks. While NAMEx could be extended to token-level momentum-based methods, such as (Teo & Nguyen, 2024) and (Puigcerver et al., 2024b), the computational cost of solving the Nash equilibrium per token remains a challenge, leaving this as an avenue for future work.

ACKNOWLEDGMENTS

This research / project is supported by the National Research Foundation Singapore under the AI Singapore Programme (AISG Award No: AISG2-TC-2023-012-SGIL). This research / project is supported by the Ministry of Education, Singapore, under the Academic Research Fund Tier 1 (FY2023) (A-8002040-00-00, A-8002039-00-00). This research / project is also supported by the NUS Presidential Young Professorship Award (A-0009807-01-00) and the NUS Artificial Intelligence Institute–Seed Funding (A-8003062-00-00).

We would like to thank FPT for their generous support in providing discounted GPU access on FPT AI Factory, which enabled the experiments in this work.

**Ethics Statement.** Given the nature of the work, we do not foresee any negative societal and ethical impacts of our work.

**Reproducibility Statement.** Source codes for our experiments are provided in the supplementary materials of the paper. The details of our experimental settings and computational infrastructure are given in Section 4, Section 5, and the Appendix. All datasets that we used in the paper are published, and they are easy to access in the Internet.

**LLM Usage Declaration.** We use large language models (LLMs) for grammar checking and correction.

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

# Supplement to "Expert Merging in Sparse Mixture of Experts with Nash Bargaining"

**Table of Contents**

## A NOTATION

Table 9: Notation

| | |
|---|---|
| $\alpha = [\alpha_1, \alpha_2, ..., \alpha_N]$ | The Nash coefficients for merging experts. |
| $\boldsymbol{x}, \boldsymbol{y}, \boldsymbol{z}, \cdots \in \mathbb{C}^n$ | Vectors |
| $\boldsymbol{X}, \boldsymbol{Y}, \boldsymbol{Z}, \cdots \in \mathbb{C}^{n \times n}$ | Matrices |
| $\boldsymbol{X}^\top$ | The transpose of matrix $\boldsymbol{X}$ |
| $\boldsymbol{I}$ | The identity matrix |
| $\Re(z), \Im(z)$ | The real or imaginary component of $z \in \mathbb{C}$ |
| $i$ | The imaginary unit. $z \in \mathbb{C} \implies z = \Re(z) + i\Im(z)$ |
| $\bar{z}$ | The complex conjugate of $z \in \mathbb{C}$ |
| $|z| := \sqrt{z\bar{z}}$ | The magnitude or modulus of $z \in \mathbb{C}$ |
| $\arg(z)$ | The argument or phase of $z \in \mathbb{C} \implies z = |z|\exp(i\arg(z))$ |
| $\mathbf{E}_m^{(l)} \in \mathbb{R}^d$ | Parameters of the base experts at the $l$-th layer of the network |
| $\mathbf{E}_i^l \in \mathbb{R}^d$ | Parameters of the $i$-th experts at the $l$-th layer of the network |
| $\tau_i^{(l)} = \mathbf{E}_i^{(l)} - \mathbf{E}_m^{(l)}$ | The domain-vector of the $i$-th experts at the $l$-th layer of the network |
| $\Delta\boldsymbol{\mathcal{E}} \in \mathbb{R}^d$ | Aggregation of the domain-vector for updating the base expert. |
| $\mathbf{E}_m^{(0)} \in \mathbb{R}^d$ | The initial base expert parameter at the first layer |
| $\gamma \in \mathbb{R}^+$ | The step size for the base expert propagation |
| $\eta \in \mathbb{R}^+$ | The step size for creating the $\hat{\mathbf{E}}_m^{(l)}$ expert that is responsible for processing input |
| $\beta \in \mathbb{C}$ | The momentum coefficient |
| $\boldsymbol{\mu} \in \mathbb{C}^d$ | The momentum buffer |
| $\lambda \in \mathbb{C}$ | Notation for an arbitrary eigenvalue |

## B RELATED WORK

**Sparse Mixture of Experts.** SMoE scales efficiently by activating only a subset of parameters per token, favoring horizontal over deep expansion (Shazeer et al., 2017; Lepikhin et al., 2021b; Fedus et al., 2022; Tran et al., 2025; Teo & Nguyen, 2025), and improves Transformer efficiency without loss. In parallel, model merging has gained traction for combining open-source models (Ilharco et al., 2022; Matena & Raffel, 2022; Yadav et al., 2023; Ramé et al., 2023; Cai et al., 2023; Lu et al., 2024), with curvature-aware methods like Fisher Information (Matena & Raffel, 2022; Jin et al., 2023) improving quality but at high cost. But, most merging approaches assume shared initialization (Yadav et al., 2023; Ilharco et al., 2022), conflicting with SMoE's independently initialized experts and making merging more difficult.

**Nash Bargaining Game.** Originally introduced by (Nash, 1950; 1953), the Nash bargaining framework has been widely studied (Kalai & Smorodinsky, 1975) and recently applied to multi-task learning (Navon et al., 2022; Shamsian et al., 2023). It has also shown success in diverse domains such as multi-armed bandits (Baek & Farias, 2021), clustering (Rezaee et al., 2021), distributed computing (Penmatsa & Chronopoulos, 2011), and economics (Aumann & Hart, 1992; Muthoo, 1999).

**Momentum in Deep Learning.** Momentum-based optimization has been widely studied, from its origins in classical methods (Polyak, 1964; Nesterov, 1983) to its adaptation for deep learning (Sutskever et al., 2013; Zhang & Mitliagkas, 2017; Nguyen et al., 2022b; Teo & Nguyen, 2024). Beyond optimization, momentum has also been examined from dynamical and architectural perspectives. Continuous-time interpretations such as Heavy Ball Neural ODE (Xia et al., 2021) and Nesterov Neural ODE (Nguyen et al., 2022a) connect accelerated methods to second-order differential equations, while Nguyen et al. (2020) incorporates momentum dynamics directly into recurrent architectures. Scheduled Restart Momentum (Wang et al., 2021) studies restart schemes for accelerated stochastic gradient descent, and recent empirical analyses (Sutskever et al., 2013; Wang et al., 2022) investigate how momentum shapes representation learning and architectural behavior in deep networks. In game-theoretic settings, Gidel et al. (2019) explored negative momentum for games, and Lorraine et al. (2022) introduced complex momentum, extending momentum methods to differentiable games using complex-valued updates.

## C   PROOFS OF THE MAIN RESULTS

**Assumption C.1.** We assume a SMoE architecture of infinite SMoE layers with $\{\mathbf{E}_m^{(l)},\mathbf{E}_1^{(l)},\mathbf{E}_2^{(l)},...,\mathbf{E}_N^{(l)}\}$ being the epxerts parameters at $l$-th layer.

**Assumption C.2.**   The norm of experts parameters is bounded, that is:

$$\|\mathbf{E}_i^{(l)}\|\le B \quad \forall l\in\{1,2,...,\infty\}\quad \forall i\in\{m,1,...,N\} \tag{12}$$

### C.1   LEMMA 3.2 PROOF SKETCH

*Proof.* The derivative of the objective function is

$$\sum_{i=1}^{N}\frac{1}{\Delta\boldsymbol{\mathcal{E}}^{\top}\tau_i}\tau_i.$$

For all $\Delta\boldsymbol{\mathcal{E}}$ such that $\Delta\boldsymbol{\mathcal{E}}^{\top}\tau_i>0$ for all $i$, the utilities increase monotonically with the norm of $\Delta\boldsymbol{\mathcal{E}}$. Hence, by Nash's Pareto optimality axiom, the optimal solution must lie on the boundary of $B_\epsilon$. At the optimal point, the gradient

$$\sum_{i=1}^{N}\frac{1}{\Delta\boldsymbol{\mathcal{E}}^{\top}\tau_i}\tau_i$$

must be in the radial direction, i.e.,

$$\sum_{i=1}^{N}\frac{1}{\Delta\boldsymbol{\mathcal{E}}^{\top}\tau_i}\tau_i\propto\Delta\boldsymbol{\mathcal{E}}.$$

Equivalently, there exists $\lambda>0$ such that

$$\sum_{i=1}^{N}\frac{1}{\Delta\boldsymbol{\mathcal{E}}^{\top}\tau_i}\tau_i=\lambda\Delta\boldsymbol{\mathcal{E}}.$$

Since the gradients $\tau_i$ are linearly independent, we can express $\Delta\boldsymbol{\mathcal{E}}$ as $\Delta\boldsymbol{\mathcal{E}}=\sum_{i=1}^{N}\alpha_i\tau_i$. Substituting this into the alignment condition, we obtain

$$\frac{1}{\Delta\boldsymbol{\mathcal{E}}^{\top}\tau_i}=\lambda\alpha_i \quad \forall i.$$

This implies $\Delta\boldsymbol{\mathcal{E}}^{\top}\tau_i=\frac{1}{\lambda\alpha_i}$. As $\Delta\boldsymbol{\mathcal{E}}^{\top}\tau_i>0$ for a descent direction, we deduce $\lambda>0$. Setting $\lambda=1$ gives the direction of $\Delta\boldsymbol{\mathcal{E}}$. Thus, finding the Nash bargaining solution reduces to finding $\alpha\in\mathbb{R}_+^N$ such that

$$\Delta\boldsymbol{\mathcal{E}}^{\top}\tau_i=\sum_{j=1}^{N}\alpha_j\tau_j^{\top}\tau_i=\frac{1}{\alpha_i} \quad \forall i.$$

This is equivalent to solving $G^{\top}G\alpha=1/\alpha$, where $1/\alpha$ is the element-wise reciprocal. $\qquad\square$

### C.2   CONVERGENCE GUARANTEE FOR NAMEX-MOMENTUM

We first have that:

$$\Delta\boldsymbol{\mathcal{E}}^{(l)}=\sum_{i=1}^{N}\alpha_i*\tau_i^{(l)}=\sum_{i=1}^{N}\alpha_i\mathbf{E}_i^{(l)}-\left(\sum_{i=1}^{N}\alpha_i\right)\mathbf{E}_m^{(l)}. \tag{13}$$

Given the analogy between expert merging and gradient descent, we apply the formulation of momentum into Eqn. 7:

$$\begin{cases} \mathbf{E}_m^{(l+1)} &=\mathbf{E}_m^{(l)}+\gamma\Delta\boldsymbol{\mathcal{E}}^{(l)}+\beta(\mathbf{E}_m^{(l)}-\mathbf{E}_m^{(l-1)}), \\ \hat{\mathbf{E}}_m^{(l+1)} &=\mathbf{E}_m^{(l+1)}+\eta\sum_{i=1}^{N}\mathbf{M}_i\cdot(s_i^{(l+1)}*\tau_i^{(l+1)}). \end{cases} \tag{14}$$

Expanding the parameter updates with the Cartesian components of $\gamma$ and $\beta$ is key for Theorem C.3, which characterizes the convergence rate:

$$\boldsymbol{\mu}^{(l+1)}=\beta\boldsymbol{\mu}^{(l)}+\Delta\boldsymbol{\mathcal{E}}^{(l)}\iff$$

$$\Re(\boldsymbol{\mu}^{(l+1)})=\Re(\beta)\Re(\boldsymbol{\mu}^{(l)})-\Im(\beta)\Im(\boldsymbol{\mu}^{(l)})+\Re(\Delta\boldsymbol{\mathcal{E}}^{(l)})$$

$$=\Re(\beta)\Re(\boldsymbol{\mu}^{(l)})-\Im(\beta)\Im(\boldsymbol{\mu}^{(l)})+\sum_{i=1}^{N}\alpha_i\mathbf{E}_i^{(l)}-\left(\sum_{i=1}^{N}\alpha_i\right)\mathbf{E}_m^{(l)}, \tag{15}$$

$$\Im(\boldsymbol{\mu}^{(l+1)})=\Im(\beta)\Re(\boldsymbol{\mu}^{(l)})+\Re(\beta)\Im(\boldsymbol{\mu}^{(l)}) \tag{16}$$

$$\mathbf{E}_m^{(l+1)} = \mathbf{E}_m^{(l)} + \Re(\gamma\boldsymbol{\mu}^{(l+1)}) \tag{17}$$

$$\mathbf{E}_m^{(l+1)} = \mathbf{E}_m^{(l)} + \gamma\Delta\boldsymbol{\mathcal{E}}^{(l)} + \Re(\gamma\beta)\Re(\boldsymbol{\mu}^{(l)}) - \Im(\gamma\beta)\Im(\boldsymbol{\mu}^{(l)})$$

$$= \mathbf{E}_m^{(l)} + \gamma\sum_{i=1}^{N}\alpha_i\mathbf{E}_i^{(l)} - \gamma\left(\sum_{i=1}^{N}\alpha_i\right)\mathbf{E}_m^{(l)} + \Re(\gamma\beta)\Re(\boldsymbol{\mu}^{(l)}) - \Im(\gamma\beta)\Im(\boldsymbol{\mu}^{(l)}) \tag{18}$$

Setting $\boldsymbol{\alpha} = \sum_{i=1}^{N}\alpha_i$, we have

$$\boldsymbol{R} = \begin{bmatrix} \Re(\beta)\boldsymbol{I} & -\Im(\beta)\boldsymbol{I} & -\boldsymbol{\alpha I} \\ \Im(\beta)\boldsymbol{I} & \Re(\beta)\boldsymbol{I} & 0 \\ \Re(\gamma\beta)\boldsymbol{I} & -\Im(\gamma\beta)\boldsymbol{I} & \boldsymbol{I}-\gamma\boldsymbol{\alpha I} \end{bmatrix} \quad \text{and} \quad \mathbf{q}^{(l)} = \left[\sum_{i=1}^{N}\alpha_i\mathbf{E}_i^{(l)} \quad 0 \quad \gamma\sum_{i=1}^{N}\alpha_i\mathbf{E}_i^{(l)}\right]^{\top} \tag{19}$$

Our parameters evolve with expert-propagation merging via:

$$[\Re(\boldsymbol{\mu}^{(l+1)}), \Im(\boldsymbol{\mu}^{(l+1)}), \mathbf{E}^{(l+1)}]^{\top} = \boldsymbol{R}[\Re(\boldsymbol{\mu}^{(l)}), \Im(\boldsymbol{\mu}^{(l)}), \mathbf{E}^{(l)}]^{\top} + \mathbf{q}^{(l)\top} \tag{20}$$

We can bound convergence rates by looking at the spectral radius of $\boldsymbol{R}$ with Theorem C.3.

**Theorem C.3** (Consequence of Prop. 4.4.1 (Bertsekas, 2008))**.** *If the spectral radius $\rho(\boldsymbol{R}) < 1$, then, for $[\boldsymbol{\mu}, \mathbf{E}_m]$ in a neighborhood of $[\boldsymbol{\mu}^*, \mathbf{E}_m^*]$, the distance of $[\boldsymbol{\mu}^{(l)}, \mathbf{E}^{(l)}]$ to the stationary point $[\boldsymbol{\mu}^*, \mathbf{E}_m^*]$ converges at a linear rate $\mathcal{O}((\rho(\boldsymbol{R})+\epsilon)^l), \forall\epsilon > 0$.*

*Proof.* We have:

$$\begin{pmatrix} \Re(\boldsymbol{\mu}^{(l+1)}) \\ \Im(\boldsymbol{\mu}^{(l+1)}) \\ \mathbf{E}^{(l+1)} \end{pmatrix} = \boldsymbol{R}\begin{pmatrix} \Re(\boldsymbol{\mu}^{(l)}) \\ \Im(\boldsymbol{\mu}^{(l)}) \\ \mathbf{E}^{(l)} \end{pmatrix} + \mathbf{q}^{(l)} \tag{21}$$

By telescoping the recurrence for the $l^{th}$ layer:

$$\begin{pmatrix} \Re(\boldsymbol{\mu}^{(l)}) \\ \Im(\boldsymbol{\mu}^{(l)}) \\ \mathbf{E}^{(l)} \end{pmatrix} = \boldsymbol{R}^l\begin{pmatrix} \Re(\boldsymbol{\mu}^{(0)}) \\ \Im(\boldsymbol{\mu}^{(0)}) \\ \mathbf{E}^{(0)} \end{pmatrix} + \sum_{i=0}^{l-1}\boldsymbol{R}^i\mathbf{q}^{(i)} \tag{22}$$

We can compare $\boldsymbol{\mu}^l$ and $\sum_{i=0}^{l-1}\boldsymbol{R}^i\mathbf{q}^{(i)}$ with the values $\boldsymbol{\mu}^*$ and $\mathbf{q}^*$ they converge to which exists if $\boldsymbol{R}$ is contractive. We do the same with $\mathbf{E}$. Because $\boldsymbol{\mu}^* = \boldsymbol{R}\boldsymbol{\mu}^* + \mathbf{q}^* = \boldsymbol{R}^l\boldsymbol{\mu}^* + \sum_{i=1}^{\infty}\boldsymbol{R}^i\mathbf{q}^{(i)}$:

$$\begin{pmatrix} \Re(\boldsymbol{\mu}^{(l)}) - \Re(\boldsymbol{\mu}^*) \\ \Im(\boldsymbol{\mu}^{(l)}) - \Im(\boldsymbol{\mu}^*) \\ \mathbf{E}^{(l)} - \mathbf{E}^* \end{pmatrix} = \boldsymbol{R}^l\begin{pmatrix} \Re(\boldsymbol{\mu}^{(0)}) - \Re(\boldsymbol{\mu}^*) \\ \Im(\boldsymbol{\mu}^{(0)}) - \Im(\boldsymbol{\mu}^*) \\ \mathbf{E}^{(0)} - \mathbf{E}^* \end{pmatrix} + \sum_{i=0}^{l-1}\boldsymbol{R}^l\mathbf{q}^{(i)} - \sum_{l=1}^{\infty}\boldsymbol{R}^i\mathbf{q}^{(i)} \tag{23}$$

By taking norms:

$$\left\|\begin{pmatrix} \Re(\boldsymbol{\mu}^{(l)}) - \Re(\boldsymbol{\mu}^*) \\ \Im(\boldsymbol{\mu}^{(l)}) - \Im(\boldsymbol{\mu}^*) \\ \mathbf{E}^{(l)} - \mathbf{E}^* \end{pmatrix}\right\| = \left\|\boldsymbol{R}^l\begin{pmatrix} \Re(\boldsymbol{\mu}^{(0)}) - \Re(\boldsymbol{\mu}^*) \\ \Im(\boldsymbol{\mu}^{(0)}) - \Im(\boldsymbol{\mu}^*) \\ \mathbf{E}^{(0)} - \mathbf{E}^* \end{pmatrix} - \sum_{i=l}^{\infty}\boldsymbol{R}^i\mathbf{q}^{(i)}\right\| \tag{24}$$

$$\implies \left\|\begin{pmatrix} \Re(\boldsymbol{\mu}^{(l)}) - \Re(\boldsymbol{\mu}^*) \\ \Im(\boldsymbol{\mu}^{(l)}) - \Im(\boldsymbol{\mu}^*) \\ \mathbf{E}^{(l)} - \mathbf{E}^* \end{pmatrix}\right\| \leq \left\|\boldsymbol{R}^l\right\|\left\|\begin{pmatrix} \Re(\boldsymbol{\mu}^{(0)}) - \Re(\boldsymbol{\mu}^*) \\ \Im(\boldsymbol{\mu}^{(0)}) - \Im(\boldsymbol{\mu}^*) \\ \mathbf{E}^{(0)} - \mathbf{E}^* \end{pmatrix}\right\| + \left\|\sum_{i=l}^{\infty}\boldsymbol{R}^i\mathbf{q}^{(i)}\right\| \tag{25}$$

$$\leq \left\|\boldsymbol{R}^l\right\|\left\|\begin{pmatrix} \Re(\boldsymbol{\mu}^{(0)}) - \Re(\boldsymbol{\mu}^*) \\ \Im(\boldsymbol{\mu}^{(0)}) - \Im(\boldsymbol{\mu}^*) \\ \mathbf{E}^{(0)} - \mathbf{E}^* \end{pmatrix}\right\| + \left\|\sum_{i=l}^{\infty}\boldsymbol{R}^i\right\|B \tag{26}$$

With Lemma 11 from (Foucart, 2012), we have there exists a matrix norm $\forall\epsilon > 0$ such that:

$$\|\boldsymbol{R}^l\| \leq D(\rho(\boldsymbol{R})+\epsilon)^l \tag{27}$$

We also have

$$0 < \left\| \sum_{i=0}^{\infty} \boldsymbol{R}^i \right\| < C \tag{28}$$

if $\boldsymbol{R}$ is contractive. Combining Equation (27) and Equation (28) we have:

$$\left\| \begin{pmatrix} \Re(\boldsymbol{\mu}^{(l)}) - \Re(\boldsymbol{\mu}^*) \\ \Im(\boldsymbol{\mu}^{(l)}) - \Im(\boldsymbol{\mu}^*) \\ \mathbf{E}^{(l)} - \mathbf{E}^* \end{pmatrix} \right\| \leq D(\rho(\boldsymbol{R}) + \epsilon)^l \left\| \begin{pmatrix} \Re(\boldsymbol{\mu}^{(0)}) - \Re(\boldsymbol{\mu}^*) \\ \Im(\boldsymbol{\mu}^{(0)}) - \Im(\boldsymbol{\mu}^*) \\ \mathbf{E}^{(0)} - \mathbf{E}^* \end{pmatrix} \right\| + BC \tag{29}$$

So, we have:

$$\left\| \begin{pmatrix} \Re(\boldsymbol{\mu}^{(l)}) - \Re(\boldsymbol{\mu}^*) \\ \Im(\boldsymbol{\mu}^{(l)}) - \Im(\boldsymbol{\mu}^*) \\ \mathbf{E}^{(l)} - \mathbf{E}^* \end{pmatrix} \right\| = \mathcal{O}((\rho(\boldsymbol{R}) + \epsilon)^l) \tag{30}$$

Thus, we converge linearly with a rate of $\mathcal{O}(\rho(\boldsymbol{R}) + \epsilon)$. $\qquad\square$

## C.3 PROPOSTION 3.5 PROOF SKETCH

**Proposition 3.5**(Convergence rate of NAMEx-Momentum). *There exist $\gamma \in \mathbb{R}^+, \beta \in \mathbb{C}$ so Algorithm 2 converges for Expert-Propagation NAMEx Momentum.*

*Proof.* We want to show that we can select $\gamma > 0$ and $\beta \in \mathbb{C}$ so that $\boldsymbol{R}$ is contractive. That is, the spectral radius of $\boldsymbol{R}$ is less than 1. Recall that,

$$\boldsymbol{R} = \begin{bmatrix} \Re(\beta)\boldsymbol{I} & -\Im(\beta)\boldsymbol{I} & -\boldsymbol{\alpha}\boldsymbol{I} \\ \Im(\beta)\boldsymbol{I} & \Re(\beta)\boldsymbol{I} & 0 \\ \Re(\gamma\beta)\boldsymbol{I} & -\Im(\gamma\beta)\boldsymbol{I} & \boldsymbol{I} - \gamma\boldsymbol{\alpha}\boldsymbol{I} \end{bmatrix} \tag{31}$$

Set $\beta = r + ui$, we have:

$$\det(\boldsymbol{R} - x\boldsymbol{I}) = \det\left( \begin{bmatrix} r-x & -u & -\boldsymbol{\alpha} \\ u & r-x & 0 \\ \gamma r & -u & 1-\gamma\boldsymbol{\alpha}-x \end{bmatrix} \otimes \boldsymbol{I} \right) = \det\left( \begin{bmatrix} r-x & -u & -\boldsymbol{\alpha} \\ u & r-x & 0 \\ \gamma r & -u & 1-\gamma\boldsymbol{\alpha}-x \end{bmatrix} \right)^d \tag{32}$$

$$\begin{vmatrix} r-x & -u & -\boldsymbol{\alpha} \\ u & r-x & 0 \\ \gamma r & -u & 1-\gamma\boldsymbol{\alpha}-x \end{vmatrix} = r^2 + \boldsymbol{\alpha}u^2 - \gamma\boldsymbol{\alpha}u^2 + u^2 - x^3 - \gamma\boldsymbol{\alpha}x^2 + 2rx^2 + x^2 - r^2x + \gamma\boldsymbol{\alpha}rx - 2rx - u^2x \tag{33}$$

$$= -x^3 + (-\gamma\boldsymbol{\alpha} + 2r + 1)x^2 + (-r^2 + \gamma\boldsymbol{\alpha}r - 2r - u^2)x + r^2 + (\boldsymbol{\alpha} - \gamma\boldsymbol{\alpha} + 1)u^2 \tag{34}$$

We can further simplify this by choosing $\gamma = \dfrac{\hat{\gamma}}{\boldsymbol{\alpha}}$ to get the following polynomial,

$$P(x) = -x^3 + (1 + 2r - \hat{\gamma})x^2 + (-r^2 + \hat{\gamma}r - 2r - u^2)x + \left( r^2 + (1 + \boldsymbol{\alpha} - \hat{\gamma})u^2 \right) \tag{35}$$

Using Fujiwara's bound (Fujiwara, 1916) we can determine one condition for $\rho(\boldsymbol{R}) < 1$, that is

$$|x| \leq 2\max\left\{ |1 + 2r - \hat{\gamma}|, \sqrt{|-r^2 + \hat{\gamma}r - 2r - u^2|}, \sqrt[3]{\left| \frac{r^2 + (1 + \boldsymbol{\alpha} - \hat{\gamma})u^2}{2} \right|} \right\} < 1 \tag{36}$$

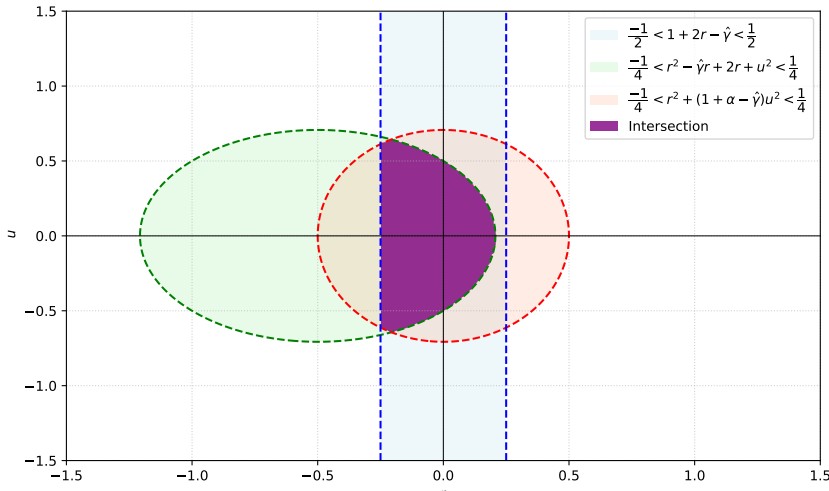

Figure 4: Graph of system of inequality (37) when $\alpha = 0.5$ and $\gamma = 2$.

$$\Rightarrow \begin{cases} \dfrac{-1}{2} < 1 + 2r - \hat{\gamma} < \dfrac{1}{2} \\[2mm] \dfrac{-1}{4} < r^2 - \hat{\gamma}r + 2r + u^2 < \dfrac{1}{4} \\[2mm] -\dfrac{1}{4} < r^2 + (1 + \alpha - \hat{\gamma})u^2 < \dfrac{1}{4} \end{cases} \tag{37}$$

$$\Leftrightarrow \begin{cases} \dfrac{-1}{4} + \dfrac{\hat{\gamma}}{2} - \dfrac{1}{2} < r < \dfrac{1}{4} + \dfrac{\hat{\gamma}}{2} - \dfrac{1}{2} \\[2mm] \dfrac{-1}{4} - r^2 - \hat{\gamma}r - 2r < u^2 < \dfrac{1}{4} - r^2 - \hat{\gamma}r - 2r \\[2mm] \dfrac{-1 - 4r^2}{1 + \alpha - \hat{\gamma}} < u^2 < \dfrac{1 - 4r^2}{1 + \alpha - \hat{\gamma}} \end{cases} \tag{38}$$

We can consider the case when $\alpha = 0.5$ and $\gamma = 2$. Figure 4 shows the region of $(r, u)$ that satisfies inequality system (37).

□

# D  ADDITIONAL DETAILS ON DATASETS

This section outlines the datasets and evaluation metrics employed in the experiments discussed in Section 4.

## D.1  WIKITEXT-103 LANGUAGE MODELING

**The WikiText-103** dataset contains a collection of Wikipedia articles designed to capture long-range contextual dependencies. It includes a training set with 28,475 articles, amounting to around 103 million words. The validation and test sets consist of 217,646 and 245,569 words, respectively, distributed across 60 articles per set.

**Model and baselines** We use the small and medium size Transformer as the baseline SMoE models. Our implementation is based on the codebase developed by (Pham et al., 2024) and (Teo & Nguyen, 2024). All model variants—SMoE, CAMEx, and NAMEx—employ 16 experts per layer, with SMoE utilizing top-1 (k=1) expert selection for each input. The models share a unified sparse routing mechanism, consisting of a linear layer to process the input, followed by Top-K and Softmax functions. Training is performed over 60 epochs for small models and 80 epochs for medium and large SMoE models.

## D.2 GLUE TEXT CLASSIFICATION

The GLUE benchmark includes **SST-2** (Socher et al., 2013) for sentiment analysis, **MRPC** (Dolan & Brockett, 2005) for paraphrase detection and sentence similarity, **CoLA** (Warstadt et al., 2019) for evaluating grammatical acceptability, **STS-B** (Cer et al., 2017) for sentence similarity measurement, **RTE** (Dagan et al., 2006) for logical reasoning, **QNLI** (Wang et al., 2018) for question-answer classification, and **MNLI** (Williams et al., 2018) for assessing entailment between sentence pairs.

**Model and baselines** We scale up T5 (Raffel et al., 2020) using SMoE upcycling (Komatsuzaki et al., 2023). For each task, we conduct a comprehensive hyperparameter search, exploring batch sizes {8, 16, 32, 64} and learning rates {$3e-4, 1e-4, 3e-5, 1e-5$} to identify the optimal fine-tuned configuration.

## D.3 IMAGENET-1K IMAGE CLASSIFICATION

**ImageNet-1k**, introduced by (Deng et al., 2009), is a widely used benchmark dataset comprising 1.28 million images for training and 50,000 images for validation across 1,000 categories. Performance is evaluated using top-1 and top-5 accuracy metrics.

For robustness evaluation, we utilize several specialized subsets. **ImageNet-A** (Hendrycks et al., 2021c) focuses on 200 challenging classes from ImageNet-1k, specifically curated to fool classifiers, highlighting their vulnerability to real-world adversarial examples. **ImageNet-O** (Hendrycks et al., 2021c) contains out-of-distribution (OOD) samples derived from ImageNet-22k, carefully selected as instances that a ResNet-50 model misclassifies with high confidence. The primary evaluation metric for ImageNet-O is the area under the precision-recall curve (AUPR). Lastly, **ImageNet-R** (Hendrycks et al., 2021a) consists of 30,000 artistic renditions representing 200 classes from ImageNet-1k, designed to assess model robustness to non-standard visual representations.

**Model and baselines** For each MoE layer, we use Algorithm 2 to merge all experts into a base expert, except in the first MoE layer, where a base expert is initialized instead. Training configurations follow Swin-MoE (Liu et al., 2021), and the code is publicly available on https://github.com/microsoft/Swin-Transformer/. For NAMEx variants, we start with checkpoints pretrained on ImageNet-22k and fine-tune them on ImageNet-1k for 30 epochs.

# E MORE EXPERIMENTAL DETAILS

This section provides additional details on the experimental setup, including model configurations, dataset processing, and training strategies used in our evaluation.

## E.1 WIKITEXT-103 LANGUAGE MODELING

We follow the setup from (Pham et al., 2024) and (Teo & Nguyen, 2024), using both small and medium-scale Transformer architectures with 16 experts per layer. All variants (SMoE, CAMEx, EP-CAMEx, NAMEx) use Top-1 routing and share the same sparse gating mechanism. Training is conducted for 60 epochs (small) and 80 epochs (medium) with AdamW optimizer and cosine learning rate scheduling.

## E.2 GLUE BENCHMARK FINE-TUNING

For text classification, we fine-tune T5-base models upcycled with SMoE layers. We conduct grid searches over batch sizes 8, 16, 32, 64 and learning rates $3\times10^{-5}, 1\times10^{-4}, 3\times10^{-4}$. Each result is averaged over five seeds to ensure statistical stability. All SMoE variants employ 8 experts per layer and share the same routing logic.

## E.3 IMAGENET-1K AND CORRUPTED VARIANTS

For vision experiments, we fine-tune Swin-MoE-Small on ImageNet-1k for 30 epochs using batch size 96 and learning rate $1\times10^{-4}$. NAMEx variants initialize $\mathbf{E}_m$ in the first MoE layer and perform merging across all others via Algorithm 1 or Algorithm 2. For robustness, we evaluate zero-shot generalization on ImageNet-A, ImageNet-O, and ImageNet-R. All reported results are averaged over three runs.

## E.4 IMPLEMENTATION AND INFRASTRUCTURE

Experiments are implemented in PyTorch and trained on 4–8 A100 GPUs depending on model size. We use automatic mixed precision (AMP) for memory efficiency. All hyperparameters, data augmentations, and merging schedules are described in Appendix D.

# F ADDITIONAL EXPERIMENTAL RESULTS

## F.1 ZERO-SHOT AND FINETUNING ON QWEN1.5-MOE (14B PARAMETERS)

To assess scalability, we integrate NAMEx-Full into Qwen1.5-MoE (14B) and evaluate it on three challenging benchmarks: MMLU (Hendrycks et al., 2021b), GSM8K (Cobbe et al., 2021), and ARC (Clark et al., 2018) in both zero-shot and fine-tuned settings. In the fine-tuned setting, all models are fine-tuned on the SmolTalk (allal et al., 2025) dataset before evaluation. As reported in Tables 10 and 11, NAMEx-Full consistently outperforms both the baseline and EP-CAMEx across routing schemes and tasks, highlighting its robustness, scalability, and architectural generality.

Table 10: Zero-shot results for Qwen1.5-MoE variants.

| Routing Strategy | Model | MMLU | GSM8K | ARC |
|---|---|---|---|---|
| Linear | Qwen1.5-MoE | 61.28 | 60.12 | 50.77 |
| | EP-CAMEx | 61.54 | 60.23 | 50.83 |
| | **NAMEx-Full** | **61.87** | **60.55** | **50.95** |
| Cosine | Qwen1.5-MoE | 61.10 | 59.88 | 50.60 |
| | EP-CAMEx | 61.40 | 60.00 | 50.68 |
| | **NAMEx-Full** | **61.85** | **60.52** | **50.93** |
| Stable-MoE | Qwen1.5-MoE | 61.35 | 60.22 | 50.81 |
| | EP-CAMEx | 61.60 | 60.35 | 50.89 |
| | **NAMEx-Full** | **61.90** | **60.60** | **50.96** |

Table 11: Results after fine-tuning on SmolTalk.

| Routing Strategy | Model | MMLU | GSM8K | ARC |
|---|---|---|---|---|
| | Qwen1.5-MoE | 61.50 | 60.52 | 51.12 |
| Linear | EP-CAMEx | 61.74 | 60.63 | 51.23 |
| | **NAMEx-Full** | **62.10** | **61.00** | **51.35** |
| | Qwen1.5-MoE | 61.30 | 60.28 | 50.95 |
| Cosine | EP-CAMEx | 61.60 | 60.50 | 51.10 |
| | **NAMEx-Full** | **62.05** | **60.95** | **51.30** |
| | Qwen1.5-MoE | 61.60 | 60.65 | 51.20 |
| Stable-MoE Routing | EP-CAMEx | 61.85 | 60.80 | 51.30 |
| | **NAMEx-Full** | **62.15** | **61.10** | **51.45** |

# G ADDITIONAL EMPIRICAL ANALYSIS

## G.1 OVERHEAD AND SCALABILITY

Table 12: Impact of NBS update frequency ($\Delta\ell$). Per-batch wall-clock.

| $\Delta\ell$ | SST-2 | MRPC | STS-B | RTE | Mean Batch (s) | % NBS Time |
|---|---|---|---|---|---|---|
| 1 | 94.88 | 92.85 | 90.32 | 77.26 | 4.70 | 85.96% |
| 2 | 94.95 | 92.38 | 90.37 | 76.89 | 2.29 | 71.18% |
| 5 | 95.18 | 92.09 | 90.13 | 77.98 | 1.14 | 42.11% |
| $L$ (first layer) | 94.46 | 92.01 | 90.12 | 75.09 | **0.69** | **4.35%** |

Table 13: Training compute and throughput. Inference is unchanged relative to baselines.

| Model | Train TFLOPs | Infer TFLOPs | Train Throughput (tok/s) |
|---|---|---|---|
| SMoE | 13.95 | 4.65 | 22,236 |
| **SMoE (Top-2)** | 18.32 | 7.44 | 17,898 |
| SMEAR | 13.95 | 4.65 | 22,236 |
| CAMEx | 14.30 | 4.65 | 21,982 |
| EP-CAMEx | 14.25 | 4.65 | 21,982 |
| NAMEx | 14.25 | 4.65 | 21,995 |
| **NAMEx-Full** | **14.25** | **4.65** | **21,897** |
| EP-CAMEx-Mom | 14.25 | 4.65 | 21,872 |
| **NAMEx-Full-Mom** | **14.25** | **4.65** | **21,783** |

In Tab. 12, we provide a detailed runtime cost analysis, "Mean Batch Runtime (sec)" includes the entire forward/backward pass and NBS step. "% NBS Occupied" isolates the share of time spent solving NBS.

In Tab. 13, we provide a comparison of training TFLOPs, inference TFLOPs, and training through-put across baselines and our proposed variants in Table 2 below. Notably, NAMEx-Full achieves competitive throughput (21,897 tokens/sec), closely matching CAMEx and EP-CAMEx despite the added complexity of solving the Nash system. While NAMEx and NAMEx-Mom show slightly lower throughput due to the large number of NBS iterations (20 iters), NAMEx-Full-Mom restores much of the efficiency by reducing the number of NBS iterations (2 iters per layer). Currently, the main overhead arises from transferring data to the CPU for solving the Nash Bargaining update step. However, this implementation detail is orthogonal to the algorithm itself and can be optimized via GPU-native solvers or batching strategies. Overall, the results demonstrate that NAMEx introduces minimal overhead and remains practical for large-scale MoE training, supporting its applicability to future deployments involving more experts per layer.

## G.2 CONVERGENCE ANALYSIS

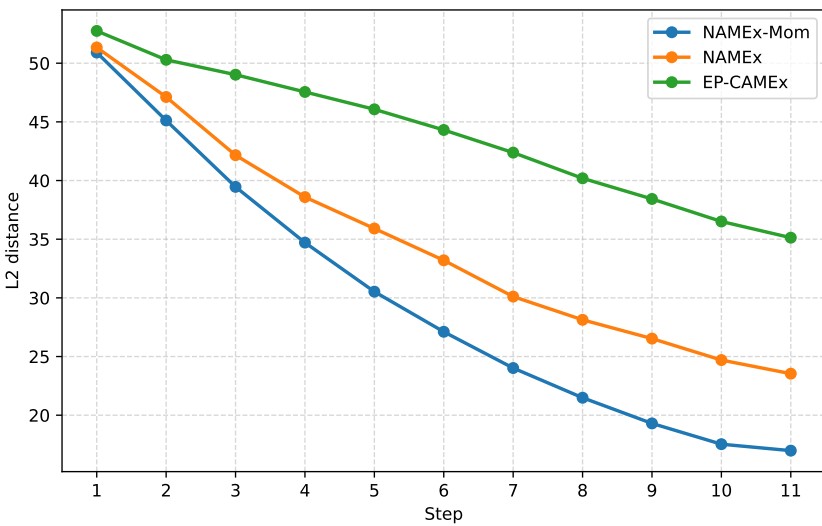

Figure 5: L2 distance between expert updates across training steps (T5-Base, 12 MoE layers). Lower values indicate better stability. The figure shows that NAMEx converges faster and more stably than EP-CAMEx.

To validate the motivation for complex momentum, we provide empirical convergence analysis in Fig. 5, which tracks the L2 distance between updates of base experts across training steps (T5-Base, 12 MoE layers). As illustrated, NAMEx–with or without momentum–shows a noticeably steeper decline in update distances, indicating faster convergence and more stable expert updates compared to EP-CAMEx. This directly supports our hypothesis that complex momentum enhances convergence stability and efficiency during expert merging.

## G.3 SWIN-MOE-S: 90-EPOCH FINE-TUNING

Table 14: ImageNet-1K Top-1 from epochs 86–90. Our best final performance is **85.046%**.

| Epoch | NAMEx-Mom | NAMEx | EP-CAMEx-Mom | EP-CAMEx | NAMEx-Full | NAMEx-Full-Mom | Swin-MoE |
|---|---|---|---|---|---|---|---|
| 86 | 84.466 | 84.264 | 83.862 | 82.238 | 84.722 | **85.022** | 83.435 |
| 87 | 84.504 | 84.252 | 83.868 | 82.286 | 84.728 | **85.028** | 83.400 |
| 88 | 84.502 | 84.240 | 83.854 | 82.234 | 84.734 | **85.034** | 83.379 |
| 89 | 84.540 | 84.228 | 83.860 | 82.282 | 84.740 | **85.040** | 83.413 |
| 90 | 84.518 | 84.216 | 83.806 | 82.230 | 84.746 | **85.046** | 83.415 |

Tab. 14 summarizes the top-1 accuracy on ImageNet-1K from epochs 86 to 90, these results confirm that our models continue to improve with more training, and that Namex-full-Mom reaches a final top-1 accuracy of 85.046%. This demonstrates both competitive final performance and strong convergence behavior. Note that due to the difference in number of GPU being used, it seems that we could not reproduce the 84.5% result as reported by the official repo of Swin-MoE.

## G.4 OTHER RESULTS

Figure 7 presents the evaluation perplexity on WikiText-103 during training. NAMEx-Mom achieves the lowest validation and test perplexities in both small- and medium-scale pre-training, outperforming

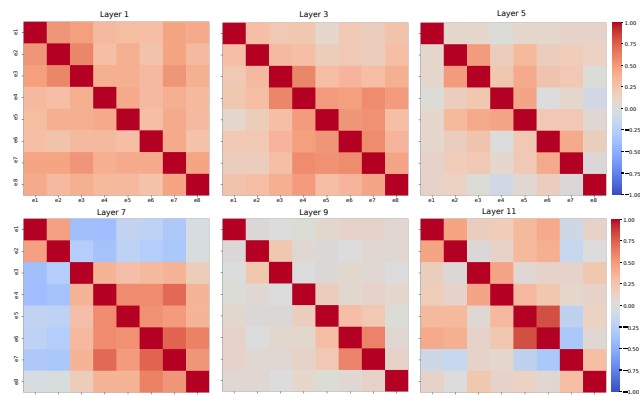

Figure 6: Cosine similarity between expert outputs in Switch-Transformers.

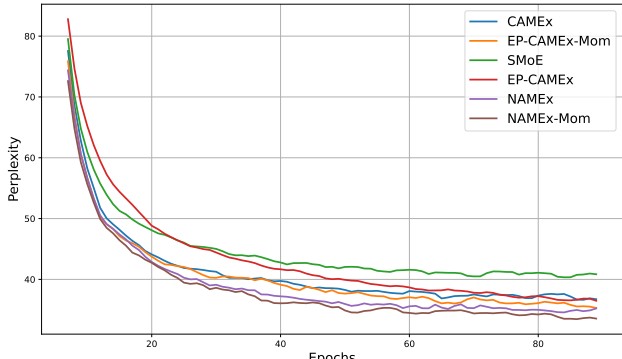

Figure 7: 5-Period Moving Average of Perplexity of different Transformers-medium variants on WikiText-103

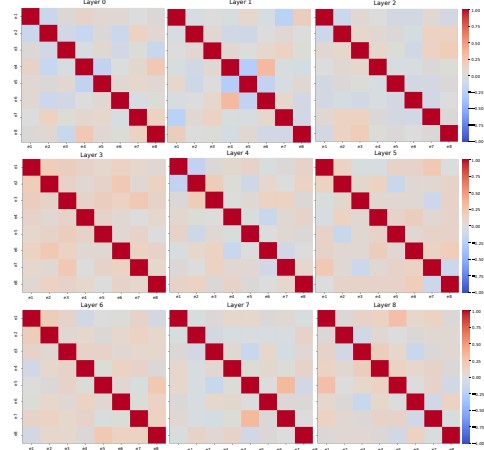

Figure 8: Cosine similarity between expert outputs in Swin-MoE.

SMoE and CAMEx-based methods. Across all settings, Nash variants, including NAMEx and NAMEx-Mom, consistently surpass their counterparts, demonstrating the effectiveness of Nash bargaining and momentum mechanisms.

Figure 6 and Figure 8 visualize the cosine similarity between experts ouput at all SMoE layers indicating a complex dynamic of how experts at progressive layers interact with each other. *This observation suggests that the behavior of experts cannot be captured optimally by using simple averaging as of the previous work.* Instead, a more effective strategy for determining the merging coefficients should account for the experts' dynamics at each specific layer.

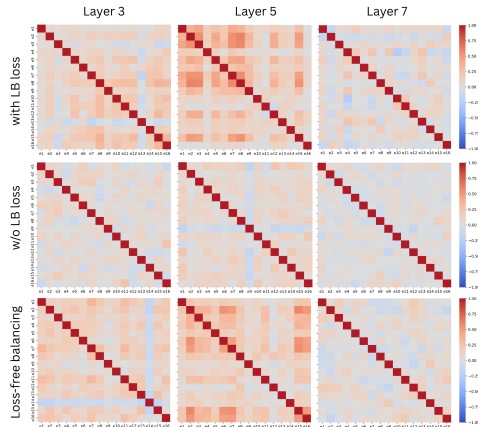

Figure 9: We compared expert interaction patterns under three settings: with Load Balancing loss, without Load Balancing loss, and loss-free balancing (in the sense of (Wang et al., 2025)).

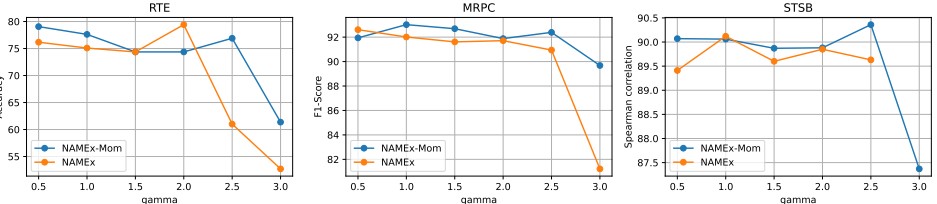

Figure 10: Impact of different values of step-size $\gamma$ on NAMEx and NAMEx-Mom performance. Overall, the optimal setting lies within the range $[0.5, 2]$. For $\gamma > 2$, the performance drops significantly, which may indicate an overshooting phenomenon.

Table 15: Pretraining and zero-shot results for NAMEx vs. ACMoE Top-1/Top-2 on ImageNet-1k and corrupted variants.

| Model | Params | Acc@1 | Acc@5 | INet-O | INet-A | INet-R |
|---|---|---|---|---|---|---|
| ACMoE-Top 1 | 280M | 75.39 | 92.56 | 18.45 | 7.13 | 30.85 |
| ACMoE-Top 2 | 280M | 76.31 | 93.14 | 19.55 | 9.42 | 32.35 |
| NAMEx | 280M | 76.85 | 93.40 | 20.11 | 9.90 | 32.93 |
| NAMEx-Full | 280M | 77.42 | 93.85 | 20.69 | 10.46 | 33.44 |
| NAMEx-Full-Mom | 280M | **78.15** | **94.23** | **21.16** | **11.02** | **33.95** |

As shown in Figure 9, the cooperative/competitive dynamics (as reflected in the off-diagonal correlation values) are much more visible when expert load is balanced–either through Load Balancing loss or loss-free mechanisms. In contrast, when training without Load Balancing loss, many experts appear less specialized, and the interaction patterns become weaker and less structured.

One hypothesis is that, without balanced token routing, some experts may be underused or even become inactive, which diminishes the emergence of meaningful cooperative or competitive behavior. Therefore, balanced expert load is not only important for preventing dead experts but also plays a crucial role in making such dynamics observable and analyzable.

In Table 15, across all ImageNet variants, the NAMEx-based models consistently outperform the ACMoE Top-1 and Top-2 baselines. In particular, NAMEx-Full and NAMEx-Full-Mom set new best accuracies on both in-distribution metrics (Acc@1 and Acc@5) and out-of-distribution benchmarks (INet-O, INet-A, INet-R). This underlines the strong generalization ability of NAMEx. Even with the same parameter budget, NAMEx variants deliver better robustness to corruptions and distribution shifts.

## H    BROADER IMPACTS

NAMEx proposes a principled, game-theoretic approach to expert merging in Sparse Mixture-of-Experts (SMoE) models, addressing key limitations of heuristic and curvature-based methods. By

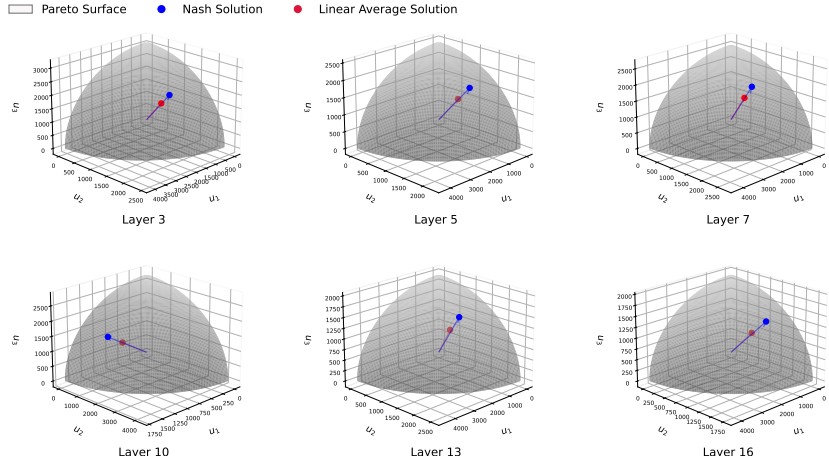

Figure 11: Visualization of expert utility trade-offs accross multiple SMoE layers. Each subplot corresponds to a different layer, with arrows indicating merging directions in the 3D utility space. The blue arrow represents the Nash Bargaining solution (NAMEx), while the red arrow denotes the average-based merging direction. The hemispherical surface is uniformly sampled to illustrate the feasible utility region. Across layers, the Nash direction consistently steers toward more balanced expert cooperation compared to the naive average.

Table 16: Performance comparison across number of NBS solving iterations for the Linear router in NAMEx-Full config (Qwen1.5-MoE). All results to be filled; we keep the same marginal-gap setup. Throughout experiments, use 2 CCP iterations per layer for NAMEx-Full (chosen config below).

| Model | No. Iterations | Zero-Shot | | | Fine-tuned (SmolTalk) | | |
|---|---|---|---|---|---|---|---|
| | | MMLU | GSM8K | ARC | MMLU | GSM8K | ARC |
| Qwen1.5-MoE | 2 (Chosen Config) | 61.87 | 60.55 | 50.95 | 62.10 | 61.00 | 51.35 |
| | 5 | 61.70 | 60.55 | 50.94 | 62.20 | 61.05 | 51.31 |
| | 20 | 61.94 | 60.57 | 50.96 | 62.14 | 61.03 | 51.34 |
| | 40 | 61.92 | 60.62 | 51.01 | 62.22 | 61.05 | 51.46 |
| | 60 | 61.81 | 60.48 | 51.08 | 62.15 | 60.98 | 51.39 |

Table 17: Performance comparison across routing strategies and models on MMLU, GSM8K, and ARC benchmarks for Qwen1.5 - MoE variants. Left: zero-shot results. Right: fine-tuned NAMEx - Full variants on SmolTalk.

| Routing Strategy | Model | Zero-Shot | | | Fine-tuned (SmolTalk) | | |
|---|---|---|---|---|---|---|---|
| | | MMLU | GSM8K | ARC | MMLU | GSM8K | ARC |
| Linear | Qwen1.5 - MoE | 61.28 | 60.12 | 50.77 | 61.50 | 60.52 | 51.12 |
| | EP - CAMEx | 61.54 | 60.23 | 50.83 | 61.74 | 60.63 | 51.23 |
| | **NAMEx - Full** (0 disagreement point) | **61.87** | **60.55** | **50.95** | **62.10** | **61.00** | **51.35** |
| | **NAMEx - Full** (mean disagreement point) | **61.78** | **60.57** | **51.23** | **61.67** | **61.04** | **51.25** |
| Cosine | Qwen1.5 - MoE | 61.10 | 59.88 | 50.60 | 61.30 | 60.28 | 50.95 |
| | EP - CAMEx | 61.40 | 60.00 | 50.68 | 61.60 | 60.50 | 51.10 |
| | **NAMEx - Full** (0 disagreement point) | **61.85** | **60.52** | **50.93** | **62.05** | **60.95** | **51.30** |
| | **NAMEx - Full** (mean disagreement point) | **61.86** | **60.45** | **50.77** | **62.01** | **60.81** | **51.37** |
| Stable-MoE | Qwen1.5 - MoE | 61.35 | 60.22 | 50.81 | 61.60 | 60.65 | 51.20 |
| | EP - CAMEx | 61.60 | 60.35 | 50.89 | 61.85 | 60.80 | 51.30 |
| | **NAMEx - Full** (0 disagreement point) | **61.90** | **60.60** | **50.96** | **62.15** | **61.10** | **51.45** |
| | **NAMEx - Full** (mean disagreement point) | **61.88** | **60.64** | **51.03** | **62.15** | **61.11** | **51.35** |

leveraging Nash Bargaining, it enables more balanced and interpretable parameter integration, particularly in settings with conflicting or specialized expert knowledge. This has direct implications for scalable deployment, as NAMEx can reduce the memory and compute footprint of large SMoE models while preserving performance. The addition of complex momentum enhances convergence stability during expert propagation, offering a robust framework for layered expert interaction. These contributions may prove valuable for future research in modular deep learning, federated optimization, and transfer learning, where efficient and fair expert combination is critical.

