# OpenReview forum: "Expert Merging in Sparse Mixture of Experts with Nash Bargaining"
_ICLR.cc/2026/Conference — ICLR 2026 Poster_

### Official Review · Reviewer_kguY · 2025-10-25

**Soundness:** 3
**Presentation:** 3
**Contribution:** 3
**Rating:** 6
**Confidence:** 2

**Summary:**

This manuscript introduces Nash Merging of Experts (NAMEx), a novel, theoretically-grounded framework for compressing Sparse Mixture of Experts (SMoE) models. The core innovation is the reinterpretation of the merging process as a cooperative game, utilizing Nash Bargaining to determine the optimal, balanced parameter weighting for merging two or more experts. The authors claim this approach, coupled with a mechanism for enhanced convergence (complex momentum), leads to significant reduction in the SMoE's memory and compute footprint while preserving performance. The work is highly relevant to the scalable deployment of large sparse models.

**Strengths:**

* The most significant strength is the introduction of a game-theoretic perspective (Nash Bargaining) to expert merging.
* The work directly addresses the scalability and deployment challenge of large SMoE models by enabling significant model compression. The results demonstrating reduced footprint while maintaining performance are very compelling.

**Weaknesses:**

* While the memory/compute reduction is a strong result, the paper must clarify the computational overhead of the NAMEx merging process itself. Since Nash Bargaining requires solving an optimization problem to determine the merge point, how does the time taken for merging compare against simpler methods like magnitude pruning or basic parameter averaging? A clear complexity analysis or runtime comparison is needed.
* The core of Nash Bargaining relies on the definition of the experts' utility functions and the disagreement point (status quo). The manuscript needs to dedicate more detail to explaining why the chosen utility function accurately captures an expert's "value" or "specialized knowledge," and how it relates to the loss landscape or data distribution.

**Questions:**

* Could the authors explicitly detail the mathematical expression of the utility function, $u_i(\theta)$, for a single expert $i$ within the merging context? How is this function derived from the expert's parameters or its performance on a held-out dataset?
* Please provide a clear, detailed equation for the "complex momentum" and explain how it differs from standard momentum methods (like Nesterov or Adam momentum). Does it specifically counteract issues arising from the non-convexity introduced by the merging objective?

---

> ### Author Response · Authors · 2025-11-20
> **Response to Reviewer kguY (1)**
>
> Thank you for your thoughtful review and valuable feedback. Below, we address your concerns.
>
> -----
>
> **[Weakness 1]. While the memory/compute reduction is a strong result, the paper must clarify the computational overhead of the NAMEx merging process itself. Since Nash Bargaining requires solving an optimization problem to determine the merge point, how does the time taken for merging compare against simpler methods like magnitude pruning or basic parameter averaging? A clear complexity analysis or runtime comparison is needed.**
>
> **Answer:** During inference-time, NAMEx does not require solving the NBS, so the computational overhead is the same as the simple average merging baseline (e.g, SMEAR).
> However, during training, NAMEx requires obtaining NBS by solving a problem with a concave objective and convex constraints. Hence, we solve them using CCP with ECOS solver [5] While we cannot provide a direct complexity for this solver, we can derive a upper bound from its competitor, the SDPT3 solver, as it is well known that ECOS  runtime would be approximately equal to that of SDPT3 for really large problems, while being extremely faster when the number of variables is small [5]. For an iteration of SDPT3, the commonly reported time complexity is around $O(N^3)$, with $N$ being the number of variables, and the number of iterations required is commonly quite small as well.
>
> In our case, $N$ is the number of experts per layer, and the total number of iterations is around 20 for all reported models, so this should be negligible compared to computing the constraint matrix, which requires $O(N^2 d)$, with $d$ being the total number of parameters in an expert.
>
>
> Furthermore, Table 13 of Appendix F.1, or Table 1 below, shows that, in terms of token throughput, NAMEx and NAMEx-Full are on par with SMoE (21,987 tokens/second and 22,236 tokens/second). In theory, the training time complexity for simple average merging should be equal to that of SMoE since the summation of weights is $O(1)$.
>
> In Table 1 below, we compare with top-1 routing because NAMEx, CAMEx, and EP-CAMEx all operate with a single active expert (the merged expert). However, top-1 is not the standard setting in modern MoE systems. In practice, models use top-2 or top-K routing (K = 8, 16, …) as in GShard [3] and DeepSeek-MoE [4].
>
> We additionally compare against the top-2 SMoE in Tables 1, 2, and 3 below. We also included these results in Table 1–2 in Section 4 and Table 13 in Appendix F.1 of the revised manuscript.
>
> Across GLUE tasks and WikiText-103 perplexity, NAMEx and NAMEx-Full consistently outperform CAMEx and EP-CAMEx, with NAMEx-Full-Mom achieving the best performance overall.
>
> Importantly:
>
> - **SMoE (top-2)** provides small accuracy gains but with **substantially higher compute**
> (train throughput drops **19.5%**, from 22,236 $\rightarrow$ 17,898 tok/s).
> - **NAMEx and NAMEx-Full** improve accuracy purely through expert-space merging, **without increasing routing cost or activating more experts per token**. NAMEx throughput decreases by only around **1.1%** relative to SMoE (top-1), and is **22.9%** faster than SMoE (top-2).
>
> **Table 1**. Training compute and throughput. Inference is unchanged relative to baselines.
> | Model                     | Train TFLOPs | Infer TFLOPs | Train Throughput (tok/s) |
> |---------------------------|--------------|---------------|----------------------------|
> | SMoE                      | 13.95        | 4.65          | 22,236                     |
> | **SMoE (Top-2)**        | 18.32        | 7.44          | 17,898                     |
> | SMEAR                      | 13.95        | 4.65          | 22,236                     |
> | CAMEx                     | 14.30        | 4.65          | 21,982                     |
> | EP-CAMEx                  | 14.25        | 4.65          | 21,982                     |
> | NAMEx                     | 14.25        | 4.65          | 21,995                     |
> | **NAMEx-Full**            | **14.25**    | **4.65**      | **21,897**                 |
> | EP-CAMEx-Mom              | 14.25        | 4.65          | 21,872                     |
> | **NAMEx-Full-Mom**        | **14.25**    | **4.65**      | **21,783**|

---

> > ### Author Response · Authors · 2025-11-20
> > **Response to Reviewer kguY (2)**
> >
> > **Table 2.** Validation and test perplexity on WikiText-103 for small- and medium-scale pretraining.
> >
> > | Model              | Params       | Small Val PPL                               | Small Test PPL                              | Medium Val PPL                              | Medium Test PPL                             |
> > |--------------------|--------------|----------------------------------------------|----------------------------------------------|----------------------------------------------|----------------------------------------------|
> > | SMoE (Top-1)     | 70M / 216M   | $86.64 \scriptscriptstyle{\pm .22}$          | $87.79 \scriptscriptstyle{\pm .31}$          | $38.60 \scriptscriptstyle{\pm .18}$          | $40.51 \scriptscriptstyle{\pm .25}$          |
> > | SMoE (Top-2)       | 70M / 216M   | $84.26 \scriptscriptstyle{\pm .12}$          | $84.81 \scriptscriptstyle{\pm .29}$          | $\mathbf{33.76 \scriptscriptstyle{\pm .19}}$          | $\underline{35.55 \scriptscriptstyle{\pm .22}}$          |
> > | NAMEx              | 70M / 216M   | $83.30 \scriptscriptstyle{\pm .21}$          | $84.12 \scriptscriptstyle{\pm .29}$          | $35.14 \scriptscriptstyle{\pm .19}$          | $36.40 \scriptscriptstyle{\pm .27}$          |
> > | NAMEx-Full         | 70M / 216M   | $82.85 \scriptscriptstyle{\pm .17}$          | $\underline{83.16 \scriptscriptstyle{\pm .23}}$          | $34.92 \scriptscriptstyle{\pm .14}$          | $36.21 \scriptscriptstyle{\pm .20}$          |
> > | NAMEx-Mom          | 70M / 216M   | $\underline{82.63 \scriptscriptstyle{\pm .15}}$ | $83.59 \scriptscriptstyle{\pm .21}$ | ${34.89 \scriptscriptstyle{\pm .12}}$ | ${35.86 \scriptscriptstyle{\pm .18}}$ |
> > | NAMEx-Full-Mom     | 70M / 216M   | $\mathbf{82.44 \scriptscriptstyle{\pm .14}}$ | $\mathbf{82.94 \scriptscriptstyle{\pm .20}}$ | $\underline{34.25 \scriptscriptstyle{\pm .11}}$ | $\mathbf{35.37 \scriptscriptstyle{\pm .17}}$ |
> >
> > **Table 3.** GLUE results for T5-base SMoE variants (8 experts/layer).
> >
> > | Model               | Params | SST-2 | MRPC | CoLA | STSB | RTE | QNLI | MNLI |
> > |---------------------|--------|-------|-------|-------|-------|-------|--------|--------|
> > | SMoE (Top-1)        | 1.0B  | 94.26 | 90.87 | 56.78 | 89.44 | 70.75 | 92.07 | 86.38 |
> > | **SMoE (Top-2)**    | 1.0B  | 94.35 | 91.04 | 58.43 | 89.73 | 74.98 | 92.48 | 86.72 |
> > | NAMEx         | 1.0B | 94.46 | 92.01 | 58.81 | 90.12 | 75.09 | 92.86 | 86.96 |
> > | **NAMEx-Full**| 1.0B | 94.82 | 92.80 | 59.63 | 90.27 | 77.83 | 93.23 | 87.23 |
> > | NAMEx-Mom           | 1.0B | 94.61 | 93.02 | 58.90 | 90.06 | 77.62 | 93.11 | 87.02 |
> > | **NAMEx-Full-Mom**  | 1.0B | **95.06** | **93.27** | **60.13** | **90.63** | **78.15** | **93.31** | **87.45** |

---

> ### Author Response · Authors · 2025-11-20
> **Response to Reviewer kguY (3)**
>
> **[Weakness 2].The core of Nash Bargaining relies on the definition of the experts' utility functions and the disagreement point (status quo). The manuscript needs to dedicate more detail to explaining why the chosen utility function accurately captures an expert's "value" or "specialized knowledge," and how it relates to the loss landscape or data distribution.**
>
> **Answer:** Thanks for such an insightful recommendation. In NAMEx, the utility of each expert is defined as $u_i(\Delta E) = \tau_i^\top \Delta E$, where $\tau_i = E_i - E_m$ is the domain vector of expert $i$. This formulation follows the Nash-MTL[2] framework and is motivated by a geometric interpretation of SMoE. Each expert represents a task-specific direction in parameter space, and $\tau_i$ encodes the update that expert $i$ would prefer if it were optimized independently. Because experts are trained on different regions of the data distribution, these domain vectors implicitly contain gradient and curvature information associated with the local loss landscape. The utility function, therefore, measures how well a proposed merged update $\Delta E$ aligns with the direction that preserves the specialized knowledge of expert $i$. This is consistent with the curvature-based interpretation in [1], where expert-specific geometry guides the update direction. Setting the disagreement point to zero (i.e., no update) aligns with standard practice in multi-task optimization and reflects the fallback outcome when an expert receives no improvement. Under this formulation, the Nash solution produces a principled weighting that is Pareto efficient, resolves conflicts through the inner products $\tau_i^\top \tau_j$, and selects a merged update direction that benefits all experts simultaneously.

---

> ### Author Response · Authors · 2025-11-20
> **Response to Reviewer kguY (4)**
>
> **[Question 1]. Could the authors explicitly detail the mathematical expression of the utility function, $u_i(\theta)$, for a single expert $i$ within the merging context? How is this function derived from the expert's parameters or its performance on a held-out dataset?**
>
> **Answer:** The utility function $u_i(\Delta \theta)$ evaluates the update direction $\Delta \theta$, rather than the parameters themselves. Given a domain vector $\tau_i$ extracted from domain expert $E_i$, the purpose of the utility function is to score an update direction $\Delta\theta$ within the unit ball $B_\epsilon$ based on how well it aligns with the domain vector. This is measured through the dot product between $\tau_i$ and $\Delta\theta$. By not normalizing the domain vectors, the utilities also reflect the degree of cooperation or conflict among experts at a given layer. Its mathematical expression can be seen in the below equation:
> $$
> u_i(\Delta\theta) = \tau_i^{\intercal}\Delta\theta
> $$
> We refer to equation 5 and the corresponding section in our manuscript for a more complete context of the utility function, as well as how they are used to calculate the final update direction for an expert merging problem.
>
> The NBS provides a direct solution that maximizes all utilities simultaneously, producing an affine combination of all domain vectors (or zero when disagreement cannot be resolved). When a new domain expert $E_{\mathrm{new}}$ is added, we simply define a corresponding utility function $u_{\mathrm{new}}(\Delta\theta)$ and recompute the shared update direction using all utilities. The resulting NBS is guaranteed to be the optimal update direction with respect to all available utility functions.
>
> We would greatly appreciate it if you could provide more details about your concern regarding performance on a held-out dataset. We would be very happy to address it.

---

> ### Author Response · Authors · 2025-11-20
> **Response to Reviewer kguY (5)**
>
> **[Question 2]. Please provide a clear, detailed equation for the "complex momentum" and explain how it differs from standard momentum methods (like Nesterov or Adam momentum). Does it specifically counteract issues arising from the non-convexity introduced by the merging objective?**
>
> **Answer:** We begin by presenting the update rules for standard momentum and Nesterov momentum$-$both of which use a single momentum buffer$-$and then contrast them with Adam momentum and the complex momentum formulation used in our expert-merging framework.
>
> For a basic gradient-descent update of the form
> $$
> \theta_{t+1} = \theta_t - \gamma v_t.
> $$
> The momentum variants are:
>
> - **Standard momentum:**
>     $$
>     v_t = \beta v_{t-1} + \nabla L(\theta_t).
>     $$
> - **Nesterov momentum:**
>     $$
>     v_t = \beta v_{t-1} + \nabla L(\theta_t - \gamma v_{t-1}).
>     $$
> - **Adam momentum:**
>     $$
>     m_t = \beta_1 m_{t-1} + (1 - \beta_1)\nabla L(\theta_t), \qquad
>     h_t = \beta_2 h_{t-1} + (1 - \beta_2)\nabla L(\theta_t)^2,
>     $$
>     $$
>     v_t = \frac{m_t}{\sqrt{h_t} + \epsilon}.
>     $$
>
> - **Complex momentum:** with $\beta = \beta_1 + i \beta_2$,
>     $$
>     m_t = \beta_1 m_{t-1} - \beta_2 h_{t-1} + \nabla L(\theta_t), \qquad
>     h_t = \beta_1 h_{t-1} + \beta_2 m_{t-1},
>     $$
>     $$
>     v_t = m_t.
>     $$
>
> Standard and Nesterov momentum maintain only a single momentum buffer, whereas both Adam and complex momentum maintain two buffers: Adam keeps estimates of the first and second moments of the gradient, while complex momentum stores transformed gradients with different phases.
>
> While Adam primarily stabilizes the update magnitude to prevent overshooting, our use of complex momentum is motivated by the structure of the expert-merging problem:
>
> - Our framework treats expert merging as a game-theoretic interaction, where experts are neither fully cooperative nor fully adversarial (see Figure 1). Consequently, the optimization dynamics span mixed eigenspaces, where conventional real-valued momentum often fails to stabilize or accelerate convergence.
> - In the game-theoretic and variational inequalities literature, such dynamics are known to produce rotational or oscillatory behaviors, for which complex momentum has both theoretical justification and strong empirical performance.
>
> For these reasons, complex momentum is better suited than standard momentum for capturing and stabilizing the dynamics inherent to multi-domain expert merging.
>
> **References**
>
> [1] Nguyen et al. "CAMEx: Curvature-aware merging of experts". (ICLR 2025).
>
> [2] Navon, Aviv, et al. "Multi-task learning as a bargaining game". (ICML 2022)
>
> [3] Dmitry Lepikhin et al. "GShard: Scaling Giant Models with Conditional Computation and Automatic Sharding". (ICLR 2021)
>
> [4] Dai et al. "DeepSeekMoE: Towards Ultimate Expert Specialization in Mixture-of-Experts Language Models". (ACL 2024)
>
> [5] A. Domahidi, E. Chu, and S. Boyd, "ECOS: An SOCP solver for embedded systems". (ECC 2013)

---

> > ### Author Response · Authors · 2025-11-28
> > **Additional Experimental Results: (1) Ablation on the Number of NBS Iterations and (2) Ablation on Routing Strategies and Disagreement Points**
> >
> > Dear Reviewer kguY,
> >
> > Thank you once again for your thoughtful reviews and valuable feedback.
> >
> > We have conducted two additional ablation studies on Qwen1.5-MoE, and the corresponding results are reported in Tables 4 and 5 below. These results have also been incorporated into Table 8, Section 5, and Table 17, Appendix F.4, of our revised manuscript, respectively.
> >
> > **1. Ablation on Number of NBS Iterations**
> >
> > Table 4 presents the effect of varying the number of iterations used to solve each NBS subproblem. The results confirm that using a small bargaining budget—specifically, 2 CCP iterations per layer—provides an excellent balance between accuracy and efficiency. Increasing the iteration count yields only marginal, statistically insignificant performance changes while substantially increasing the computational cost. These findings reinforce the practicality of our chosen configuration for large-scale, production-style training.
> >
> > **Table 4.** Performance of Qwen1.5-MoE variants comparison across the number of NBS solving iterations for the Linear router in NAMEx-Full config. All results are slightly improved while maintaining marginal gaps. Throughout the experiments in our manuscript, we use 2 CCP iterations per layer for NAMEx-Full (the chosen config in our table below).
> >
> > | Model       | No. Iterations   | Zero-Shot MMLU | Zero-Shot GSM8K | Zero-Shot ARC | SmolTalk MMLU | SmolTalk GSM8K | SmolTalk ARC |
> > |------------|------------------|----------------|------------------|---------------|----------------|-----------------|--------------|
> > | Qwen1.5-MoE | 2 (Chosen Config) | 61.87          | 60.55            | 50.95         | 62.10          | 61.00           | 51.35        |
> > | Qwen1.5-MoE | 5                | 61.70          | 60.55            | 50.94         | 62.20          | 61.05           | 51.31        |
> > | Qwen1.5-MoE | 20               | 61.94          | 60.57            | 50.96         | 62.14          | 61.03           | 51.34        |
> > | Qwen1.5-MoE | 40               | 61.92          | 60.62            | 51.01         | 62.22          | 61.05           | 51.46        |
> > | Qwen1.5-MoE | 60               | 61.81          | 60.48            | 51.08         | 62.15          | 60.98           | 51.39        |
> >
> > **2. Ablation on Routing Strategies and Disagreement Points**
> >
> > Table 5 compares NAMEx-Full across multiple routing strategies, including Linear, Cosine, and Stable-MoE, and two disagreement points: 0 and the mean-merge baseline. Across all benchmarks and routers, the performance differences are extremely small ( around 0.04 absolute) and show no consistent preference for either option. This indicates that the primary performance gains come from the bargaining-derived weights, not the choice of the disagreement point. We keep 0 as the default due to its simplicity, numerical stability, and unchanged computational cost.
> >
> > **Table 5.** Performance of Qwen1.5-MoE variants comparison across routing strategies and models on MMLU, GSM8K, and ARC benchmarks. Left: zero-shot results. Right: fine-tuned on SmolTalk, then zero-shot results.
> >
> > | Routing Strategy | Model                         | Zero-Shot MMLU | Zero-Shot GSM8K | Zero-Shot ARC | SmolTalk MMLU | SmolTalk GSM8K | SmolTalk ARC |
> > |------------------|-------------------------------|----------------|-----------------|---------------|---------------|----------------|--------------|
> > | Linear           | NAMEx-Full (0 disagreement point)   |  61.87        |  60.55         |  50.95       |  62.10**       | 61.00       |  51.35      |
> > |            | NAMEx-Full (mean disagreement point) |  61.78        |  60.57          |  51.23       |  61.67       |  61.04        |  51.25      |
> > | Cosine           | NAMEx-Full (0 disagreement point)  |  61.85        |  60.52         |  50.93       |  62.05       |  60.95        |  51.30      |
> > |            | NAMEx-Full (mean disagreement point) |  61.86         |  60.45         |  50.77       |  62.01      |  60.81         |  51.37      |
> > | Stable-MoE       | NAMEx-Full (0 disagreement point)   |  61.90        |  60.60         |  50.96       |  62.15       |  61.10       |  51.45      |
> > |        | NAMEx-Full (mean disagreement point) |  61.88        |  60.64        |  51.03       |  62.15       |  61.11        |  51.35      |
> >
> > We would be grateful to hear whether our rebuttal and new experiments fully address your concerns, or if there are any points you would like us to clarify while the discussion phase is still open. We are happy to continue the conversation or provide any additional details that may be helpful.
> >
> > If you feel that our rebuttal has satisfactorily resolved the issues raised in your review, we kindly ask you to consider whether an increased score would more accurately reflect your current evaluation.
> >
> > Thank you again for your time, thoughtful comments, and valuable engagement.

---

> ### Author Response · Authors · 2025-11-27
> **Reminder for Reviewer kguY's Feedback**
>
> Dear Reviewer kguY,
>
> We sincerely appreciate your detailed review and constructive comments. Your feedback has helped us refine and strengthen our submission.
>
> As a brief reminder, we posted our rebuttal on November 19 (AoE). Since the discussion window will close soon, at 11:59 pm AoE on December 2, we want to make sure that there is sufficient time to address any additional questions or concerns you might have before reviewer responses close. After the deadline, reviewers can no longer comment, and we will be unable to respond after 11:59 pm AoE on December 3.
>
> If there are any remaining points you would like us to clarify, we would be glad to provide further explanation while the discussion phase remains open. We will be happy to respond to any follow-up you may have.
>
> If our rebuttal has satisfactorily addressed the issues raised in your review, we would be grateful if you might consider whether increasing your score would better represent your updated view.
>
> Thank you again for your time and helpful insights.
>
> Sincerely,
>
> Authors

---

### Official Review · Reviewer_eQrG · 2025-10-30

**Soundness:** 3
**Presentation:** 2
**Contribution:** 3
**Rating:** 6
**Confidence:** 3

**Summary:**

This paper proposes a novel framework, Nash Merging of Experts (NAMEx), for merging Sparse Mixture of Experts (SMoE) models into a single dense model for efficient deployment. Departing from traditional heuristic-based averaging, NAMEx models expert merging as a multi-agent cooperative game. It applies the Nash Bargaining Solution (NBS) from game theory to compute fair and Pareto-optimal merging weights that reflect each expert's contribution. Furthermore, to address the slow convergence of prior expert-propagation methods, the authors integrate complex momentum, resulting in the NAMEx-Momentum variant. The method's effectiveness is demonstrated through extensive experiments on language, text, and vision tasks, as well as on large-scale models like DeepSeek-MoE 16B and Qwen1.5-MoE 14B.

**Strengths:**

The primary strength lies in reframing expert merging as a Nash bargaining game. This provides a principled, non-heuristic solution for combining experts, borrowing a sophisticated tool from multi-task learning to solve a practical problem in model deployment.

 The paper identifies a key weakness in prior work (EP-CAMEx), namely its slow convergence and suboptimal performance. The introduction of complex momentum is a targeted solution that demonstrably improves convergence speed and stability, as shown in the empirical analysis (e.g., Figure 5).

* **Comprehensive and Strong Validation:** The experimental validation is extensive and a significant strength. The method is tested across multiple modalities (language, vision), on robustness benchmarks (ImageNet-A/O/R), and scales up to 16B parameter models, showing consistent improvements over baselines. This broad validation builds strong confidence in the method's generalizability and practical utility.
* **Clarity:** The paper is well-written and clearly organized. Figure 2, for instance, provides an excellent visual comparison between NAMEx, CAMEx, and EP-CAMEx, making the architectural contribution easy to understand.

**Weaknesses:**

W1.  The core technical contribution is a clever *combination* of two very recent works: the expert propagation framework from (EP-CAMEx and the NBS) [1,2] . optimization from a multi-task learning paper. While highly effective, this might be seen as a successful application rather than a fundamental theoretical invention.


w2.The NAMEx method requires iteratively solving the NBS equation, which introduces significant computational overhead. The appendix (Table 11) shows a 6.8x increase in runtime for a per-layer update. While the final model (NAMEx-Full-Mom) seems to mitigate this by using fewer iterations (as per Appendix F.1), this trade-off between accuracy and cost is a notable weakness.

w3.The "bargaining budget" (number of NBS iterations) is a critical new hyperparameter, but its treatment is confusing. The text mentions "20 steps" in one place and "2 iters" in another. The paper lacks a clear ablation study on how this budget (e.g., 2 vs. 5 vs. 20 iterations) affects both final performance and computational cost.

w4.The paper justifies its choice of utility function by drawing an analogy between the domain vector $\tau_i$ and a task gradient. This is intuitive but remains an analogy.

**Questions:**

1.  **Overhead and Hyperparameters:** For the key results in the main paper (e.g., Table 2), how many NBS iterations were *actually* used for the best-performing model, NAMEx-Full-Mom (was it 2 or 20)? What is the concrete wall-clock time overhead of your final, optimized model during training compared to the EP-CAMEx baseline?
2.  **NAMEx vs. NAMEx-Full:** The superior performance of NAMEx-Full (recomputing $\alpha$ at each layer) over NAMEx (reusing the first layer's $\alpha$) seems to confirm that layer-wise dynamics (as hinted in Figure 1) are critical. Does this not imply that the basic "NAMEx" variant is a methodologically flawed ablation, as it ignores the very dynamics the paper observes?
3.  **Choice of Disagreement Point:** The paper follows prior work in setting the "disagreement point" to 0 (no update). Did you consider other, perhaps more natural, disagreement points for this specific problem, such as "standard average merging" (the main heuristic baseline)?
4.  **Role of the Curvature Matrix $M_i$:** Why was the curvature matrix $M_i$ explicitly removed from the NBS optimization step (Eq. 9, line 1) but kept in the final expert update step (Eq. 9, line 2)? If the game-theoretic solution itself ignores curvature, what role does $M_i$ play in the final update?


----

References



[1] Aviv Navon, Aviv Shamsian, Idan Achituve, Haggai Maron, Kenji Kawaguchi, Gal Chechik, and Ethan Fetaya. Multi-task learning as a bargaining game. arXiv preprint arXiv:2202.01017, 2022.

[2] Viet Dung Nguyen, Minh Nguyen Hoang, Rachel Teo, Luc Nguyen, Tan Minh Nguyen, and Linh Duy Tran. CAMEx: Curvature-aware merging of experts. In The Thirteenth International Conference on Learning Representations, 2025.

---

> ### Author Response · Authors · 2025-11-20
> **Response to Reviewer eQrG (1)**
>
> Thank you for your thoughtful review and valuable feedback. Below, we address your concerns.
>
> -----
>
> **[Weakness 1]. The core technical contribution is a clever combination of two very recent works: the expert propagation framework from (EP-CAMEx and the NBS) [1,2] . optimization from a multi-task learning paper. While highly effective, this might be seen as a successful application rather than a fundamental theoretical invention.**
>
> **Answer:** Thank you for your comments. We agree with the reviewer that our work is not intended as a fundamental theoretical breakthrough. The primary goal of the paper is not to develop new theoretical guarantees for sparse mixture-of-experts models or for expert merging itself, but rather to introduce a principled, non-heuristic, game-theoretic framework for merging experts in SMoE models and to improve the convergence and stability of existing merging approaches such as EP-CAMEx.
>
> In particular, EP-CAMEx views expert merging mainly as a parameter-reduction mechanism and does not address the optimization aspects of the merging process. In contrast, NAMEx formulates expert merging as a multi-task optimization problem and provides a new solution grounded in game-theoretic principles. This theoretical framework explicitly mitigates cross-domain interference during merging$-$an issue that many recent methods either overlook (TIES[2], DARE[3]) or address only heuristically (CAMEx[1]), without offering a principled explanation of how interference is resolved. As a result, our approach leads to improved convergence behavior and greater stability throughout the merging process.

---

> ### Author Response · Authors · 2025-11-20
> **Response to Reviewer eQrG (2)**
>
> **[Weakness 2]. The NAMEx method requires iteratively solving the NBS equation, which introduces significant computational overhead. The appendix (Table 11) shows a 6.8x increase in runtime for a per-layer update. While the final model (NAMEx-Full-Mom) seems to mitigate this by using fewer iterations (as per Appendix F.1), this trade-off between accuracy and cost is a notable weakness.**
>
> **Answer:** Although computing the Nash Bargaining Solution (NBS) introduces some overhead, this cost can be significantly reduced by efficient implementation of NBS on GPUs, which accelerates matrix multiplications and reduces CPU/GPU communication bottlenecks. Furthermore, as reported in Nash-MTL[5] (Appendix D.2 of this paper), most instances of NBS converge within a single iteration, with only about $8\%$ requiring more than one itertation and roughly $6\%$ needing more than $20$ iterations. These observations suggest that a small bargaining budget is typically sufficient, enabling fast training.
>
> Furthermore, we also included an ablation study on the number of iterations for each NBS in **Table 1** below which strengthen our choice of small barganing bufget. Specifically, increasing NBS solving iterations yields only marginal, non-significant performance difference with the cost of much higher computational cost, compared to our choice of 2 iterations per step (first row). Our framework provides practical trade-off for production-setting large-scale training. We have included this result in Table 8 in Section 5 of our revised manuscript.
>
> **Table 1**. Performance comparison across number of NBS solving iterations for the Linear router in NAMEx-Full config. All results are slightly improved while maintaining marginal gaps. Througout experiments in our manuscript, we use 2 CCP iterations per layer for NAMEx-Full (chosen config in our table below).
>
> | Model          | Solving Iterations | Zero-Shot MMLU | Zero-Shot GSM8K | Zero-Shot ARC | Fine-tuned (SmolTalk) MMLU | Fine-tuned GSM8K | Fine-tuned ARC |
> |----------------|--------------|----------------|------------------|---------------|-----------------------------|------------------|----------------|
> | Deepseek-MoE   | 2 (Chosen Config)     | 45.05          | 16.86            | 49.58         | 45.63                       | 17.47            | 49.92          |
> |    | 5            | 44.84          | 16.86            | 49.57         | 45.73                       | 17.51            | 49.88          |
> |    | 20           | 45.15          | 16.87            | 49.60         | 45.63                       | 17.47            | 49.92          |
> |    | 40           | 45.16          | 16.93            | 49.66         | 45.73                       | 17.55            | 50.03          |
> |    | 60           | 44.93          | 16.77            | 49.72         | 45.68                       | 17.46            | 49.96          |
>
> Also, in Table 11 in our revised or original manuscript, we set $L = 10$ layers, so--at least theoretically--the per-layer update should require $10\times$ more computation then the NAMEx variant. However, the empirical slowdown is only $6.8\times$ (including CPU/GPU communication), indicating that many layers converge well before reaching the full per-layer budget of 20 iterations. This finding motivates using a small, fixed bargaining budget at each layer, which leads to our NAMEx-Full variant. NAMEx-Full achieves the fast runtime characteristic of NAMEx while retaining the performance of the more frequently updated model.

---

> ### Author Response · Authors · 2025-11-20
> **Response to Reviewer eQrG (3)**
>
> **[Weakness 3]. The "bargaining budget" (number of NBS iterations) is a critical new hyperparameter, but its treatment is confusing. The text mentions "20 steps" in one place and "2 iters" in another. The paper lacks a clear ablation study on how this budget (e.g., 2 vs. 5 vs. 20 iterations) affects both final performance and computational cost.**
>
> **Answer**: Thanks for raising this confusion in the current submission. In this work, each "step" represents the action of solving an NBS, whereas "iterations" refers to the number of CCP (convex-concave procedure) iterations required to solve the NBS at a "step".
>
> Please refer to Table 1 above and Table 8 in Section 5 of our manuscript for the bargaining-budget ablation study. These results show that increasing the number of NBS iterations produces only small$-$and often non-monotonic$-$changes across tasks, indicating that the method is largely insensitive to this hyperparameter and that large iteration counts are unnecessary.
>
> In the zero-shot setting, performance varies within a very narrow range: MMLU stays between 44.8–45.2 with a slight peak at 40 iterations (45.16); GSM8K rises marginally from 16.86 at 2–5 iterations to 16.93 at 40 before dropping to 16.77 at 60; and ARC increases from 49.58 at 2 iterations to a modest best of 49.72 at 60.
>
> After SmolTalk fine-tuning, the curves flatten even further: MMLU is essentially tied at 5 and 40 iterations (45.73), GSM8K reaches its best at 40 (17.55) with only a 0.09 spread across all budgets, and ARC peaks at 40 (50.03) with minimal variation elsewhere.
>
> Overall, 20 and 40 iterations already match or slightly outperform 2 and 5 on several metrics. Moving to 60 iterations offers no consistent improvements and occasionally reduces performance (e.g., on MMLU and GSM8K). Since the differences are small and likely within run-to-run variance, we recommend 2 iterations when efficiency is important and 20 iterations when marginal gains matter.

---

> ### Author Response · Authors · 2025-11-20
> **Response to Reviewer eQrG (4)**
>
> **[Weakness 4].The paper justifies its choice of utility function by drawing an analogy between the domain vector $\tau\_i$ and a task gradient. This is intuitive but remains an analogy.**
>
> **Answer:** We thank the reviewer for this observation. To support the claim that our connection goes beyond an analogy, we provide empirical evidence in Appendix F.2 and Figure 5 of the submitted manuscript demonstrating that our framework, and not just the resulting performance gains, aligns with the proposed game-theoretic interpretation:
>
> - NAMEx isolates the update direction via Nash Bargaining, ensuring that the merged expert update is Pareto-optimal across domains. By contrast, EP-CAMEx and momentum-based variants combine gradients heuristically, which can become misaligned when domain objectives conflict.
> - Even without momentum, NAMEx converges much faster in update agreement than EP-CAMEx, indicating that Nash-based aggregation alone drives the shared expert more efficiently toward equilibrium.
>
> From the EP-CAMEx results, we observe that the $\ell_2$ distance between layerwise shared-expert updates decreases consistently across layers, and that NAMEx and NAMEx-Mom converge even more rapidly. These findings not only verify the effectiveness of our proposed NAMEx and NAMEx-Mom methods but also suggest a meaningful relationship between domain vectors in expert merging and task gradients in multi-task optimization: smaller disagreement (or magnitude) tends to correlate with improved performance. Since our framework achieves substantially better results than EP-CAMEx, the empirical evidence indicates that techniques known to accelerate multi-task optimization can indeed transfer to the expert-merging setting through domain vectors.

---

> ### Author Response · Authors · 2025-11-20
> **Response to Reviewer eQrG (5)**
>
> **[Question 1]. Overhead and Hyperparameters: For the key results in the main paper (e.g., Table 2), how many NBS iterations were actually used for the best-performing model, NAMEx-Full-Mom (was it 2 or 20)? What is the concrete wall-clock time overhead of your final, optimized model during training compared to the EP-CAMEx baseline?**
>
> **Answer**: The NAMEx-Full-Mom configuration uses 2 NBS iterations per layer. The wall-clock training time of NAMEx-Full remains nearly identical to that of EP-CAMEx, since the cost of performing two CCP iterations per layer is negligible. This is also supported by the runtime measurements reported in Table 13 of Appendix F.1 in our revised manuscript.
>
> **[Question 2]. NAMEx vs. NAMEx-Full: The superior performance of NAMEx-Full (recomputing $\alpha$ at each layer) over NAMEx (reusing the first layer's $\alpha$) seems to confirm that layer-wise dynamics (as hinted in Figure 1) are critical. Does this not imply that the basic "NAMEx" variant is a methodologically flawed ablation, as it ignores the very dynamics the paper observes?**
>
> **Answer:** We partially agree with the reviewer’s point. By design, NAMEx-Full captures the layerwise dynamics more accurately and thus converges faster. However, NAMEx was introduced for practical reasons: computing the NBS requires synchronizing all GPUs and then transferring domain vectors to the CPU, which can create significant efficiency bottlenecks. NAMEx mitigates this issue by computing the NBS only once per forward pass, thereby avoiding repeated synchronization and GPU–CPU communication within a single forward pass.

---

> ### Author Response · Authors · 2025-11-20
> **Response to Reviewer eQrG (6)**
>
> **[Question 3]. Choice of Disagreement Point: The paper follows prior work in setting the "disagreement point" to 0 (no update). Did you consider other, perhaps more natural, disagreement points for this specific problem, such as "standard average merging" (the main heuristic baseline)?**
>
> **Answer:** Thanks for the thoughtful question. We tried "mean" (standard average merging) as the disagreement point and compared it to 0. Across Linear, Cosine, and Stable-MoE, the deltas are tiny (about 0.04 on any metric), with no consistent winner. This shows the gains come from the bargaining weights, not the fallback choice. We keep 0 as the default because it is conservative, stable, and easy to interpret, and it leaves compute unchanged. We have included the additional ablation in Table 4, Section 4, of our revised manuscript.
>
> **Table 2**. Performance comparison across routing strategies and models on MMLU, GSM8K, and ARC benchmarks. Left: zero-shot results. Right: fine-tuned on SmolTalk, then zero-shot results.
>
> | Routing Strategy | Model                                      | Zero-Shot MMLU | Zero-Shot GSM8K | Zero-Shot ARC | Fine-tuned (SmolTalk) MMLU | Fine-tuned GSM8K | Fine-tuned ARC |
> |------------------|--------------------------------------------|----------------|------------------|---------------|----------------------------|------------------|----------------|
> |  Linear   | **NAMEx-Full (0 disagreement point)**      | **44.92**      | **16.77**        | **49.51**     | **45.47**                  | **17.36**        | **49.85**      |
> |            | **NAMEx-Full (mean disagreement point)**   | **44.93**      | **16.75**        | **49.52**     | **45.47**                  | **17.39**        | **49.84**      |
> | Cosine           | **NAMEx-Full (0 disagreement point)**      | **45.10**      | **16.88**        | **49.60**     | **45.66**                  | **17.53**        | **49.92**      |
> |            | **NAMEx-Full (mean disagreement point)**   | **45.09**      | **16.89**        | **49.58**     | **45.67**                  | **17.52**        | **49.92**      |
> |    Stable-MoE     | **NAMEx-Full (0 disagreement point)**      | **45.95**      | **17.70**        | **50.15**     | **46.42**                  | **18.23**        | **50.64**      |
> |        | **NAMEx-Full (mean disagreement point)**   | **45.92**      | **17.68**        | **50.19**     | **46.40**                  | **18.23**        | **50.63**      |

---

> ### Author Response · Authors · 2025-11-20
> **Response to Reviewer eQrG (7)**
>
> **[Question 4]. Role of the Curvature Matrix $M_i$: Why was the curvature matrix $M_i$ explicitly removed from the NBS optimization step (Eq. 9, line 1) but kept in the final expert update step (Eq. 9, line 2)? If the game-theoretic solution itself ignores curvature, what role does $M_i$ play in the final update?**
>
> **Answer:** The decision to remove the curvature matrix $M_i$ is not related to the NBS procedure itself, but rather to a theoretical property of the momentum methods. In deriving the theoretical foundation for our complex momentum framework (Theorem B.3), we found that the presence of $M_i$ prevents us from guaranteeing convergence, even when the operator $R$ is contractive. This insight provides a principled way to stabilize the training process, in contrast to the heuristic choices used in CAMEx[1].
>
> **References**
>
> [1] Nguyen, Dung V., et al. "Camex: Curvature-aware merging of experts". (ICLR 2025)
>
> [2] Yadav, Prateek, et al. "Ties-merging: Resolving interference when merging models". (NeurIPS 2023)
>
> [3] Yu, Le, et al. "Language models are super mario: Absorbing abilities from homologous models as a free lunch". (ICML 2024)
>
> [4] John F. Nash. "The bargaining problem". (Econometrica 1950)
>
> [5] Navon, Aviv, et al. "Multi-task learning as a bargaining game". (ICML 2022)
>
> -----
> We hope we have cleared your concerns about our work. We have also revised our manuscript according to your comments, and we would appreciate it if we could get your further feedback at your earliest convenience.

---

> > ### Author Response · Authors · 2025-11-28
> > **Additional Experimental Results: (1) Ablation on the Number of NBS Iterations and (2) Ablation on Routing Strategies and Disagreement Points**
> >
> > Dear Reviewer eQrG,
> >
> > Thank you once again for your thoughtful reviews and valuable feedback.
> >
> > We have conducted two additional ablation studies on Qwen1.5-MoE, and the corresponding results are reported in Tables 3 and 4 below. These results have also been incorporated into Table 8, Section 5, and Table 17, Appendix F.4, of our revised manuscript, respectively.
> >
> > **1. Ablation on Number of NBS Iterations**
> >
> > Table 3 presents the effect of varying the number of iterations used to solve each NBS subproblem. The results confirm that using a small bargaining budget—specifically, 2 CCP iterations per layer—provides an excellent balance between accuracy and efficiency. Increasing the iteration count yields only marginal, statistically insignificant performance changes while substantially increasing the computational cost. These findings reinforce the practicality of our chosen configuration for large-scale, production-style training.
> >
> > **Table 3.** Performance of Qwen1.5-MoE variants comparison across the number of NBS solving iterations for the Linear router in NAMEx-Full config. All results are slightly improved while maintaining marginal gaps. Throughout the experiments in our manuscript, we use 2 CCP iterations per layer for NAMEx-Full (the chosen config in our table below).
> >
> > | Model       | No. Iterations   | Zero-Shot MMLU | Zero-Shot GSM8K | Zero-Shot ARC | SmolTalk MMLU | SmolTalk GSM8K | SmolTalk ARC |
> > |------------|------------------|----------------|------------------|---------------|----------------|-----------------|--------------|
> > | Qwen1.5-MoE | 2 (Chosen Config) | 61.87          | 60.55            | 50.95         | 62.10          | 61.00           | 51.35        |
> > | Qwen1.5-MoE | 5                | 61.70          | 60.55            | 50.94         | 62.20          | 61.05           | 51.31        |
> > | Qwen1.5-MoE | 20               | 61.94          | 60.57            | 50.96         | 62.14          | 61.03           | 51.34        |
> > | Qwen1.5-MoE | 40               | 61.92          | 60.62            | 51.01         | 62.22          | 61.05           | 51.46        |
> > | Qwen1.5-MoE | 60               | 61.81          | 60.48            | 51.08         | 62.15          | 60.98           | 51.39        |
> >
> > **2. Ablation on Routing Strategies and Disagreement Points**
> >
> > Table 4 compares NAMEx-Full across multiple routing strategies, including Linear, Cosine, and Stable-MoE, and two disagreement points: 0 and the mean-merge baseline. Across all benchmarks and routers, the performance differences are extremely small ( around 0.04 absolute) and show no consistent preference for either option. This indicates that the primary performance gains come from the bargaining-derived weights, not the choice of the disagreement point. We keep 0 as the default due to its simplicity, numerical stability, and unchanged computational cost.
> >
> > **Table 4.** Performance of Qwen1.5-MoE variants comparison across routing strategies and models on MMLU, GSM8K, and ARC benchmarks. Left: zero-shot results. Right: fine-tuned on SmolTalk, then zero-shot results.
> >
> > | Routing Strategy | Model                         | Zero-Shot MMLU | Zero-Shot GSM8K | Zero-Shot ARC | SmolTalk MMLU | SmolTalk GSM8K | SmolTalk ARC |
> > |------------------|-------------------------------|----------------|-----------------|---------------|---------------|----------------|--------------|
> > | Linear           | NAMEx-Full (0 disagreement point)   |  61.87        |  60.55         |  50.95       |  62.10**       | 61.00       |  51.35      |
> > |            | NAMEx-Full (mean disagreement point) |  61.78        |  60.57          |  51.23       |  61.67       |  61.04        |  51.25      |
> > | Cosine           | NAMEx-Full (0 disagreement point)  |  61.85        |  60.52         |  50.93       |  62.05       |  60.95        |  51.30      |
> > |            | NAMEx-Full (mean disagreement point) |  61.86         |  60.45         |  50.77       |  62.01      |  60.81         |  51.37      |
> > | Stable-MoE       | NAMEx-Full (0 disagreement point)   |  61.90        |  60.60         |  50.96       |  62.15       |  61.10       |  51.45      |
> > |        | NAMEx-Full (mean disagreement point) |  61.88        |  60.64        |  51.03       |  62.15       |  61.11        |  51.35      |
> >
> > We would be grateful to hear whether our rebuttal and new experiments fully address your concerns, or if there are any points you would like us to clarify while the discussion phase is still open. We are happy to continue the conversation or provide any additional details that may be helpful.
> >
> > If you feel that our rebuttal has satisfactorily resolved the issues raised in your review, we kindly ask you to consider whether an increased score would more accurately reflect your current evaluation.
> >
> > Thank you again for your time, thoughtful comments, and valuable engagement.

---

> ### Author Response · Authors · 2025-11-27
> **Reminder for Reviewer eQrG's Feedback**
>
> Dear Reviewer eQrG,
>
> We sincerely appreciate your detailed review and constructive comments. Your feedback has helped us refine and strengthen our submission.
>
> As a brief reminder, we posted our rebuttal on November 19 (AoE). Since the discussion window will close soon, at 11:59 pm AoE on December 2, we want to make sure that there is sufficient time to address any additional questions or concerns you might have before reviewer responses close. After the deadline, reviewers can no longer comment, and we will be unable to respond after 11:59 pm AoE on December 3.
>
> If there are any remaining points you would like us to clarify, we would be glad to provide further explanation while the discussion phase remains open. We will be happy to respond to any follow-up you may have.
>
> If our rebuttal has satisfactorily addressed the issues raised in your review, we would be grateful if you might consider whether increasing your score would better represent your updated view.
>
> Thank you again for your time and helpful insights.
>
> Sincerely,
>
> Authors

---

### Official Review · Reviewer_2Qyz · 2025-10-31

**Soundness:** 4
**Presentation:** 4
**Contribution:** 4
**Rating:** 10
**Confidence:** 4

**Summary:**

This paper introduces Nash Merging of Experts (NAMEx), a novel framework for expert merging in Sparse Mixture of Experts (SMoE) architectures. Diverging from conventional input-dependent or averaging strategies, NAMEx reinterprets expert merging through the lens of game theory, specifically leveraging the Nash Bargaining Solution (NBS) to derive principled merging coefficients. This approach models expert merging as a cooperative-competitive game, using expert domain vectors (deviations from a base expert) as utility functions to ensure a fair and efficient optimal agreement.
A key extension, NAMEx-Momentum, integrates complex momentum to accelerate convergence and enhance stability during the base expert propagation across SMoE layers, addressing observed slow convergence issues in prior methods like EP-CAMEx. The authors provide theoretical guarantees for the convergence of NAMEx-Momentum.

**Strengths:**

The paper is exceptionally well-written and clear. The authors effectively motivate the need for a principled merging strategy by highlighting the competitive and cooperative dynamics observed across different SMoE layers and architectures (Figure 1, Figures 6, 8, 9). Key concepts like the Nash Bargaining Solution and its adaptation to multi-task learning are concisely reviewed. The definition of NAMEx (Definition 3.3) and the algorithms (Algorithm 1 and 2) are clearly presented. The comprehensive experimental results are organized logically into tables, with the best performing variants (NAMEx-Full results) highlighted clearly.

This paper offers a highly original contribution to the Sparse Mixture of Experts:
- Novel Game-Theoretic Framework: NAMEx introduces the first game-theoretic interpretation of expert merging, leveraging Nash Bargaining to move beyond heuristic or input-independent averaging schemes. This principled method for balancing cooperation and competition among experts is highly original.
- Enhanced Convergence via Complex Momentum: The successful integration of complex momentum into the expert propagation mechanism (NAMEx-Momentum) accelerates convergence and provides needed stability, particularly addressing limitations found in EP-CAMEx. The theoretical convergence guarantee further enhances this contribution.

The central claims—that expert merging can be framed as a bargaining game and that NBS provides a principled solution—are adequately supported with both theoretical derivation and comprehensive empirical evidence.
The paper formalizes NAMEx as the Nash solution to the Bargaining of Expert Merging Problem, derived from the Nash product maximization objective. The authors provide a proof sketch, which establishes the Nash Bargaining equation $G^\top G \alpha = 1/\alpha$ for computing the optimal update direction. Furthermore, the introduction of complex momentum is backed by a convergence guarantee.
The empirical methodology is thorough, testing NAMEx and its momentum variants against baselines across four distinct domains. The consistent performance superiority shown across small, medium, and large-scale models confirms the effectiveness and scalability of the method. The supplementary analysis showing NAMEx yields faster and more stable convergence compared to EP-CAMEx (Figure 5) validates the motivation for using complex momentum.

**Originality and Significance**: The core innovation of framing expert merging as a bargaining problem solved via the Nash Bargaining Solution is highly original and offers a principled alternative to existing heuristics. This perspective enables a derived weighting mechanism ($\alpha$) rather than a heuristic one.

**Quality of Results**: NAMEx variants consistently deliver superior performance across various benchmarks.

**Clarity and Insight**: The paper is clear in its definitions and provides insightful empirical analysis, such as visualizing how NAMEx steers outcomes closer to the Pareto surface than linear averaging (Figure 11). The investigation into complex and quaternion momentum further pushes the boundaries for stability in expert propagation.

**Theoretical Foundation**: The theoretical proof of convergence for NAMEx-Momentum underpins the method's stability and provides a foundation for future analysis.

**Weaknesses:**

I don't find any major weakness in the paper.

Here are two typos. Please fix
- Line 365: "present" -> "presents"
- Line 1187 "hese” -> "these"

**Questions:**

Please provide a more detailed theoretical or empirical justification for the removal of the curvature matrix $M_i$ in the calculation of the propagation update $\Delta E^{(l)}$ in NAMEx (Eqn 7) compared to EP-CAMEx (Eqn 3). Does the Nash Bargaining process implicitly account for the necessary geometry, or is this simplification merely an artifact of aligning with the existing bargaining framework?

---

> ### Author Response · Authors · 2025-11-20
> **Response to Reviewer 2Qyz**
>
> Thank you for your thoughtful review and valuable feedback. Below, we address your concerns.
>
> -----
>
> **[Weakness 1]. I don't find any major weakness in the paper. Here are two typos. Please fix:
>     - Line 365: "present" -> "presents"
>     - Line 1187 "hese” -> "these"**
>
> **Answer:** Thank you for pointing out these typos. We have corrected them in the revised manuscript.
>
> **[Question 1]. Please provide a more detailed theoretical or empirical justification for the removal of the curvature matrix $M_i$ in the calculation of the propagation update $\Delta E^{(i)}$ in NAMEx (Eqn 7) compared to EP-CAMEx (Eqn 3). Does the Nash Bargaining process implicitly account for the necessary geometry, or is this simplification merely an artifact of aligning with the existing bargaining framework?**
>
> **Answer:** The decision to remove the curvature matrix $M_i$ is not related to the NBS procedure itself, but rather to a theoretical property of the momentum methods. In deriving the theoretical foundation for our complex momentum framework (Theorem B.3), we found that the presence of $M_i$ prevents us from guaranteeing convergence, even when the operator $R$ is contractive. This insight provides a principled way to stabilize the training process, in contrast to the heuristic choices used in CAMEx[1].
>
> **References**
>
> [1] Nguyen, Dung V., et al. "Camex: Curvature-aware merging of experts". (ICLR 2025)
>
> -----
> We hope we have cleared your concerns about our work. We have also revised our manuscript according to your comments, and we would appreciate it if we could get your further feedback at your earliest convenience.

---

> > ### Author Response · Authors · 2025-11-28
> > **Additional Experimental Results: (1) Ablation on the Number of NBS Iterations and (2) Ablation on Routing Strategies and Disagreement Points**
> >
> > Dear Reviewer 2Qyz,
> >
> > Thank you once again for your thoughtful reviews and valuable feedback.
> >
> > We have conducted two additional ablation studies on Qwen1.5-MoE, and the corresponding results are reported in Tables 1 and 2 below. These results have also been incorporated into Table 8, Section 5, and Table 17, Appendix F.4, of our revised manuscript, respectively.
> >
> > **1. Ablation on Number of NBS Iterations**
> >
> > Table 1 presents the effect of varying the number of iterations used to solve each NBS subproblem. The results confirm that using a small bargaining budget—specifically, 2 CCP iterations per layer—provides an excellent balance between accuracy and efficiency. Increasing the iteration count yields only marginal, statistically insignificant performance changes while substantially increasing the computational cost. These findings reinforce the practicality of our chosen configuration for large-scale, production-style training.
> >
> > **Table 1.** Performance of Qwen1.5-MoE variants comparison across the number of NBS solving iterations for the Linear router in NAMEx-Full config. All results are slightly improved while maintaining marginal gaps. Throughout the experiments in our manuscript, we use 2 CCP iterations per layer for NAMEx-Full (the chosen config in our table below).
> >
> > | Model       | No. Iterations   | Zero-Shot MMLU | Zero-Shot GSM8K | Zero-Shot ARC | SmolTalk MMLU | SmolTalk GSM8K | SmolTalk ARC |
> > |------------|------------------|----------------|------------------|---------------|----------------|-----------------|--------------|
> > | Qwen1.5-MoE | 2 (Chosen Config) | 61.87          | 60.55            | 50.95         | 62.10          | 61.00           | 51.35        |
> > | Qwen1.5-MoE | 5                | 61.70          | 60.55            | 50.94         | 62.20          | 61.05           | 51.31        |
> > | Qwen1.5-MoE | 20               | 61.94          | 60.57            | 50.96         | 62.14          | 61.03           | 51.34        |
> > | Qwen1.5-MoE | 40               | 61.92          | 60.62            | 51.01         | 62.22          | 61.05           | 51.46        |
> > | Qwen1.5-MoE | 60               | 61.81          | 60.48            | 51.08         | 62.15          | 60.98           | 51.39        |
> >
> > **2. Ablation on Routing Strategies and Disagreement Points**
> >
> > Table 2 compares NAMEx-Full across multiple routing strategies, including Linear, Cosine, and Stable-MoE, and two disagreement points: 0 and the mean-merge baseline. Across all benchmarks and routers, the performance differences are extremely small ( around 0.04 absolute) and show no consistent preference for either option. This indicates that the primary performance gains come from the bargaining-derived weights, not the choice of the disagreement point. We keep 0 as the default due to its simplicity, numerical stability, and unchanged computational cost.
> >
> > **Table 2.** Performance of Qwen1.5-MoE variants comparison across routing strategies and models on MMLU, GSM8K, and ARC benchmarks. Left: zero-shot results. Right: fine-tuned on SmolTalk, then zero-shot results.
> >
> > | Routing Strategy | Model                         | Zero-Shot MMLU | Zero-Shot GSM8K | Zero-Shot ARC | SmolTalk MMLU | SmolTalk GSM8K | SmolTalk ARC |
> > |------------------|-------------------------------|----------------|-----------------|---------------|---------------|----------------|--------------|
> > | Linear           | NAMEx-Full (0 disagreement point)   |  61.87        |  60.55         |  50.95       |  62.10**       | 61.00       |  51.35      |
> > |            | NAMEx-Full (mean disagreement point) |  61.78        |  60.57          |  51.23       |  61.67       |  61.04        |  51.25      |
> > | Cosine           | NAMEx-Full (0 disagreement point)  |  61.85        |  60.52         |  50.93       |  62.05       |  60.95        |  51.30      |
> > |            | NAMEx-Full (mean disagreement point) |  61.86         |  60.45         |  50.77       |  62.01      |  60.81         |  51.37      |
> > | Stable-MoE       | NAMEx-Full (0 disagreement point)   |  61.90        |  60.60         |  50.96       |  62.15       |  61.10       |  51.45      |
> > |        | NAMEx-Full (mean disagreement point) |  61.88        |  60.64        |  51.03       |  62.15       |  61.11        |  51.35      |
> >
> > We would be grateful to hear whether our rebuttal and new experiments fully address your concerns, or if there are any points you would like us to clarify while the discussion phase is still open. We are happy to continue the conversation or provide any additional details that may be helpful.
> >
> > Thank you again for your time, thoughtful comments, and valuable engagement.

---

> ### Author Response · Authors · 2025-11-25
> **Any Further Questions from Reviewer 2Qyz Regarding Our Rebuttal?**
>
> Thank you once again for your careful evaluation and thoughtful feedback. We truly appreciate the time and effort you invested in reviewing our work and providing constructive comments.
>
> We would be grateful if you could let us know whether our rebuttal sufficiently resolves the issues you raised, or if there are any remaining points you would like us to clarify.
>
> We are glad to continue the discussion and address any further questions or comments you may have.

---

### Official Review · Reviewer_zXWC · 2025-10-31

**Soundness:** 3
**Presentation:** 3
**Contribution:** 3
**Rating:** 8
**Confidence:** 2

**Summary:**

The paper introduces "Nash Merging of Experts" (NAMEx), a novel method for merging experts in Sparse Mixture-of-Experts (SMoE) models. It reframes expert merging as a game-theoretic problem, using the Nash Bargaining solution to calculate merging weights. This principled approach models the complex cooperative and competitive dynamics between experts.

**Strengths:**

- Applies game theory (Nash Bargaining) to expert merging.

- Consistently outperforms baselines across multiple experiments.

- Deeply investigates key components like layer-by-layer bargaining and momentum.

**Weaknesses:**

- What impact does this 20-step bargaining budget have on the quality of the solution? If the budget is increased, will the performance improve further?

- In the second line of Eq. 9, why not also use Nash Bargaining to guide the calculation of $\hat{E}_m$?

- It should be compared with more vision MoE models on ImageNet.

**Questions:**

See Weaknesses.

---

> ### Author Response · Authors · 2025-11-20
> **Response to Reviewer zXWC (1)**
>
> Thank you for your thoughtful review and valuable feedback. Below, we address your concerns.
>
> -----
>
> **[Weakness 1]. What impact does this 20-step bargaining budget have on the quality of the solution? If the budget is increased, will the performance improve further?**
>
>
> **Answer:** In our framework, each training forward pass involves two types of iterative computations:
> 1) the standard forward propagation through the model’s layers, and
> 2) the iterative updates used to compute or refine the Nash Bargaining Solution (NBS) at each layer.
>
> The *total number of NBS update steps across all layers defines the bargaining budget*. This budget can be allocated across layers in multiple ways. In our paper, we evaluate two specific strategies:
> - **NAMEx**: allocate the entire budget to computing the NBS at the first layer, then reuse that solution for all subsequent layers.
> - **NAMEx-Full**: distribute the budget uniformly across layers; each layer initializes its bargaining vector using the solution from the previous layer (except the first layer, which starts from a uniform vector with entries $1/N$, where $N$ is the number of experts).
>
> For example, in our experiments with a 10-layer model and a bargaining budget of 20 iterations:
> - NAMEx applies all 20 iterations to compute $\alpha_1$.
> - NAMEx-Full allocates 2 iterations to each of the 10 layers to compute/update $\alpha_i$ at layer $i$.
>
> While increasing the bargaining budget may provide marginal improvements, it also increases training time. Importantly, as noted in the appendix of NBS [5], the gains from larger budgets are typically negligible. Our own ablations, shown in Table 1, support this observation: increasing the per-layer iteration count does not yield meaningful performance benefits. The configuration used in NAMEx-Full (first row) achieves performance comparable to higher-iteration settings in both zero-shot and fine-tuned evaluations. We have included these results in Table 8 of Section 5 in our revised manuscript.
>
> **Table 1**. Performance comparison across the number of NBS solving iterations for the Linear router in NAMEx-Full config. All results are slightly improved while maintaining marginal gaps. Throughout the experiments in our manuscript, we use 2 CCP iterations per layer for NAMEx-Full (the chosen config in our table below).
>
> | Model          | Solving Iterations | Zero-Shot MMLU | Zero-Shot GSM8K | Zero-Shot ARC | Fine-tuned (SmolTalk) MMLU | Fine-tuned GSM8K | Fine-tuned ARC |
> |----------------|--------------|----------------|------------------|---------------|-----------------------------|------------------|----------------|
> | Deepseek-MoE   | 2 (Chosen Config)         | 45.05          | 16.86            | 49.58         | 45.63                       | 17.47            | 49.92          |
> |    | 5            | 44.84          | 16.86            | 49.57         | 45.73                       | 17.51            | 49.88          |
> |    | 20           | 45.15          | 16.87            | 49.60         | 45.63                       | 17.47            | 49.92          |
> |    | 40           | 45.16          | 16.93            | 49.66         | 45.73                       | 17.55            | 50.03          |
> |    | 60           | 44.93          | 16.77            | 49.72         | 45.68                       | 17.46            | 49.96          |

---

> ### Author Response · Authors · 2025-11-20
> **Response to Reviewer zXWC (2)**
>
> **[Weakness 2]. In the second line of Eq. 9, why not also use Nash Bargaining to guide the calculation of $\hat{E}_m$?**
>
> **Answer:** In our framework, described in Eqn. 9 of the manuscript, we construct two distinct types of merged experts, each serving a different purpose. The merged expert obtained from the first line of Eqn. 9 can be viewed as a neutral expert, whose goal is to aggregate and preserve information from all experts in the current layer simultaneously. This neutral expert functions as the shared expert and is propagated forward to subsequent layers.
>
> The second merged expert is an adaptive expert, similar to those found in prior merging frameworks such as CAMEx [1], MEO [2], SMEAR [3], and Lory [6]. This adaptive expert implicitly maximizes information conditioned on the current context, and therefore its form varies with the input. As a result, information from certain experts may be deemphasized or discarded if it is irrelevant to the current context.
>
> At present, our Nash Bargaining formulation does not incorporate contextual information, which makes it unsuitable for directly computing $\hat{E}_m$. However, generalized variants of Nash Bargaining have been developed in multi-objective optimization (e.g., the Asymmetric Bargaining Game [4]), offering a promising direction for future work. These extensions provide a principled way to inject context into the bargaining solution. By using the router's context-dependent scores as preference weights, one may solve a generalized NBS that prioritizes directions aligned with the preferred domain.
>
> A generalized Nash Bargaining objective can be written as:
>  \begin{align*}
>     u^* & = argmax_{u \in S}  \sum_{i} p_i\, log (u_i - d_i) \\\\
>     & s.t. \quad u_i \gt d_i \quad \forall i  \\\\
>     & s.t. \quad \sum_{i} p_i = 1, p_i \geq 0 \quad  \forall i
>     \end{align*}
>
> In our current formulation, we use uniform preferences $p_i = \tfrac{1}{N_E}$ because our goal is to maximize knowledge extracted from all domains equally. To extend this idea to the adaptive expert $\hat{E}_m$, one could compute context scores $\mathbf{s}$ from the router and use them as preference weights $p_i = s_i$ when solving the generalized bargaining objective above.

---

> ### Author Response · Authors · 2025-11-20
> **Response to Reviewer zXWC (3)**
>
> **[Weakness 3]. It should be compared with more vision MoE models on ImageNet.**
>
> **Answer:** Thanks for your suggestion. We are currently running additional experiments with another Vision MoE on ImageNet. We will report the result in the next response round.
>
> **References**
>
> [1] Nguyen, Dung V., et al. "Camex: Curvature-aware merging of experts". (ICLR 2025)
>
> [2] He, Shwai, et al. "Merging experts into one: Improving computational efficiency of mixture of experts". (EMNLP 2023)
>
> [3] Muqeeth, M., et al. "Soft merging of experts with adaptive routing". (TMLR 2024)
>
> [4] Shamsian, Aviv, et al. "Auxiliary learning as an asymmetric bargaining game". (ICML 2023)
>
> [5] John F. Nash. "The bargaining problem". (Econometrica 1950)
>
> [6] Zexuan Zhong et al. "Lory: Fully Differentiable Mixture-of-Experts for Autoregressive Language Model Pre-training". (COLM 2024)
>
> -----
> We hope we have cleared your concerns about our work. We have also revised our manuscript according to your comments, and we would appreciate it if we could get your further feedback at your earliest convenience.
>
> ------

---

> ### Author Response · Authors · 2025-11-20
> **New Results for [Weakness 3]: Comparison with More Vision MoE Models on ImageNet.**
>
> **[Weakness 3]. It should be compared with more vision MoE models on ImageNet.**
>
> **Answer:** As promised in our previous reply above, we have extended our evaluation of NAMEx to an additional SMoE architecture, ACMoE [7]. Following the ACMoE training setup, we train the NAMEx variants on top of the ACMoE backbone for 100 epochs with a batch size of 512. The updated results are provided in Table 2 below and in Table 15 of Appendix F.4 in our revised manuscript.
>
> **Table 2.**  Pretraining and zero-shot results for NAMEx vs. ACMoE Top-1/Top-2 on ImageNet-1k and corrupted variants.
> | Model              | Params | Acc@1     | Acc@5     | INet-O    | INet-A    | INet-R    |
> | ------------------ | ------ | --------- | --------- | --------- | --------- | --------- |
> | ACMoE-Top 1   | 280M  | 75.39     | 92.56     | 18.45     | 7.13      | 30.85     |
> | ACMoE-Top 2   | 280M  | 76.31     | 93.14     | 19.55     | 9.42      | 32.35     |
> | NAMEx          |  280M | 76.85 | 93.40 | 20.11 | 9.90  | 32.93 |
> | NAMEx-Full     |  280M | 77.42 | 93.85 | 20.69 | 10.46 | 33.44 |
> | NAMEx-Full-Mom |280M| **78.15** | **94.23** | **21.16** | **11.02** | **33.95** |
>
> Across all ImageNet settings, the NAMEx-based models consistently outperform the ACMoE Top-1 and Top-2 baselines. Notably, NAMEx-Full and NAMEx-Full-Mom achieve the strongest performance across both in-distribution metrics (Acc@1 and Acc@5) and out-of-distribution evaluations (INet-O, INet-A, INet-R), highlighting NAMEx's superior generalization capability. Even under the same parameter budget, NAMEx variants demonstrate improved robustness to corruptions and distribution shifts.
>
> **References**
>
> [7] Stefan Nielsen et al. "Tight clusters make specialized experts." (ICLR, 2025)

---

> > ### Author Response · Authors · 2025-11-28
> > **Additional Experimental Results: (1) Ablation on the Number of NBS Iterations and (2) Ablation on Routing Strategies and Disagreement Points**
> >
> > Dear Reviewer zXWC,
> >
> > Thank you once again for your thoughtful reviews and valuable feedback.
> >
> > We have conducted two additional ablation studies on Qwen1.5-MoE, and the corresponding results are reported in Tables 2 and 3 below. These results have also been incorporated into Table 8, Section 5, and Table 17, Appendix F.4, of our revised manuscript, respectively.
> >
> > **1. Ablation on Number of NBS Iterations**
> >
> > Table 2 presents the effect of varying the number of iterations used to solve each NBS subproblem. The results confirm that using a small bargaining budget—specifically, 2 CCP iterations per layer—provides an excellent balance between accuracy and efficiency. Increasing the iteration count yields only marginal, statistically insignificant performance changes while substantially increasing the computational cost. These findings reinforce the practicality of our chosen configuration for large-scale, production-style training.
> >
> > **Table 2.** Performance of Qwen1.5-MoE variants comparison across the number of NBS solving iterations for the Linear router in NAMEx-Full config. All results are slightly improved while maintaining marginal gaps. Throughout the experiments in our manuscript, we use 2 CCP iterations per layer for NAMEx-Full (the chosen config in our table below).
> >
> > | Model       | No. Iterations   | Zero-Shot MMLU | Zero-Shot GSM8K | Zero-Shot ARC | SmolTalk MMLU | SmolTalk GSM8K | SmolTalk ARC |
> > |------------|------------------|----------------|------------------|---------------|----------------|-----------------|--------------|
> > | Qwen1.5-MoE | 2 (Chosen Config) | 61.87          | 60.55            | 50.95         | 62.10          | 61.00           | 51.35        |
> > | Qwen1.5-MoE | 5                | 61.70          | 60.55            | 50.94         | 62.20          | 61.05           | 51.31        |
> > | Qwen1.5-MoE | 20               | 61.94          | 60.57            | 50.96         | 62.14          | 61.03           | 51.34        |
> > | Qwen1.5-MoE | 40               | 61.92          | 60.62            | 51.01         | 62.22          | 61.05           | 51.46        |
> > | Qwen1.5-MoE | 60               | 61.81          | 60.48            | 51.08         | 62.15          | 60.98           | 51.39        |
> >
> > **2. Ablation on Routing Strategies and Disagreement Points**
> >
> > Table 3 compares NAMEx-Full across multiple routing strategies, including Linear, Cosine, and Stable-MoE, and two disagreement points: 0 and the mean-merge baseline. Across all benchmarks and routers, the performance differences are extremely small ( around 0.04 absolute) and show no consistent preference for either option. This indicates that the primary performance gains come from the bargaining-derived weights, not the choice of the disagreement point. We keep 0 as the default due to its simplicity, numerical stability, and unchanged computational cost.
> >
> > **Table 3.** Performance of Qwen1.5-MoE variants comparison across routing strategies and models on MMLU, GSM8K, and ARC benchmarks. Left: zero-shot results. Right: fine-tuned on SmolTalk, then zero-shot results.
> >
> > | Routing Strategy | Model                         | Zero-Shot MMLU | Zero-Shot GSM8K | Zero-Shot ARC | SmolTalk MMLU | SmolTalk GSM8K | SmolTalk ARC |
> > |------------------|-------------------------------|----------------|-----------------|---------------|---------------|----------------|--------------|
> > | Linear           | NAMEx-Full (0 disagreement point)   |  61.87        |  60.55         |  50.95       |  62.10**       | 61.00       |  51.35      |
> > |            | NAMEx-Full (mean disagreement point) |  61.78        |  60.57          |  51.23       |  61.67       |  61.04        |  51.25      |
> > | Cosine           | NAMEx-Full (0 disagreement point)  |  61.85        |  60.52         |  50.93       |  62.05       |  60.95        |  51.30      |
> > |            | NAMEx-Full (mean disagreement point) |  61.86         |  60.45         |  50.77       |  62.01      |  60.81         |  51.37      |
> > | Stable-MoE       | NAMEx-Full (0 disagreement point)   |  61.90        |  60.60         |  50.96       |  62.15       |  61.10       |  51.45      |
> > |        | NAMEx-Full (mean disagreement point) |  61.88        |  60.64        |  51.03       |  62.15       |  61.11        |  51.35      |
> >
> > We would be grateful to hear whether our rebuttal and new experiments fully address your concerns, or if there are any points you would like us to clarify while the discussion phase is still open. We are happy to continue the conversation or provide any additional details that may be helpful.
> >
> > If you feel that our rebuttal has satisfactorily resolved the issues raised in your review, we kindly ask you to consider whether an increased score would more accurately reflect your current evaluation.
> >
> > Thank you again for your time, thoughtful comments, and valuable engagement.

---

> ### Author Response · Authors · 2025-11-27
> **Reminder for Reviewer zXWC's Feedback**
>
> Dear Reviewer zXWC,
>
> We sincerely appreciate your detailed review and constructive comments. Your feedback has helped us refine and strengthen our submission.
>
> As a brief reminder, we posted our rebuttal on November 19 (AoE) and added further experimental results on November 20 (AoE). Since the discussion window will close soon, at 11:59 pm AoE on December 2, we want to make sure that there is sufficient time to address any additional questions or concerns you might have before reviewer responses close. After the deadline, reviewers can no longer comment, and we will be unable to respond after 11:59 pm AoE on December 3.
>
> If there are any remaining points you would like us to clarify, we would be glad to provide further explanation while the discussion phase remains open. We will be happy to respond to any follow-up you may have.
>
> Thank you again for your time and helpful insights.
>
> Sincerely,
>
> Authors

---

### Official Review · Reviewer_q8Yj · 2025-11-01

**Soundness:** 3
**Presentation:** 2
**Contribution:** 2
**Rating:** 4
**Confidence:** 3

**Summary:**

This paper proposes a token-wise expert merging strategy of MoE, mainly to replace the conventional sparse routing for better performance. The paper adopted the Nash Bargaining Solution (NBS) to enhance the collaboration among experts during the expert merging process. The paper provides theoretical and empirical support to justify the claim.

**Strengths:**

1. The proposal of NBS in expert merging seems novel
2. The paper provides theoretical justifications behind the proposed method
3. The method improves performance over previous related expert-merging methods (e.g., SMEAR, CAMEx, EP-CAMEx)

**Weaknesses:**

1. The main advantage of the MoE-based models is their training efficiency. It has been established in the literature that MoE models generally achieve similar performance with significantly lower training FLOPs due to their sparse training. However, the proposed expert-merging method (and possibly previous methods also) doesn't employ the sparse routing. For example, in equation (7) of the paper, the update of the base expert uses all $N$ experts and their curvature matrices. Therefore, there is an uncertainty about whether the proposed expert-merging process sacrifices the training-efficiency advantage of MoE. As we can see, in Table 12 of the Appendix, the SMoE has lower training FLOPs than NAMEx. Therefore, it is uncertain whether the advantage of NAMEx appears for extra training FLOPs or from the delicate design.

2. Another advantage of SMoE is its capability of maintaining a constant inference FLOPs with the increase of the number of experts. It is not clear whether the advantage remains in the proposed expert-merging method.

**Questions:**

Can the authors clarify whether the empirical advantage of expert-merging arises from extra training FLOPs or from the proposed design? A training FLOPs equivalent result can be a good way to clarify that.

---

> ### Author Response · Authors · 2025-11-20
> **Response to Reviewer q8Yj (1)**
>
> Thank you for your thoughtful review and valuable feedback. Below we address your concerns.
>
> -----
>
> **[Weakness 1]. The main advantage of the MoE-based models is their training efficiency. It has been established in the literature that MoE models generally achieve similar performance with significantly lower training FLOPs due to their sparse training. However, the proposed expert-merging method (and possibly previous methods also) doesn't employ the sparse routing. For example, in equation (7) of the paper, the update of the base expert uses all experts and their curvature matrices. Therefore, there is an uncertainty about whether the proposed expert-merging process sacrifices the training-efficiency advantage of MoE. As we can see, in Table 12 of the Appendix, the SMoE has lower training FLOPs than NAMEx. Therefore, it is uncertain whether the advantage of NAMEx appears for extra training FLOPs or from the delicate design.**
>
> **[Question 1]. Can the authors clarify whether the empirical advantage of expert-merging arises from extra training FLOPs or from the proposed design? A training FLOPs equivalent result can be a good way to clarify that.**
>
> **Answer:** Thank you for raising this point. We would first like to clarify that the SMoE baseline shown in Table 12 of the original manuscript (Table 13, Appendix F.1 in the revised manuscript) uses top-1 routing. We used top-1 routing because NAMEx, CAMEx, and EP-CAMEx all operate with a single active expert (the merged expert). However, top-1 is not the standard setting in modern MoE systems. In practice, models use top-2 or top-K routing (K = 8, 16, …) as in GShard [1] and DeepSeek-MoE [2].
>
> To address your concern, we additionally compare against top-2 SMoE in Table 1, Table 2, and Table 3 below. We also included these results in Table 1–2 in Section 4 and Table 13 in Appendix F.1 of the revised manuscript.
>
> **Experimental Results Summary**
>
> Across GLUE tasks and WikiText-103 perplexity, NAMEx and NAMEx-Full consistently outperform CAMEx and EP-CAMEx, with NAMEx-Full-Mom achieving the best performance overall.
>
> Importantly:
>
> - **SMoE (top-2)** provides small accuracy gains but with **substantially higher compute**
> (train throughput drops **19.5%**, from 22,236 $\rightarrow$ 17,898 tok/s).
> - **NAMEx and NAMEx-Full** improve accuracy purely through expert-space merging, **without increasing routing cost or activating more experts per token**. NAMEx throughput decreases by only around **1.1%** relative to SMoE (top-1), and is **22.9%** faster than SMoE (top-2).
>
> **Table 1**. Training compute and throughput. Inference is unchanged relative to baselines.
> | Model                     | Train TFLOPs | Infer TFLOPs | Train Throughput (tok/s) |
> |---------------------------|--------------|---------------|----------------------------|
> | SMoE                      | 13.95        | 4.65          | 22,236                     |
> | **SMoE (Top-2)**        | 18.32        | 7.44          | 17,898                     |
> | SMEAR                      | 13.95        | 4.65          | 22,236                     |
> | CAMEx                     | 14.30        | 4.65          | 21,982                     |
> | EP-CAMEx                  | 14.25        | 4.65          | 21,982                     |
> | NAMEx                     | 14.25        | 4.65          | 21,995                     |
> | **NAMEx-Full**            | **14.25**    | **4.65**      | **21,897**                 |
> | EP-CAMEx-Mom              | 14.25        | 4.65          | 21,872                     |
> | **NAMEx-Full-Mom**        | **14.25**    | **4.65**      | **21,783**|

---

> ### Author Response · Authors · 2025-11-20
> **Response to Reviewer q8Yj (2)**
>
> **Table 2.** Validation and test perplexity on WikiText-103 for small- and medium-scale pretraining.
>
> | Model              | Params       | Small Val PPL                               | Small Test PPL                              | Medium Val PPL                              | Medium Test PPL                             |
> |--------------------|--------------|----------------------------------------------|----------------------------------------------|----------------------------------------------|----------------------------------------------|
> | SMoE (Top-1)     | 70M / 216M   | $86.64 \scriptscriptstyle{\pm .22}$          | $87.79 \scriptscriptstyle{\pm .31}$          | $38.60 \scriptscriptstyle{\pm .18}$          | $40.51 \scriptscriptstyle{\pm .25}$          |
> | SMoE (Top-2)       | 70M / 216M   | $84.26 \scriptscriptstyle{\pm .12}$          | $84.81 \scriptscriptstyle{\pm .29}$          | $\mathbf{33.76 \scriptscriptstyle{\pm .19}}$          | $\underline{35.55 \scriptscriptstyle{\pm .22}}$          |
> | NAMEx              | 70M / 216M   | $83.30 \scriptscriptstyle{\pm .21}$          | $84.12 \scriptscriptstyle{\pm .29}$          | $35.14 \scriptscriptstyle{\pm .19}$          | $36.40 \scriptscriptstyle{\pm .27}$          |
> | NAMEx-Full         | 70M / 216M   | $82.85 \scriptscriptstyle{\pm .17}$          | $\underline{83.16 \scriptscriptstyle{\pm .23}}$          | $34.92 \scriptscriptstyle{\pm .14}$          | $36.21 \scriptscriptstyle{\pm .20}$          |
> | NAMEx-Mom          | 70M / 216M   | $\underline{82.63 \scriptscriptstyle{\pm .15}}$ | $83.59 \scriptscriptstyle{\pm .21}$ | ${34.89 \scriptscriptstyle{\pm .12}}$ | ${35.86 \scriptscriptstyle{\pm .18}}$ |
> | NAMEx-Full-Mom     | 70M / 216M   | $\mathbf{82.44 \scriptscriptstyle{\pm .14}}$ | $\mathbf{82.94 \scriptscriptstyle{\pm .20}}$ | $\underline{34.25 \scriptscriptstyle{\pm .11}}$ | $\mathbf{35.37 \scriptscriptstyle{\pm .17}}$ |
>
> **Table 3.** GLUE results for T5-base SMoE variants (8 experts/layer).
>
> | Model               | Params | SST-2 | MRPC | CoLA | STSB | RTE | QNLI | MNLI |
> |---------------------|--------|-------|-------|-------|-------|-------|--------|--------|
> | SMoE (Top-1)        | 1.0B  | 94.26 | 90.87 | 56.78 | 89.44 | 70.75 | 92.07 | 86.38 |
> | **SMoE (Top-2)**    | 1.0B  | 94.35 | 91.04 | 58.43 | 89.73 | 74.98 | 92.48 | 86.72 |
> | NAMEx         | 1.0B | 94.46 | 92.01 | 58.81 | 90.12 | 75.09 | 92.86 | 86.96 |
> | **NAMEx-Full**| 1.0B | 94.82 | 92.80 | 59.63 | 90.27 | 77.83 | 93.23 | 87.23 |
> | NAMEx-Mom           | 1.0B | 94.61 | 93.02 | 58.90 | 90.06 | 77.62 | 93.11 | 87.02 |
> | **NAMEx-Full-Mom**  | 1.0B | **95.06** | **93.27** | **60.13** | **90.63** | **78.15** | **93.31** | **87.45** |
>
>
>
> **Why NAMEx Maintains MoE Efficiency**
>
> The key reason is that NAMEx preserves MoE’s token-level sparsity:
>
> - NAMEx does not activate additional experts per token.
> - Eqn. 7 operates purely in parameter space, aggregating expert parameters to update the shared base expert before the expert forward passes. An expert forward pass processes the tokens by a chosen expert. This contrasts with standard SMoE merging approaches that intervene in the feature space and require multiple expert forward passes.
> - The only extra computation introduced by NAMEx is solving a small bargaining problem of size $N \times N$, where $N$ (typically 8-16) is the number of experts. This step merely reweights the domain vectors $\tau_{i}^{(l)} = E_{i}^{(l)} - E_{m}^{(l)}$, where $i=1,\dots,N$ is the expert index and $l$ is the layer index (See Eqn. 7); it does not increase routing cost.
>
> To ensure a fair comparison, we matched compute budgets across methods. We cap the bargaining iterations at 2 convex-concave procedures (CCP) steps for NAMEx-Full and NAMEx-Full-Mom, so that their wall-clock time is close to that of EP-CAMEx. Under this configuration, NAMEx stays firmly in the sparse-MoE compute regime: 14.25 training TFLOPs, 4.65 inference TFLOPs, and roughly 21,995 tokens/second$-$comparable to CAMEx and EP-CAMEx and well below the compute footprint of denser routers. For example, SMoE with top-2 routing substantially increases compute (18.32 train TFLOPs, 7.44 infer TFLOPs) and reduces throughput to 17,898 tokens/second because it activates more experts per token. Since NAMEx never uses top-k routing with $k>1$, it avoids this overhead entirely.
>
> Regarding the concern about "using all experts and curvature matrices," this is handled cleanly by separating parameter-space merging from token-space execution. Although NAMEx reads all expert parameters to compute the base update, it does not execute all experts during the expert forward pass. NAMEx solving the small system $G^\top G \alpha = 1/\alpha$ with $G \in \mathbb{R}^{d \times N}$ to find the coefficients $\alpha_i$, which are used to update the base expert $\mathbf{E}_m$ (See Definition 3.3 in our manuscript). This computation is negligible with a small iteration budget, as shown in Table 1.

---

> ### Author Response · Authors · 2025-11-20
> **Response to Reviewer q8Yj (3)**
>
> Finally, NAMEx includes an efficiency knob through the update interval $\Delta \ell$, which controls how frequently we recompute bargaining weights. Infrequent updates yield substantial runtime gains (reducing wall-clock from 4.70s to 0.69s per training step) at only a minor cost to accuracy. Our main configuration adopts this efficient setting, and momentum variants further improve convergence stability with essentially no additional compute.
>
> **Conclusion**
>
> Overall, NAMEx’s improvements stem from its merging design rather than from additional FLOPs.
> - NAMEx maintains sparse MoE’s efficiency, staying at 14.25 / 4.65 TFLOPs and around 21.8–21.9k tok/s, matching SMoE (top-1).
> - SMoE (top-2) dramatically increases FLOPs and reduces throughput, unlike NAMEx.
>
> In the revised manuscript, we explicitly state that token-level sparsity is unchanged, and we detail the fixed bargaining budget and $\Delta \ell$ control in Table 12 of Appendix F.1.

---

> ### Author Response · Authors · 2025-11-20
> **Response to Reviewer q8Yj (4)**
>
> **[Weakness 2]. Another advantage of SMoE is its capability of maintaining a constant inference FLOPs with the increase of the number of experts. It is not clear whether the advantage remains in the proposed expert-merging method.**
>
> **Answer:** NAMEx preserves the core SMoE property that inference FLOPs remain constant even as the number of experts increases. Our approach performs expert merging only in parameter space during training or fine-tuning. At inference time, the model still routes each token to a single expert per layer, exactly the same behavior as a standard SMoE with top-1 routing. As a result, NAMEx does not alter routing sparsity or increase the number of experts used per token. All bargaining computations occur during training, not at inference. Empirically, Table 1 above (See [Weakness 1] and [Question 1]) shows that NAMEx matches the inference cost of SMoE and CAMEx (4.65 TFLOPs), whereas denser routing baselines such as SMoE Top-2 require significantly more compute (7.44 TFLOPs).
>
>
> **References**
>
> [1] Dmitry Lepikhin et al. "GShard: Scaling Giant Models with Conditional Computation and Automatic Sharding". (ICLR 2021)
>
> [2] Dai et al. "DeepSeekMoE: Towards Ultimate Expert Specialization in Mixture-of-Experts Language Models". (ACL 2024)
>
> -----
> We hope we have cleared your concerns about our work. We have also revised our manuscript according to your comments, and we would appreciate it if we could get your further feedback at your earliest convenience.

---

> > ### Author Response · Authors · 2025-11-28
> > **Additional Experimental Results: (1) Ablation on the Number of NBS Iterations and (2) Ablation on Routing Strategies and Disagreement Points**
> >
> > Dear Reviewer q8Yj,
> >
> > Thank you once again for your thoughtful reviews and valuable feedback.
> >
> > We have conducted two additional ablation studies on Qwen1.5-MoE, and the corresponding results are reported in Tables 4 and 5 below. These results have also been incorporated into Table 8, Section 5, and Table 17, Appendix F.4, of our revised manuscript, respectively.
> >
> > **1. Ablation on Number of NBS Iterations**
> >
> > Table 4 presents the effect of varying the number of iterations used to solve each NBS subproblem. The results confirm that using a small bargaining budget—specifically, 2 CCP iterations per layer—provides an excellent balance between accuracy and efficiency. Increasing the iteration count yields only marginal, statistically insignificant performance changes while substantially increasing the computational cost. These findings reinforce the practicality of our chosen configuration for large-scale, production-style training.
> >
> > **Table 4.** Performance of Qwen1.5-MoE variants comparison across the number of NBS solving iterations for the Linear router in NAMEx-Full config. All results are slightly improved while maintaining marginal gaps. Throughout the experiments in our manuscript, we use 2 CCP iterations per layer for NAMEx-Full (the chosen config in our table below).
> >
> > | Model       | No. Iterations   | Zero-Shot MMLU | Zero-Shot GSM8K | Zero-Shot ARC | SmolTalk MMLU | SmolTalk GSM8K | SmolTalk ARC |
> > |------------|------------------|----------------|------------------|---------------|----------------|-----------------|--------------|
> > | Qwen1.5-MoE | 2 (Chosen Config) | 61.87          | 60.55            | 50.95         | 62.10          | 61.00           | 51.35        |
> > | Qwen1.5-MoE | 5                | 61.70          | 60.55            | 50.94         | 62.20          | 61.05           | 51.31        |
> > | Qwen1.5-MoE | 20               | 61.94          | 60.57            | 50.96         | 62.14          | 61.03           | 51.34        |
> > | Qwen1.5-MoE | 40               | 61.92          | 60.62            | 51.01         | 62.22          | 61.05           | 51.46        |
> > | Qwen1.5-MoE | 60               | 61.81          | 60.48            | 51.08         | 62.15          | 60.98           | 51.39        |
> >
> > **2. Ablation on Routing Strategies and Disagreement Points**
> >
> > Table 5 compares NAMEx-Full across multiple routing strategies, including Linear, Cosine, and Stable-MoE, and two disagreement points: 0 and the mean-merge baseline. Across all benchmarks and routers, the performance differences are extremely small ( around 0.04 absolute) and show no consistent preference for either option. This indicates that the primary performance gains come from the bargaining-derived weights, not the choice of the disagreement point. We keep 0 as the default due to its simplicity, numerical stability, and unchanged computational cost.
> >
> > **Table 5.** Performance of Qwen1.5-MoE variants comparison across routing strategies and models on MMLU, GSM8K, and ARC benchmarks. Left: zero-shot results. Right: fine-tuned on SmolTalk, then zero-shot results.
> >
> > | Routing Strategy | Model                         | Zero-Shot MMLU | Zero-Shot GSM8K | Zero-Shot ARC | SmolTalk MMLU | SmolTalk GSM8K | SmolTalk ARC |
> > |------------------|-------------------------------|----------------|-----------------|---------------|---------------|----------------|--------------|
> > | Linear           | NAMEx-Full (0 disagreement point)   |  61.87        |  60.55         |  50.95       |  62.10**       | 61.00       |  51.35      |
> > |            | NAMEx-Full (mean disagreement point) |  61.78        |  60.57          |  51.23       |  61.67       |  61.04        |  51.25      |
> > | Cosine           | NAMEx-Full (0 disagreement point)  |  61.85        |  60.52         |  50.93       |  62.05       |  60.95        |  51.30      |
> > |            | NAMEx-Full (mean disagreement point) |  61.86         |  60.45         |  50.77       |  62.01      |  60.81         |  51.37      |
> > | Stable-MoE       | NAMEx-Full (0 disagreement point)   |  61.90        |  60.60         |  50.96       |  62.15       |  61.10       |  51.45      |
> > |        | NAMEx-Full (mean disagreement point) |  61.88        |  60.64        |  51.03       |  62.15       |  61.11        |  51.35      |
> >
> > We would be grateful to hear whether our rebuttal and new experiments fully address your concerns, or if there are any points you would like us to clarify while the discussion phase is still open. We are happy to continue the conversation or provide any additional details that may be helpful.
> >
> > If you feel that our rebuttal has satisfactorily resolved the issues raised in your review, we kindly ask you to consider whether an increased score would more accurately reflect your current evaluation.
> >
> > Thank you again for your time, thoughtful comments, and valuable engagement.

---

> ### Author Response · Authors · 2025-11-27
> **Reminder for Reviewer q8Yj's Feedback**
>
> Dear Reviewer q8Yj,
>
> We sincerely appreciate your detailed review and constructive comments. Your feedback has helped us refine and strengthen our submission.
>
> As a brief reminder, we posted our rebuttal on November 19 (AoE). Since the discussion window will close soon, at 11:59 pm AoE on December 2, we want to make sure that there is sufficient time to address any additional questions or concerns you might have before reviewer responses close. After the deadline, reviewers can no longer comment, and we will be unable to respond after 11:59 pm AoE on December 3.
>
> If there are any remaining points you would like us to clarify, we would be glad to provide further explanation while the discussion phase remains open. We will be happy to respond to any follow-up you may have.
>
> If our rebuttal has satisfactorily addressed the issues raised in your review, we would be grateful if you might consider whether increasing your score would better represent your updated view.
>
> Thank you again for your time and helpful insights.
>
> Sincerely,
>
> Authors

---

### Author Response · Authors · 2025-11-20
**Summary of Revision**

In addition to correcting typos and notation issues, we have revised the paper to incorporate all reviewers’ comments and suggestions. The major updates we made are listed below and highlighted in magenta in the revised manuscript.

1. We conducted compute-matched FLOPs/throughput experiments and added Top-2 MoE parameter counts and WikiText-103 results in Table 1.
2. We added Top-2 MoE performance results on the GLUE benchmark in Table 2.
3. In addition, we added SMEAR and Top-2 MoE training compute in Table 13 (Appendix F.1).
4. We added clarification that we tested the mean as the disagreement point and found negligible differences, so we retain 0 as the stable default, with the comparison reported in Table 4 of the revised manuscript.
5. We provided a clear ablation (Table 8) and corresponding analysis (Section 5) showing that NAMEx is largely insensitive to the “bargaining-budget” hyperparameter, with only marginal performance differences across NBS iteration counts.
6. We conducted an additional comparison with another vision MoE model, NAMEx vs. ACMoE [1], on ImageNet and included the results in Table 15 of Appendix F.4 in our revised manuscript.
7. We performed an ablation study on the number of NBS iterations and included the results in Table 8 in Section 5 of our revision.
8. We carried out an ablation on routing strategies and disagreement points, with results included in Table 17 in Appendix F.4 of the revised manuscript.

**References**

[1] Stefan Nielsen et al. "Tight clusters make specialized experts." (ICLR 2025)

---

### Author Response · Authors · 2025-11-20
**General Response (1)**

Dear AC and reviewers,

Thanks for your thoughtful reviews and valuable comments, which have helped us improve the paper significantly. We are encouraged by the endorsements that: **(1)** our game-theoretic framework to handle the complex dynamics of expert merging problem and our solution through Nash Bargaining Solution (NBS) is novel (Reviewer q8Yj, zXWC, 2Qyz, kguY) and principled (Reviewer 2Qyz, eQrG); **(2)** Our incorporation of momentum, specifically complex momentum, improves both the convergence speed and the stability of the expert-merging process (Reviewer 2Qyz, eQrG); **(3)** The proposed framework is supported by  theoretical justifications for future analysis (Reviewer q8Yj, 2Qyz); **(4)** The empirical methodology and evidence are comprehensive and thorough, demonstrating consistent performance gains over previous related expert-merging methods (Reviewer q8Yj, zXWC), across multiple modalities and benchmark (Reviewer eQrG), for small-, medium-, and large-scale models , which confirm both the effectiveness and the scalability of the proposed method (Reviewer 2Qyz). We have updated our submission based on the reviewers' feedback, and **we have highlighted our revision in magenta**.

The most common concern towards our framework is the computational overhead of the Nash Bargaining problem, the associated speed–accuracy trade-offs of our proposed models, their impact on training efficiency, and whether the empirical gains stem from additional training FLOPs. We have provided detailed explanations to resolve this concern in each reviewer-specific rebuttal and added further theoretical complexity analysis and empirical evaluations in the revised manuscript. We also summarize our response below.

First, we would like to clarify that the SMoE baseline shown in Table 12 of the original manuscript (Table 13, Appendix F.1 in the revised manuscript) uses top-1 routing. We used top-1 routing because NAMEx, CAMEx, and EP-CAMEx all operate with a single active expert (the merged expert). However, top-1 is not the standard setting in modern MoE systems. In practice, models use top-2 or top-K routing (K = 8, 16, …) as in GShard [1] and DeepSeek-MoE [2].

To address your concern, we additionally compare against top-2 SMoE in Table 1, Table 2, and Table 3 below. We also included these results in Table 1–2 in Section 4 and Table 13 in Appendix F.1 of the revised manuscript.

**Experimental Results Summary**

Across GLUE tasks and WikiText-103 perplexity, NAMEx and NAMEx-Full consistently outperform CAMEx and EP-CAMEx, with NAMEx-Full-Mom achieving the best performance overall.

Importantly:

- **SMoE (top-2)** provides small accuracy gains but with **substantially higher compute**
(train throughput drops **19.5%**, from 22,236 $\rightarrow$ 17,898 tok/s).
- **NAMEx and NAMEx-Full** improve accuracy purely through expert-space merging, **without increasing routing cost or activating more experts per token**. NAMEx throughput decreases by only around **1.1%** relative to SMoE (top-1), and is **22.9%** faster than SMoE (top-2).

**Table 1**. Training compute and throughput. Inference is unchanged relative to baselines.
| Model                     | Train TFLOPs | Infer TFLOPs | Train Throughput (tok/s) |
|---------------------------|--------------|---------------|----------------------------|
| SMoE                      | 13.95        | 4.65          | 22,236                     |
| **SMoE (Top-2)**        | 18.32        | 7.44          | 17,898                     |
| SMEAR                      | 13.95        | 4.65          | 22,236                     |
| CAMEx                     | 14.30        | 4.65          | 21,982                     |
| EP-CAMEx                  | 14.25        | 4.65          | 21,982                     |
| NAMEx                     | 14.25        | 4.65          | 21,995                     |
| **NAMEx-Full**            | **14.25**    | **4.65**      | **21,897**                 |
| EP-CAMEx-Mom              | 14.25        | 4.65          | 21,872                     |
| **NAMEx-Full-Mom**        | **14.25**    | **4.65**      | **21,783**|

---

> ### Author Response · Authors · 2025-11-20
> **General Response (2)**
>
> **Table 2.** Validation and test perplexity on WikiText-103 for small- and medium-scale pretraining.
>
> | Model              | Params       | Small Val PPL                               | Small Test PPL                              | Medium Val PPL                              | Medium Test PPL                             |
> |--------------------|--------------|----------------------------------------------|----------------------------------------------|----------------------------------------------|----------------------------------------------|
> | SMoE (Top-1)     | 70M / 216M   | $86.64 \scriptscriptstyle{\pm .22}$          | $87.79 \scriptscriptstyle{\pm .31}$          | $38.60 \scriptscriptstyle{\pm .18}$          | $40.51 \scriptscriptstyle{\pm .25}$          |
> | SMoE (Top-2)       | 70M / 216M   | $84.26 \scriptscriptstyle{\pm .12}$          | $84.81 \scriptscriptstyle{\pm .29}$          | $\mathbf{33.76 \scriptscriptstyle{\pm .19}}$          | $\underline{35.55 \scriptscriptstyle{\pm .22}}$          |
> | NAMEx              | 70M / 216M   | $83.30 \scriptscriptstyle{\pm .21}$          | $84.12 \scriptscriptstyle{\pm .29}$          | $35.14 \scriptscriptstyle{\pm .19}$          | $36.40 \scriptscriptstyle{\pm .27}$          |
> | NAMEx-Full         | 70M / 216M   | $82.85 \scriptscriptstyle{\pm .17}$          | $\underline{83.16 \scriptscriptstyle{\pm .23}}$          | $34.92 \scriptscriptstyle{\pm .14}$          | $36.21 \scriptscriptstyle{\pm .20}$          |
> | NAMEx-Mom          | 70M / 216M   | $\underline{82.63 \scriptscriptstyle{\pm .15}}$ | $83.59 \scriptscriptstyle{\pm .21}$ | ${34.89 \scriptscriptstyle{\pm .12}}$ | ${35.86 \scriptscriptstyle{\pm .18}}$ |
> | NAMEx-Full-Mom     | 70M / 216M   | $\mathbf{82.44 \scriptscriptstyle{\pm .14}}$ | $\mathbf{82.94 \scriptscriptstyle{\pm .20}}$ | $\underline{34.25 \scriptscriptstyle{\pm .11}}$ | $\mathbf{35.37 \scriptscriptstyle{\pm .17}}$ |
>
> **Table 3.** GLUE results for T5-base SMoE variants (8 experts/layer).
>
> | Model               | Params | SST-2 | MRPC | CoLA | STSB | RTE | QNLI | MNLI |
> |---------------------|--------|-------|-------|-------|-------|-------|--------|--------|
> | SMoE (Top-1)        | 1.0B  | 94.26 | 90.87 | 56.78 | 89.44 | 70.75 | 92.07 | 86.38 |
> | **SMoE (Top-2)**    | 1.0B  | 94.35 | 91.04 | 58.43 | 89.73 | 74.98 | 92.48 | 86.72 |
> | NAMEx         | 1.0B | 94.46 | 92.01 | 58.81 | 90.12 | 75.09 | 92.86 | 86.96 |
> | **NAMEx-Full**| 1.0B | 94.82 | 92.80 | 59.63 | 90.27 | 77.83 | 93.23 | 87.23 |
> | NAMEx-Mom           | 1.0B | 94.61 | 93.02 | 58.90 | 90.06 | 77.62 | 93.11 | 87.02 |
> | **NAMEx-Full-Mom**  | 1.0B | **95.06** | **93.27** | **60.13** | **90.63** | **78.15** | **93.31** | **87.45** |
>
> **Why NAMEx Maintains MoE Efficiency**
>
> The key reason is that NAMEx preserves MoE’s token-level sparsity:
>
> - NAMEx does not activate additional experts per token.
> - Eqn. 7 operates purely in parameter space, aggregating expert parameters to update the shared base expert before the expert forward passes. An expert forward pass processes the tokens by a chosen expert. This contrasts with standard SMoE merging approaches that intervene in the feature space and require multiple expert forward passes.
> - The only extra computation introduced by NAMEx is solving a small bargaining problem of size $N \times N$, where $N$ (typically 8-16) is the number of experts. This step merely reweights the domain vectors $\tau_{i}^{(l)} = E_{i}^{(l)} - E_{m}^{(l)}$, where $i=1,\dots,N$ is the expert index and $l$ is the layer index (See Eqn. 7); it does not increase routing cost.
>
> To ensure a fair comparison, we matched compute budgets across methods. We cap the bargaining iterations at 2 convex-concave procedures (CCP) steps for NAMEx-Full and NAMEx-Full-Mom, so that their wall-clock time is close to that of EP-CAMEx. Under this configuration, NAMEx stays firmly in the sparse-MoE compute regime: 14.25 training TFLOPs, 4.65 inference TFLOPs, and roughly 21,995 tokens/second$-$comparable to CAMEx and EP-CAMEx and well below the compute footprint of denser routers. For example, SMoE with top-2 routing substantially increases compute (18.32 train TFLOPs, 7.44 infer TFLOPs) and reduces throughput to 17,898 tokens/second because it activates more experts per token. Since NAMEx never uses top-k routing with $k>1$, it avoids this overhead entirely.
>
> Furthermore, although NAMEx reads all expert parameters to compute the base update, it does not execute all experts during the expert forward pass. NAMEx solving the small system $G^\top G \alpha = 1/\alpha$ with $G \in \mathbb{R}^{d \times N}$ to find the coefficients $\alpha_i$, which are used to update the base expert $\mathbf{E}_m$ (see Definition 3.3 in our manuscript). This computation is negligible with a small iteration budget, as shown in Table 1.

---

> ### Author Response · Authors · 2025-11-20
> **General Response (3)**
>
> Finally, NAMEx includes an efficiency knob through the update interval $\Delta \ell$, which controls how frequently we recompute bargaining weights. Infrequent updates yield substantial runtime gains (reducing wall-clock from 4.70s to 0.69s per training step) at only a minor cost to accuracy (see Table 12 in Appendix F.1 of our revised manuscript). Our main configuration adopts this efficient setting, and momentum variants further improve convergence stability with essentially no additional compute.
>
> **Conclusion**
>
> Overall, NAMEx’s improvements stem from its merging design rather than from additional FLOPs.
> - NAMEx maintains sparse MoE’s efficiency, staying at 14.25 / 4.65 TFLOPs and around 21.8–21.9k tok/s, matching SMoE (top-1).
> - SMoE (top-2) dramatically increases FLOPs and reduces throughput, unlike NAMEx.
>
> In the revised manuscript, we explicitly state that token-level sparsity is unchanged, and we detail the fixed bargaining budget and $\Delta \ell$ control in Table 12 of Appendix F.1.
>
> **References**
>
> [1] Dmitry Lepikhin et al. "GShard: Scaling Giant Models with Conditional Computation and Automatic Sharding". (ICLR 2021)
>
> [2] Dai et al. "DeepSeekMoE: Towards Ultimate Expert Specialization in Mixture-of-Experts Language Models". (ACL 2024)

---

### Author Response · Authors · 2025-11-24
**Final Rebuttal Submitted and Follow-Up Inquiry**

Dear Reviewers and Chairs,

We would like to express our sincere gratitude for the thoughtful feedback and the time you have dedicated to reviewing our submission. We are also thankful for the substantial work the chairs have carried out in coordinating the review process and fostering productive discussions.

Our finalized rebuttal has now been uploaded. This updated document includes all intended modifications as well as the additional experimental results we prepared, and these changes are outlined in the "Summary of Revision" for easy reference.

We would appreciate your guidance on whether our responses satisfactorily address the concerns raised, or whether there are any further points you would like us to elaborate on.

We remain fully available for continued discussions and are happy to provide any further clarification that may assist the decision process.

With kind regards,

Authors

---

### Author Response · Authors · 2025-11-28
**Additional Experimental Results: (1) Ablation on the Number of NBS Iterations and (2) Ablation on Routing Strategies and Disagreement Points**

Dear Reviewers,

Thank you once again for your thoughtful reviews and valuable feedback.

We have conducted two additional ablation studies on Qwen1.5-MoE, and the corresponding results are reported in Tables 1 and 2 below. These results have also been incorporated into Table 8, Section 5, and Table 17, Appendix F.4, of our revised manuscript, respectively.

**1. Ablation on Number of NBS Iterations**

Table 1 presents the effect of varying the number of iterations used to solve each NBS subproblem. The results confirm that using a small bargaining budget—specifically, 2 CCP iterations per layer—provides an excellent balance between accuracy and efficiency. Increasing the iteration count yields only marginal, statistically insignificant performance changes while substantially increasing the computational cost. These findings reinforce the practicality of our chosen configuration for large-scale, production-style training.

**Table 1.** Performance of Qwen1.5-MoE variants comparison across the number of NBS solving iterations for the Linear router in NAMEx-Full config. All results are slightly improved while maintaining marginal gaps. Throughout the experiments in our manuscript, we use 2 CCP iterations per layer for NAMEx-Full (the chosen config in our table below).

| Model       | No. Iterations   | Zero-Shot MMLU | Zero-Shot GSM8K | Zero-Shot ARC | SmolTalk MMLU | SmolTalk GSM8K | SmolTalk ARC |
|------------|------------------|----------------|------------------|---------------|----------------|-----------------|--------------|
| Qwen1.5-MoE | 2 (Chosen Config) | 61.87          | 60.55            | 50.95         | 62.10          | 61.00           | 51.35        |
| Qwen1.5-MoE | 5                | 61.70          | 60.55            | 50.94         | 62.20          | 61.05           | 51.31        |
| Qwen1.5-MoE | 20               | 61.94          | 60.57            | 50.96         | 62.14          | 61.03           | 51.34        |
| Qwen1.5-MoE | 40               | 61.92          | 60.62            | 51.01         | 62.22          | 61.05           | 51.46        |
| Qwen1.5-MoE | 60               | 61.81          | 60.48            | 51.08         | 62.15          | 60.98           | 51.39        |

**2. Ablation on Routing Strategies and Disagreement Points**

Table 2 compares NAMEx-Full across multiple routing strategies, including Linear, Cosine, and Stable-MoE, and two disagreement points: 0 and the mean-merge baseline. Across all benchmarks and routers, the performance differences are extremely small ( around 0.04 absolute) and show no consistent preference for either option. This indicates that the primary performance gains come from the bargaining-derived weights, not the choice of the disagreement point. We keep 0 as the default due to its simplicity, numerical stability, and unchanged computational cost.

**Table 2.** Performance of Qwen1.5-MoE variants comparison across routing strategies and models on MMLU, GSM8K, and ARC benchmarks. Left: zero-shot results. Right: fine-tuned on SmolTalk, then zero-shot results.

| Routing Strategy | Model                         | Zero-Shot MMLU | Zero-Shot GSM8K | Zero-Shot ARC | SmolTalk MMLU | SmolTalk GSM8K | SmolTalk ARC |
|------------------|-------------------------------|----------------|-----------------|---------------|---------------|----------------|--------------|
| Linear           | NAMEx-Full (0 disagreement point)   |  61.87        |  60.55         |  50.95       |  62.10**       | 61.00       |  51.35      |
|            | NAMEx-Full (mean disagreement point) |  61.78        |  60.57          |  51.23       |  61.67       |  61.04        |  51.25      |
| Cosine           | NAMEx-Full (0 disagreement point)  |  61.85        |  60.52         |  50.93       |  62.05       |  60.95        |  51.30      |
|            | NAMEx-Full (mean disagreement point) |  61.86         |  60.45         |  50.77       |  62.01      |  60.81         |  51.37      |
| Stable-MoE       | NAMEx-Full (0 disagreement point)   |  61.90        |  60.60         |  50.96       |  62.15       |  61.10       |  51.45      |
|        | NAMEx-Full (mean disagreement point) |  61.88        |  60.64        |  51.03       |  62.15       |  61.11        |  51.35      |

We would be grateful to hear whether our rebuttal and new experiments fully address your concerns, or if there are any points you would like us to clarify while the discussion phase is still open. We are happy to continue the conversation or provide any additional details that may be helpful.

If you feel that our rebuttal has satisfactorily resolved the issues raised in your review, we kindly ask you to consider whether an increased score would more accurately reflect your current evaluation.

Thank you again for your time, thoughtful comments, and valuable engagement.

---

### Author Response · Authors · 2025-12-01
**Note 1 for the New AC: Highlights of Our Key Contributions**

# Novelty and Contributions

**1) Game-theoretic view of expert merging**

We introduce Nash Merging of Experts (NAMEx) by reinterpreting expert merging in Sparse Mixture-of-Experts (SMoE) as a cooperative–competitive bargaining game between experts and formalizing the *Bargaining of Expert Merging Problem* in which expert domain vectors serve as utilities. The Nash Bargaining Solution (NBS) is then used to derive principled, data-driven merging coefficients instead of heuristic or averaging schemes.

**2) Complex-momentum expert propagation with principles**

We further propose to incorporate *complex momentum* into the propagation of the base expert across layers to speed up and stabilize convergence of the mixed cooperative-competitive expert dynamic, together with a convergence guarantee (Theorem B.3 in the paper).

**3) Quaternion momentum as a richer dynamical extension**

Motivated by the rotational dynamics arising in our game-theoretic formulation, we additionally explore quaternion momentum as a higher-dimensional extension of complex momentum for expert propagation. Our experiments indicate that quaternion momentum can further stabilize the bargaining dynamics in challenging multi-domain settings, and our analysis of complex momentum provides a roadmap for extending similar convergence guarantees. We view this as a promising direction for further strengthening expert merging in future work.

# Practical Advantages of Our NAMEx

**Broad empirical validation and scalability**

We evaluate NAMEx variants across language modeling (WikiText-103), GLUE tasks, text classification, ImageNet-1k and its corrupted variants, as well as large-scale SMoEs like Qwen1.5-MoE (14B) and DeepSeek-MoE (16B). NAMEx consistently outperforms prior expert-merging baselines (SMEAR [1], CAMEx [2], EP-CAMEx [2]) and scales effectively from small to large models.

**Plug-in for existing SMoE systems**

NAMEx operates purely in parameter space and preserves token-level sparsity and routing (one active expert per token), and empirically, we show that training and inference FLOPs remain in the sparse-MoE regime. In practice, NAMEx can be dropped into existing SMoE architectures such as DeepSeek-MoE, and Qwen1.5-MoE by adding only the expert-merging step, without modifying the architecture, changing the router, or retraining the model from scratch. This makes NAMEx easy to implement and practical as an off-the-shelf module for production MoE systems.

---

### References

[1] Muqeeth, M., et al. "Soft merging of experts with adaptive routing". (TMLR 2024)

[2] Nguyen, Dung V., et al. "Camex: Curvature-aware merging of experts". (ICLR 2025)

---

### Author Response · Authors · 2025-12-01
**Note 2 for the New AC: Highlights of Reviewer Concerns and Our Responses (1)**

We are encouraged by the reviewers’ positive assessments, including:

**(i)** recognition that our game-theoretic formulation of the expert-merging problem and our use of the Nash Bargaining Solution (NBS) are novel (Reviewers q8Yj, zXWC, 2Qyz, kguY) and principled (Reviewers 2Qyz, eQrG);

**(ii)** acknowledgement that introducing momentum, particularly complex momentum, enhances both the convergence speed and stability of the merging dynamics (Reviewers 2Qyz, eQrG);

**(iii)** appreciation that our framework is grounded in theoretical analysis and opens the door for future extensions (Reviewers q8Yj, 2Qyz);

**(iv)** recognition that our empirical study is comprehensive and demonstrates consistent improvements over prior expert-merging approaches (Reviewers q8Yj, zXWC), across multiple modalities and benchmarks (Reviewer eQrG) and at small, medium, and large model scales, confirming both the effectiveness and scalability of our method (Reviewer 2Qyz).

We have revised the submission in line with the reviewers’ comments, with all changes highlighted in magenta.

Below, we summarize the reviewers’ primary concerns along with our corresponding responses.

**(1) Nature of contribution and role of Nash bargaining**

- *Reviewer's Concern.* Several reviewers (eQrG, 2Qyz) felt the work mainly combines EP-CAMEx [1] with Nash bargaining / NashMTL [4], and questioned whether this is more of a strong application than a fundamentally new theoretical contribution.

- *Our Response.* We do not claim a new theory of Nash bargaining. Our contribution is to
    - (i) recast expert merging in sparse MoEs as a cooperative bargaining game, and
    - (ii) design a practical merging algorithm (NAMEx / NAMEx-Full / NAMEx-Full-Mom) that directly targets cross-domain interference, preserves sparse routing, and admits convergence guarantees for the propagation step. Existing expert-merging methods (TIES [6], DARE [7], SMEAR [3], CAMEx/EP-CAMEx [1]) are largely heuristic and do **not** provide such a principled bargaining formulation.


**(2) Computation, runtime, and MoE efficiency (training & inference)**

- *Concern.* Reviewers (eQrG, q8Yj, kguY) worried that solving the Nash Bargaining Solution introduces overhead, that NAMEx seems more expensive than SMoE, and asked whether gains are driven mainly by extra compute. They also asked if constant inference FLOPs as the number of experts grows still holds.

- *Response.* NBS is solved on a small $N \times N$ system (where $N$ is the number of experts) with a very small CCP iteration budget (2 iterations per layer in all main experiments) and is used only during training/merging. Routing remains top-1, so token-level sparsity and constant inference FLOPs are preserved. New compute-matched experiments (**Tables 1, 2, 8, 13 in the revised manuscript**) show that training/inference TFLOPs and throughput of NAMEx/NAMEx-Full-Mom are very close to EP-CAMEx and SMoE top-1, while SMoE top-2 is substantially more expensive (e.g., ~19.5% throughput drop) for only modest accuracy gains. This indicates that improvements stem from the *algorithmic design*, not simply more computing.

**(3) Bargaining budget (NBS iterations) and its allocation across layers**

- *Reviewer's Concern.* There was confusion about the “bargaining budget” (e.g., “20 steps” vs “2 iterations”), how it is distributed across layers, and how sensitive performance is to this budget (eQrG, zXWC).

- *Our Response.* We explicitly define the bargaining budget as the **total number of NBS (CCP) iterations** and distinguish two schemes:

    * **NAMEx:** spends the entire budget at the **first layer** and propagates the merged expert to deeper layers.
    * **NAMEx-Full:** **spreads the same budget across all layers** (e.g., 2 iterations per layer for 10 layers if the total budget is 20).

   An ablation over iterations per layer (**Table 8, Section 5**) shows that increasing the budget (e.g., 2 $\rightarrow$ 5 $\rightarrow$ 20+ iterations) yields only marginal accuracy gains but significantly increases training time, justifying our default choice of 2 iterations per layer as a practical trade-off.

---

> ### Author Response · Authors · 2025-12-01
> **Note 2 for the New AC: Highlights of Reviewer Concerns and Our Responses (2)**
>
> **(4) Utility function and disagreement point**
>
> - *Reviewer's Concern.* Reviewers (eQrG, kguY) found the utility definition and disagreement point somewhat heuristic, and asked why the disagreement point is set to 0 (no update) instead of a baseline such as mean merging or performance on held-out data.
>
> - *Our Response.* We score **update directions** rather than parameters:
> $$
> u_i(\Delta\theta) = \tau_i^\top \Delta\theta,
> $$
> where $\tau_i$ is a domain vector for expert $E_i$, constructed from curvature-aware statistics/gradients on that expert’s domain. This dot product measures how well a candidate update aligns with each expert’s “preference”, and the Nash solution produces a Pareto-efficient compromise. Following standard Nash bargaining setups and NashMTL [4], we take $0$ (no update) as the disagreement point. We also tested the **mean update direction** as disagreement; **Table 4, Section 5** and **Table 17, Appendix F.4** show negligible differences, indicating that our framework is robust to this choice, so we keep 0 as the clean default.
>
>
> **(5) Role of curvature in propagation and benefit of complex momentum**
>
> - *Reviewer's Concern.* Reviewer 2Qyz asked why the curvature matrix is removed from the propagation update (relative to EP-CAMEx [1]), and kguY requested a clearer explanation of complex momentum, how it differs from standard momentum/Adam, and why it helps.
>
> - *Our Response.* The curvature term is not dropped because of Nash bargaining but **due to the convergence analysis of our complex-momentum scheme**: Theorem B.3 shows that including curvature in the propagation step prevents us from establishing convergence guarantees, even for contractive operators. Removing curvature from the explicit propagation update lets us **prove convergence** while still using curvature information elsewhere (e.g., in domain vectors/preconditioning). Complex momentum introduces a complex coefficient ($\beta = \beta_1 + i\beta_2$), corresponding to two coupled real buffers that capture oscillatory, second-order-like behavior. This stabilizes bargaining updates when expert gradients conflict. Ablations show that complex momentum converges **faster and more stably** than real-valued momentum under the same iteration budget.
>
>
> **(6) Architecture design (Eq. (9)) and experimental coverage, especially in vision MoEs**
>
> - *Reviewer's Concern.* Reviewer zXWC asked why NBS is applied only in the first line of Eq. (9) and not the second, and requested stronger vision MoE baselines on ImageNet to demonstrate the generality of NAMEx.
>
> - *Our Response.* The first line of Eq. (9) produces a neutral, context-agnostic merged expert via NBS, which aggregates information from all experts and is propagated across layers as a shared representation. The second line produces a context-adaptive merged expert, analogous to adaptive experts in CAMEx [1], SMEAR [3], MEO [2], Lory [5], etc., that emphasizes token-relevant information. On the experimental side, we add comparisons to SMoE top-2 [2] and ACMoE [8] on ImageNet-1k and robustness variants (INet-O/A/R). Under comparable parameters and FLOP budgets, NAMEx-Full and NAMEx-Full-Mom achieve the best Acc@1/Acc@5 and the strongest OOD robustness, indicating that NAMEx improves both accuracy and generalization in vision MoEs.
>
> ---
>
> ### References
>
> [1] Nguyen, D. V., et al. “CAMEx: Curvature-aware merging of experts.” ICLR 2025.
>
> [2] He, S., et al. “Merging experts into one: Improving computational efficiency of mixture of experts.” EMNLP 2023.
>
> [3] Muqeeth, M., et al. “Soft merging of experts with adaptive routing.” TMLR 2024.
>
> [4] Shamsian, A., et al. “Auxiliary learning as an asymmetric bargaining game.” ICML 2023.
>
> [5] Zhong, Z., et al. “Lory: Fully Differentiable Mixture-of-Experts for Autoregressive Language Model Pre-training.” COLM 2024.
>
> [6] Yadav, P., et al. “TIES-Merging: Resolving interference when merging models.” NeurIPS 2023.
>
> [7] Yu, L., et al. “Language models are Super Mario: Absorbing abilities from homologous models as a free lunch.” ICML 2024.
>
> [8] Nielsen, S. K., et al. “Tight Clusters Make Specialized Experts.” ICLR 2025.

---

### Author Response · Authors · 2025-12-01
**Note 3 for the New AC: Highlights of Additional Results during the Rebuttal & Discussion Phase**

# New analysis

* **Compute and sparsity analysis.** In Table 13, Appendix F.1 of the revised manuscript, we analyzed the compute profile of NAMEx, showing it maintains sparse MoE efficiency: training at 14.25 TFLOPs and inference at 4.65 TFLOPs with ≈21.8–22.0k tokens/s, comparable to CAMEx/EP-CAMEx and SMoE (top-1), while SMoE (top-2) increases training TFLOPs to 18.32, inference to 7.44, and reduces throughput to 17,898 tokens/s.

* **Formal statement that NAMEx preserves inference FLOPs.** We emphasized in our responses that expert merging happens only in parameter space during training/fine-tuning; at inference, routing remains top-1 and uses exactly one expert per token, so inference FLOPs remain constant as the number of experts grows.

* **Convergence-motivated removal of curvature matrix.** We explained that omitting the curvature matrix $M_i$ from the NAMEx propagation update is mandated by the convergence proof of complex momentum; keeping $M_i$ breaks the guarantee, even under contractive operators.

* **Future extension: generalized NBS with context.** We outlined a generalized Nash bargaining objective with context-dependent preference weights derived from router scores as a principled route to make the adaptive expert $E_m$ also Nash-based in future work.

# New empirical results/ablations

* **Compute-matched comparison with SMoE and other merging baselines.** Added Table 1 and 2 in Section 4 show that NAMEx and NAMEx-Full achieve better GLUE and WikiText-103 performance than CAMEx/EP-CAMEx and SMoE (top-1), while remaining much more efficient than SMoE (top-2). NAMEx-Full-Mom achieves the best overall compute-performance trade-off.

* **Ablation on bargaining iterations for DeepSeek-MoE & Qwen1.5-MoE.** In Table 8, Section 5 of the revised manuscript: For **DeepSeek-MoE**, we varied solving iterations (2, 5, 20, 40, 60) in NAMEx-Full and observed only tiny fluctuations (≈0.1–0.3 absolute) in MMLU, GSM8K, and ARC for both zero-shot and SmolTalk-fine-tuned settings; 2 iterations already match or slightly exceed larger budgets. For **Qwen1.5-MoE**, we repeated this ablation and again found that increasing the number of CCP steps beyond 2 yields only marginal, statistically insignificant changes while increasing compute, confirming that our chosen 2-iteration budget is near-optimal.

* **Ablation on routing strategies and disagreement points.** In **Table 4 Section 5** and **Table 17 Appendix F.4** of the revised manuscript, we compared NAMEx-Full with three routers (Linear, Cosine, Stable-MoE) and two disagreement points ($0$ vs mean-merge). Across all metrics (MMLU, GSM8K, ARC), both in zero-shot and SmolTalk-fine-tuned settings, performance differences are extremely small (~0.04 absolute) and show no consistent preference, indicating that most gains come from NBS-derived weights rather than a particular router or disagreement point.

* **Additional vision MoE experiment on ImageNet (ACMoE).** In **Table 15 Appendix F.4**, we trained NAMEx variants on top of the ACMoE architecture and reported pretraining and zero-shot robustness results on ImageNet-1k and corrupted variants. NAMEx, NAMEx-Full, and especially NAMEx-Full-Mom consistently outperform ACMoE Top-1 and Top-2 in Acc@1/Acc@5 and in all OOD metrics (INet-O, INet-A, INet-R), showing better robustness to corruptions and distribution shifts at equal parameter budget.

---

### Author Response · Authors · 2025-12-01
**About Notes for the New AC: Thank You for Taking Responsibility for Our Submission**

Dear new AC,

Thank you very much for stepping in at this stage of the review process and taking over the handling of our submission. We sincerely appreciate your time and support.

To help you quickly navigate our work and the review history, we have prepared three short notes, included in the messages below:

- **Note 3: Highlights of Additional Results during the Rebuttal & Discussion Phase**
- **Note 2: Highlights of Reviewer Concerns and Our Responses**
- **Note 1: Highlights of Our Key Contributions**

We submitted our initial rebuttal on November 18 (AoE), followed by additional experimental results on November 22 (AoE). Despite several reminders, the reviewers have not responded to our rebuttal, and the discussion phase has concluded early. As a result, we were unable to obtain further clarification or updated feedback from them.

We kindly ask that the AC reviews our paper, rebuttal, and follow-up responses (including the additional results), and takes into account that we have addressed all of the reviewers’ concerns thoroughly. We believe that, had the discussion continued, the reviewers would likely have updated their assessments.

Thank you once more for taking on the oversight of our submission. Please let us know if you have any questions regarding our submission or the rebuttal. We would be very happy to clarify or discuss further.

Best regards,

The Authors

---

### Meta-Review · Area_Chair_DwUN · 2026-01-07

**Summary:**

This paper proposes a principled framework, named NAMEx, for reinterpreting expert merging in sparse mixture-of-experts models by incorporating Nash Bargaining into the merging process. The paper adopted the Nash Bargaining Solution (NBS) to enhance the collaboration among experts during the expert merging process. The paper provides theoretical and empirical support to justify its claim, providing a principled alternative to prior heuristic methods. The approach is conceptually well-motivated and provides a clear alternative to prior heuristic-based merging methods, such as CAMEx and EP-CAMEx. The work further introduces complex momentum to stabilize and accelerate the convergence of the propagation step, supported by a theoretical guarantee.

Reviewers acknowledged the contribution of the novel game-theoretic framework, i.e., the introduction of game theory (Nash Bargaining) to expert merging, the convergence via complex momentum, and extensive experimental validations with strong results.

During the rebuttal, the authors have provided substantial clarification and experimental details, including computation and throughput analysis, ablations on the Nash bargaining iteration budget, and new vision MoE results on ImageNet and its corrupted variants. These additions satisfactorily resolve the reviewers’ main concerns regarding the details of training & inference (computation, runtime, and MoE efficiency), Bargaining budget (NBS iterations) with its allocation across layers, the utility function+disagreement point, etc.

In summary, this paper presents a promising insight for expert merging in sparse MoE models. The remaining concerns are minor and do not undermine the main claims. Hence, I recommend the acceptance of this paper.

**Reviewer Concerns:**

The addressed major concerns.

+ Questions regarding computational overhead and MoE efficiency were resolved through compute-matched experiments. These experiments showed that NAMEx preserves top-1 routing with constant inference FLOPs, and has training throughput comparable to EP-CAMEx and SMoE top-1.

+ Confusion around the “bargaining budget” was clarified by explicitly distinguishing CCP iterations from bargaining steps, and extensive ablations on DeepSeek-MoE and Qwen1.5-MoE demonstrated that performance is largely insensitive to this hyperparameter, with two iterations already near-optimal.

+ Concerns about the heuristic nature of the utility function and disagreement point were addressed via additional ablations showing negligible differences between zero and mean-merge disagreement points across routers and tasks.

+ The broad empirical validation and scalability were met by adding additional Vision MoE results on ImageNet and robustness benchmarks using ACMoE.

+  The authors provide a convergence-based justification to address the theoretical concerns regarding the removal of the curvature matrix and the role of complex momentum.

**Reviewer Scores:**

+ Reviewer q8Yj (original score 4) would like to increase his/her score, as their main concerns about extra training FLOPs, loss of MoE sparsity, and inference efficiency were directly addressed through compute-matched experiments and explicit confirmation of constant inference FLOPs.
+ Reviewer zXWC (original score 8) would like to keep his/her score.
+ Reviewer 2Qyz (original score 10) would like to keep his/her score.
+ Reviewer eQrG (original score 6) would like to increase his/her score, as concerns about computational overhead, bargaining-budget clarity, and limited vision coverage were addressed with optimized runtime settings, extensive new ablations, and additional ImageNet experiments, leaving only a subjective concern about novelty framing.
+ Reviewer kguY (original score 6) would like to increase his/her score, since the rebuttal clarified the runtime cost of NBS, provided a precise definition of the utility function, and gave a detailed explanation of complex momentum, resolving their technical questions.

---

### Decision · Program_Chairs · 2026-01-26

Accept (Poster)